# Cleavage of CAD by caspase-3 determines the cancer cell fate during chemotherapy

Jingsong Ma[1,2,8], Jiabao Zhao[3,8], Chensong Zhang [3], Jinshui Tan[1,2], Ao Cheng[4], Zhuo Niu[4], Zeyang Lin[5], Guangchao Pan[1,2], Chao Chen[6], Yang Ding[6], Mengya Zhong[7], Yifan Zhuang[1], Yubo Xiong[1,2], Huiwen Zhou[1,2], Shengyi Zhou[1,2], Meijuan Xu[1,2], Wenjie Ye[1,2], Funan Li[6] ✉, Yongxi Song[4] ✉, Zhenning Wang [4] ✉ & Xuehui Hong [1,2] ✉

Metabolic heterogeneity resulting from the intra-tumoral heterogeneity mediates massive adverse outcomes of tumor therapy, including chemotherapeutic resistance, but the mechanisms inside remain largely unknown. Here, we find that the de novo pyrimidine synthesis pathway determines the chemosensitivity. Chemotherapeutic drugs promote the degradation of cytosolic Carbamoyl-phosphate synthetase II, Aspartate transcarbamylase, and Dihydroorotase (CAD), an enzyme that is rate-limiting for pyrimidine synthesis, leading to apoptosis. We also find that CAD needs to be cleaved by caspase-3 on its Asp1371 residue, before its degradation. Overexpressing CAD or mutating Asp1371 to block caspase-3 cleavage confers chemoresistance in xenograft and Cldn18-ATK gastric cancer models. Importantly, mutations related to Asp1371 of CAD are found in tumor samples that failed neoadjuvant chemotherapy and pharmacological targeting of CAD-Asp1371 mutations using RMY-186 ameliorates chemotherapy efficacy. Our work reveals the vulnerability of de novo pyrimidine synthesis during chemotherapy, highlighting CAD as a promising therapeutic target and biomarker.

Gastric cancer (GC) and colorectal cancer (CRC) are two major types of malignant tumors that occur in the digestive system[1,2]. With increasing incidence rates, these cancers are highly invasive, have a high metastasis rate, and a poor prognosis. Unfortunately, GC and CRC are difficult to treat due to limited effective targets, with only a few patients benefiting from immunotherapy and targeted therapy, both of which have varying clinical effects[3,4]. This leaves chemotherapy as the primary treatment option. However, GC and CRC chemotherapy often leads to chemoresistance, which largely reduces the long-term clinical benefit for patients. For instance, less than 40% of patients show an objective response rate (ORR) to chemotherapy, and only about 5% of patients achieve a pathologic complete response after chemotherapy. As a result, the mortality rates of GC and CRC remain persistently high[5]. Therefore, it is crucial to understand the mechanisms that confer GC and CRC resistance to chemotherapy drugs in order to reduce the incidence of chemoresistance.

Recent studies have suggested that it is the high intrinsic heterogeneity of GC and CRC, particularly the dysregulation of metabolic

[1]Department of Gastrointestinal Surgery, Zhongshan Hospital of Xiamen University, School of Medicine, Xiamen University, Xiamen, China. [2]Xiamen Municipal Key Laboratory of Gastrointestinal Oncology, Xiamen, China. [3]State Key Laboratory for Cellular Stress Biology, Innovation Centre for Cell Signalling Network, School of Life Sciences, Xiamen University, Xiamen, China. [4]Department of Surgical Oncology and General Surgery, Key Laboratory of Precision Diagnosis and Treatment of Gastrointestinal Tumors, Ministry of Education, The First Affiliated Hospital of China Medical University, Shenyang, China. [5]Department of Pathology, Zhongshan Hospital of Xiamen University, School of Medicine, Xiamen University, Xiamen, China. [6]Fujian Provincial Key Laboratory of Innovative Drug Target Research, School of Pharmaceutical Sciences, Xiamen University, Xiamen, China. [7]Department of Radiology, The First Affiliated Hospital of Xiamen University, School of Medicine, Xiamen University, Xiamen, China. [8]These authors contributed equally: Jingsong Ma, Jiabao Zhao. ✉e-mail: fnlee5@xmu.edu.cn; yxsong@cmu.edu.cn; znwang@cmu.edu.cn; hongxu@xmu.edu.cn

reprogramming inside, may confer them resistant to chemotherapy drugs[6–8]. Although still preliminary, it was shown that the synthesis of purine and pyrimidine nucleotides may play a role in the dysregulated metabolic pathways leading to drug resistance. Clinical evidence suggests that in patients with locally advanced rectal cancer (LARC) who exhibit heterogeneous responses to the preoperative neoadjuvant chemoradiotherapy (nCRT), showed a positive correlation between the levels of enzymes responsible for nucleotide synthesis in tumor tissues, the concentrations of nucleotide biosynthesis-related metabolites in blood, and the degree of chemoresistance. In addition, nucleoside supplementation has been observed to protect cultured LARC cells from the effects of the chemotherapy drug 5-fluorouracil (5-FU)[9].

Upregulation of nucleotide synthesis is also crucial for the survival of GC and CRC cells, which is one of the leading causes of poor outcomes in chemotherapy[10,11]. For example, in high metastatic colorectal cancer (mCRC) tissues, various intermediate metabolites in pyrimidine biosynthesis have been found to be significantly upregulated, which can support the survival of mCRC cells by acting as an alternative fuel for the glycolytic pathway under nutrient-deprivation microenvironments observed in these tumor tissues[12,13]. Such a mechanism may also play a role in the survival of CRC cells during metastasis, as pyrimidine biosynthesis driven by phosphoenolpyruvate carboxykinase 1 (PCK1) was significantly upregulated in the mCRC liver colonization model using patient-derived metastatic tumor xenografts. Consistently, administration of leflunomide to inhibit the pyrimidine biosynthesis through inhibiting dihydroorotate dehydrogenase (DHODH) largely blocked the liver metastasis of mCRC[14].

Promotion of nucleotide synthesis also contributes to the proliferation of GC and CRC cells. We have previously found that GC cells express high levels of U2AF homology motif kinase 1 (UHMK1), which phosphorylates nuclear receptor coactivator (NCOA3) at Ser1062 and Thr1067. The phosphorylated NCOA3 interacts with the transcription factor, activating transcription factor 4 (ATF4), promoting the nuclear translocation and transcriptional activity of ATF4. This leads to an upregulation of de novo purine metabolism-related target genes, ultimately promoting the development and progression of GC[15]. Interestingly, the NCOA3-ATF4 pair is also necessary for the increase of de novo purine synthesis in hepatocarcinoma (HCC), in which its negative regulator, dual-specificity tyrosine phosphorylation-regulated kinase 3 (DYRK3), which phosphorylates NCOA3 and blocks the nuclear translocation of ATF4, is significantly downregulated[16]. At least in HCC, such an upregulation of nucleotide synthesis has been shown to be responsible for the 5-FU resistance of HCC cells[17].

Here, we identify that the multifunctional enzyme CAD acts as a rate-limiting enzyme for de novo pyrimidine synthesis[18,19], and is a substrate of caspase-3 when caspase-3 is activated by chemotherapy drugs. The caspase-3-mediated cleavage and degradation of CAD is a necessary step for the death of cancer cells induced by chemotherapy drugs in GC and CRC. Overexpression of CAD or its mutations with its cleavage site 1371 aspartic acid (Asp1371, or D1371) mutated, rendering it incapable of being cleaved by caspase-3, enhances chemoresistance. Moreover, we find that pharmacologically inducing the degradation of the cleavage-resistant CAD can increase the sensitivity of GC and CRC to clinically relevant chemotherapy agents. Therefore, CAD is a critical suppressor of chemotherapy-induced cell death and a promising molecular target for treating drug resistance.

## Results

### CAD levels decrease during chemotherapy

We started by analyzing the impacts of chemotherapy drugs on pyrimidine metabolism in different GC and CRC cells, such as HGC27, MKN45, HCT116, and SW480, which are known to be sensitive to chemotherapy. We treated them with common antitumor drugs,

including 5-FU, oxaliplatin (Oxa), doxorubicin (Dox), or paclitaxel (PTX), at a dose that would induce apoptosis, which was evidenced by a significant increase in cleaved-poly (ADP-ribose) polymerase (c-PARP) and P53 (Fig. 1a and Supplementary Fig. 1a). As expected, we also observed a significant upregulation of H2A.X phosphorylation levels in cells treated with 5-FU as a representative chemotherapy drug (Fig. 1b–d and Supplementary Fig. 1b–d), consistent with an increased DNA double-strand break caused by the drug[20]. As a control, we observed no increase in pyroptosis (determined by the lactate dehydrogenase (LDH) release assay) under the 5-FU treatment (Supplementary Fig. 1e), which is consistent with the observations that 5-FU exclusively induces apoptosis[21]. We also observed a significant decrease in pyrimidine nucleotides (e.g., CTP, UTP, dCTP, and dTTP) levels through performing the mass spectrometry (Fig. 1e and Supplementary Fig. 1f). This scarcity of dNTP/NTP pool resulted in the arrest of the cell cycle in the S-phase (Supplementary Fig. 1g). Importantly, supplementation of uridine, or deoxythymidine (dT), and deoxycytidine (dC) that can help replenish the products of de novo pyrimidine synthesis pathway, prevented the 5-FU-induced apoptosis (Fig. 1b–d and Supplementary Fig. 1b–d), while supplementation of dihydroorotate (DHOA) that cannot enter and be converted into the products of the de novo pyrimidine synthesis pathway, failed to do so[22]. These results indicate that chemotherapy drugs strongly disrupt the de novo pyrimidine synthesis pathway in GC and CRC cells.

We therefore determined the changes of enzymes involved in the de novo pyrimidine synthesis in these cells under the treatment of chemotherapy drugs[23], and found a significant reduction in the levels of the tri-functional enzyme CAD. Such decrease in CAD levels appeared unique, as other pyrimidine synthesis-related enzymes, such as DHODH and uridine 5′-monophosphate synthase (UMPS) remained unaffected (Fig. 1a and Supplementary Fig. 1a). We also observed a positive correlation between the decrease in CAD and the occurrence of apoptosis in GC and CRC cells. In particular, we found that the protein levels of CAD began to decrease at the same time as the levels of P53 and c-PARP started to increase in cells treated with 5-FU (Fig. 1f, g and Supplementary Fig. 1h, i). Similar to chemotherapy drugs, ionizing radiation (IR) induces tumor cell death by causing DNA damage[24]. The reduction of CAD and the increase of P53 and c-PARP exhibit a dose-dependent manner in irradiated cells. Furthermore, restoring CAD protein levels through ectopic expression can rescue cells from IR-induced apoptosis (Supplementary Fig. 1j, k).

Moreover, the protein levels of CAD began to decrease at the concentration that exactly matched the threshold of 5-FU needed for apoptosis induction. Furthermore, restoration of the protein level of CAD by ectopic expression of this protein largely compensated for the scarcity of pyrimidine nucleotides under the 5-FU treatment, and also blocked the 5-FU-induced apoptosis (Fig. 1h and Supplementary Fig. 1l), H2A.X phosphorylation, as well as the S-phase cell cycle arrest (Fig. 1b–d and Supplementary Fig. 1b–d, g). Consistently, gene expression difference analysis in the 5-FU-treated GC cells with CAD ectopically expressed shows a significant upregulation of genes related to the DNA replication, signal transduction in response to DNA damage, DNA repair, and DNA recombination (Fig. 1i and Supplementary Fig. 1m). The causal effects of CAD decrease and the cancer cell apoptosis could also be observed in the subpopulations of apoptotic, suspended (detached from culture dish) HGC27, MKN45, HCT116, and SW480 cells, in which the levels of CAD were significantly lower as compared to the attached, live subpopulations, although they are both treated with 5-FU (Fig. 1j and Supplementary Fig. 1n). As HGC27, MKN45, HCT116, and SW480 are considered heterogeneous and are commonly used to establish chemoresistant cell lines, the data above suggest that reducing the levels of CAD could help increase chemosensitivity and promote apoptosis in GC and CRC cells.

We also determined the relationship between CAD decrease and the chemotherapy-induced apoptosis in tissue levels. As shown in

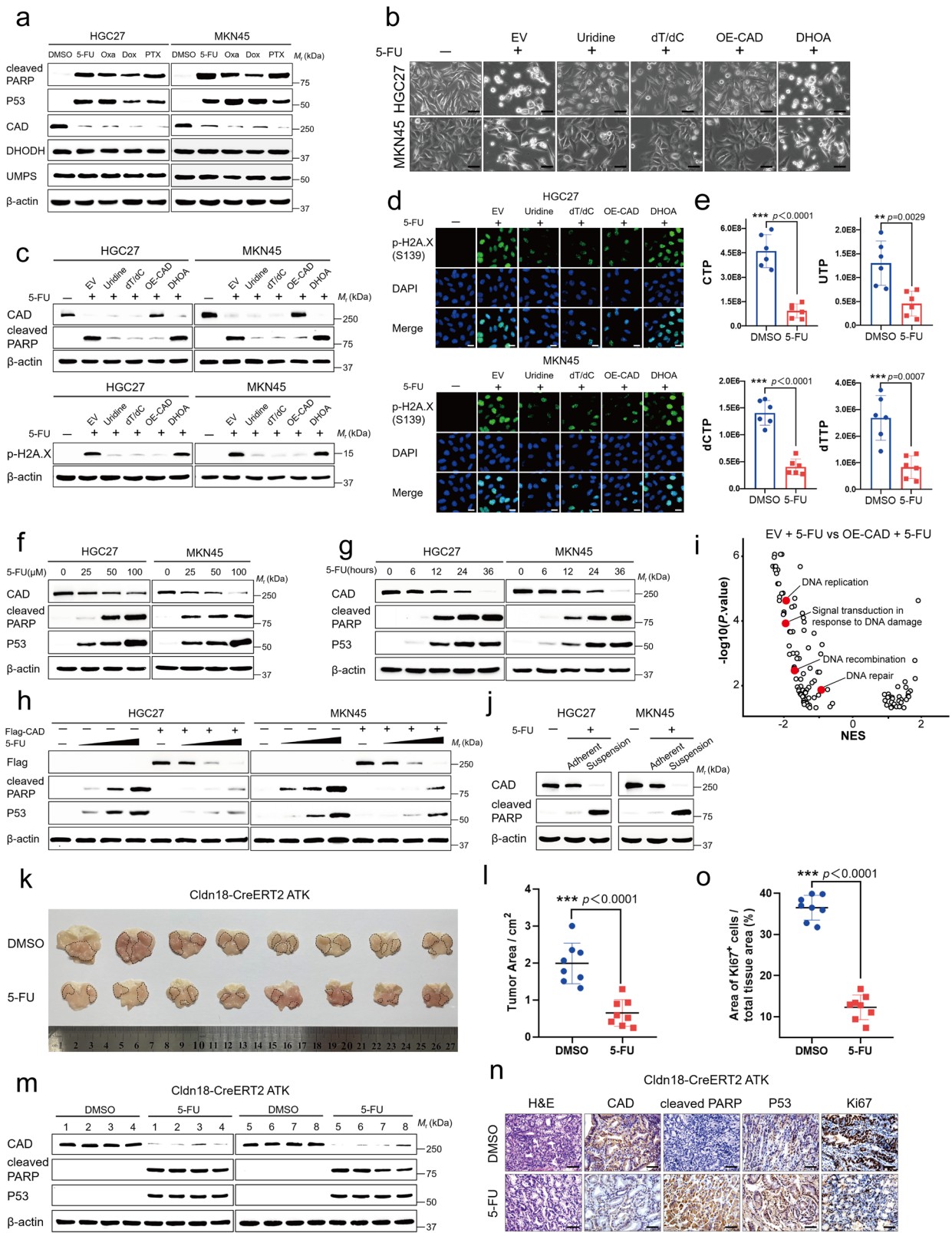

Fig. 1k, l, we found that in the *Cldn18*-CreERT2; *Apc*fl/fl; *Trp53*fl/fl; Kras[G12D] (a.k.a. Cldn18-ATK) mice, which were generated by specifically knocking out APC and Trp53, and expressing Kras[G12D] in the gastric epithelium using a Cre recombinase under the *Cldn18* promoter[25], the CAD levels in hyperplastic gastric tumor tissues were significantly reduced after the treatment with 5-FU, resulting in a significant

reduction in tumor size. Such a reduction of CAD was accompanied by an overall increase in apoptosis (determined by the levels of c-PARP and P53) and a decrease in proliferation (determined by the levels of Ki67) in these tissues, as seen in GC cell lines above (Fig. 1m–o). Therefore, we identified CAD as a critical suppressor of chemotherapy-induced apoptosis.

**Fig. 1 | CAD plays a key role in chemotherapy resistance. a** Western blot showing cleaved PARP, P53, CAD, DHODH and UMPS protein levels in HGC27 and MKN45 cells were treated with 5-FU (50 μM), Oxa (20 μM), Dox (20 μM) and PTX (10 nM) for 24 h. **b** Representative photomicrographs of HGC27 and MKN45 cells expressing empty vector (EV), CAD, or supplemented with uridine (100 μM), dT/dC (20 μM) and DHOA (250 μM), then treated with 5-FU (50 μM). Scale bar, 50 μm. **c, d** Representative Immunoblot analysis (**c**) with CAD, cleaved PARP, and p-H2A.X antibody and Immunofluorescence (**d**) with p-H2A.X antibody of HGC27 and MKN45 cells as described in (**b**). Scale bar, 20 μm. **e** Relative quantification of CTP, UTP, dCTP, and dTTP levels in MKN45 cells treated with 5-FU (50 μM) or DMSO for 24 h. *n* = 6 biologically independent samples per group. **f, g** Protein expression of CAD, cleaved PARP and P53 were determined after HGC27 and MKN45 cells were treated with different concentrations and time period of 5-FU. **h** HGC27 and MKN45 cells overexpressing Flag-CAD were treated with different concentrations of 5-FU

for 24 h. Flag, cleaved PARP and P53 expression were detected. **i** Normalized enrichment scores for all gene ontology sets differentially expressed. **j** Western blot analysis of CAD, cleaved PARP protein expression in 5-FU (100 μM) treatment still attached and suspended cells. **k, l** Whole-mount images (**k**) and the statistical analysis (**l**) of stomach from untreated or 5-FU-treated Cldn18-ATK mice. Tumors were marked with black dotted lines. *n* = 8 biologically independent samples per group. **m** Protein expression of CAD, cleaved PARP and P53 were determined in tumor samples. **n** Representative images of H&E and IHC staining of stomach sections. Scale bar, 50 μm. **o** Quantification and statistical analysis of Ki67 staining images in (**n**). *n* = 8 biologically independent samples per group. Data in (**a, c, f, g** and **j**) derive from the same biological experiment but were processed on parallel gels (see Source Data for gel-specific details). Data were expressed as means ± SD, two-tailed Student's *t*-test. \*\**p* < 0.01, \*\*\**p* < 0.001.

## Caspase-3 mediates the decrease of CAD

We next determined through which mechanisms the protein level of CAD is decreased during chemotherapy. We found that the mRNA levels of CAD, as determined by the real-time quantitative PCR (RT-PCR) analysis, remained unchanged in GC and CRC cells after the treatments of chemotherapy drugs (Supplementary Fig. 2a), excluding the possibility that chemotherapy drugs downregulate CAD through the suppression of CAD expression. The above hypothesis was further confirmed using CAD mRNA stability assays (Supplementary Fig. 2b). We also discovered that the decrease in CAD protein levels could not be rescued by either the proteasome inhibitor, MG-132, or the lysosomal inhibitor, chloroquine (CQ) (Supplementary Fig. 2c, d). This led us to suspect that the CAD protein was modified in a way that made it unrecognizable by its antibody due to changes in its recognition motif or molecular weight. Indeed, we found that the Flag-tagged CAD protein, when ectopically expressed in GC and CRC cells, could still be probed under 5-FU treatment using the anti-Flag antibody for immunoblotting, albeit at a lower molecular weight (Fig. 2a and Supplementary Fig. 2e), suggesting that CAD is cleaved in these cells. We also explored other apoptosis inducers, such as another chemotherapy drug Oxa, Venetoclax (Vene), Staurosporine (STS) and Raptinal (Rap), and found that they also induce CAD cleavage (Supplementary Fig. 2f). Therefore, CAD cleavage can occur in response to various cytotoxic agents, including chemotherapy drugs.

We next investigated how CAD is cleaved when treated with 5-FU. We observed that the cleavage of various caspases, such as caspase-3, caspase-6, and caspase-7 which are known to be induced by 5-FU, occurred simultaneously with CAD cleavage (Fig. 2b and Supplementary Fig. 2g). Inhibition of caspases by Z-VAD-FMK, which is a pan inhibitor, blocked the cleavage of CAD in the GC and CRC cell lines, suggesting that caspases is responsible for CAD cleavage under 5-FU treatment (Fig. 2c and Supplementary Fig. 2h). We, therefore, determined which caspase is responsible for the CAD cleavage. Among the effector caspases, we found that overexpression of caspase-3 and caspase-6, but not caspase-7, led to the cleavage of CAD, even in the absence of 5-FU (Fig. 2d, e and Supplementary Fig. 2i). The caspase-3/-6-mediated CAD-cleavage was confirmed in cell-free systems, where the bacterially expressed and purified caspase-3 and caspase-6[26,27], but not caspase-7, could cleave the Flag-tagged CAD which was expressed and eluted in HEK-293T cells (Fig. 2f and Supplementary Fig. 2r, s; see validation data in Supplementary Fig. 2l-n, as PARP1[28], RIPK1[29] and HDAC3[30], the reported substrate of caspase-3, caspases-6, and caspase-7, are respectively used). Z-DEVD-FMK, a caspase-3-specific inhibitor, or Ac-DNLD-CHO, a caspase-3/7-specific inhibitor, significantly blocked the 5-FU-induced CAD cleavage (lanes 3 and 5 of Fig. 2g and Supplementary Fig. 2t). Although overexpressed or recombinant caspase-6 was able to cleave CAD, the endogenous caspase-6 was not required for the 5-FU-induced CAD cleavage in GC and CRC cells, as the caspase-6 inhibitor, Z-VEID-FMK, did not block the 5-FU-induced CAD cleavage (lane 4 of Fig. 2g and Supplementary Fig. 2t). This might be

due to the relatively low expression of caspase-6 compared to other caspases, such as caspase-3, observed in GC and CRC cells (Supplementary Fig. 2u–w). Supporting this conclusion, we found that knockout of caspase-3, but not caspase-6, in HGC27, MKN45, HCT116, or SW480 cells prevented the cleavage of CAD caused by 5-FU. Importantly, the removal of caspase-6 in cells that lacked caspase-3 did not further attenuate the 5-FU-mediated CAD cleavage (Fig. 2h and Supplementary Fig. 2x). By reintroducing caspase-3 into the GC and CRC cells that lacked caspase-3, we were able to restore the CAD cleavage (Fig. 2i and Supplementary Fig. 2y). The mutation of caspase-3 at its Cys163 (C163) residue[24,31], which inactivates the caspase, also blocked CAD cleavage (Fig. 2j).

We also investigated the initiator caspases upstream of caspase-3 and their effect on CAD cleavage. We found that overexpression of caspase-8 or caspase-9 (Supplementary Fig. 2j, k) could lead to the cleavage of CAD, but these effects were likely to act through activating caspase-3, as neither of them could cleave CAD in the cell-free system (Supplementary Fig. 2s; see validated in Supplementary Fig. 2o–q, in which BID[32] and caspase-3[33] are used as the substrates, respectively). Consistent with the notion that 5-FU induces apoptosis through the intrinsic pathway, we observed that caspase-9, which is specifically involved in the intrinsic apoptosis pathway, was related to the CAD cleavage, as the caspase-9-specific inhibitor Z-LEHD-FMK (lane 7 of Fig. 2g and Supplementary Fig. 2t) blocked the 5-FU-mediated CAD cleavage. In comparison, caspase-8, another upstream caspase of caspase-3 that is specifically involved in the extrinsic pathway of apoptosis, did not participate in the 5-FU-mediated CAD cleavage (lane 6 of Fig. 2g and Supplementary Fig. 2t, in which Z-IETD-FMK is the caspase-8-specific inhibitor). Therefore, 5-FU activates caspase-9, which triggers caspase-3, initiating the intrinsic apoptosis pathway, and ultimately leading to CAD cleavage. Therefore, we concluded that caspase-3 is responsible for the 5-FU-mediated CAD cleavage.

We also investigated the effects of caspase-3-mediated CAD cleavage in vivo. To this end, we generated the caspase-3-knockout mice and then crossed them with the Cldn18-ATK mice to elicit hyperplastic gastric tumors. As a result, although 5-FU intervention can reduce the tumor size of caspase-3-knockout mice to a certain extent, it failed to achieve statistical significance (Fig. 2k, l), which may be due to direct cellular damage effects induced by 5-FU, and this damage occurs independently of caspase-3[34,35]. Unlike in the wildtype Cldn18-ATK mice (Fig. 1m–o), the effects of 5-FU on CAD cleavage, apoptosis induction (Fig. 2m), and the proliferation inhibition (Fig. 2n, o; assessed by Ki67 staining) were absent in caspase-3-knockout mice.

## Caspase-3 cleaves CAD at the Asp1371 site

We next determined the site(s)/residue(s) of CAD cleaved by caspase-3. It was once considered that caspase-3 had a preferred substrate motif, DEXD, consisting of Asp, Glu, and X (which could be Val(V), Thr(T), or His(H)), located at positions -3, -2, and -1 relative to the Asp site to be cleaved[33,36]. However, recent findings suggest a more flexible

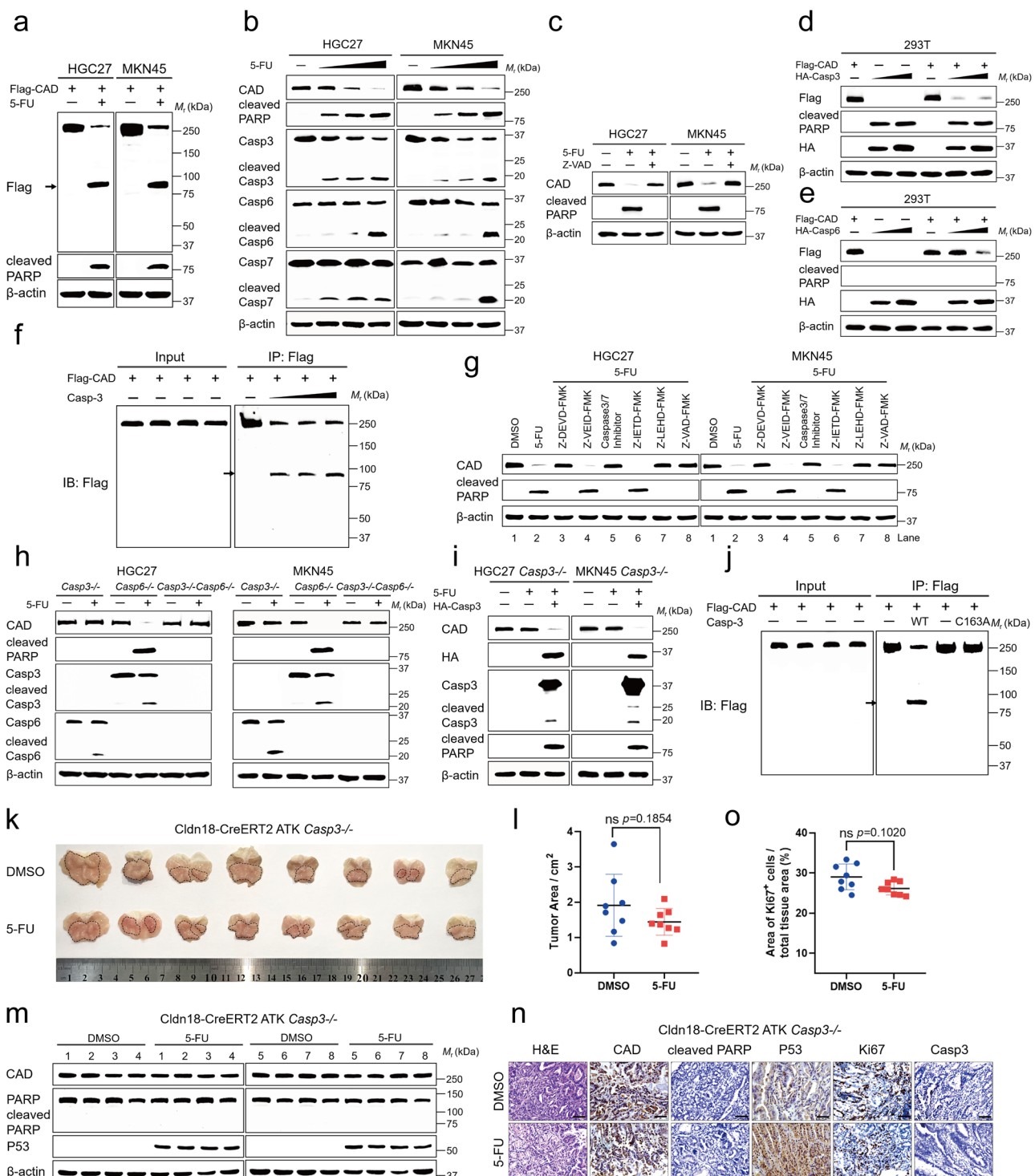

**Fig. 2 | CAD is cleaved by caspase-3 during apoptosis. a** Western blot of HGC27 and MKN45 cells transfected with the indicated CAD construct and treated for 24 h with 5-FU. **b** Immunoblotting analysis of the indicated proteins in HGC27 and MKN45 cells were treated with 5-FU with concentration gradients. **c** HGC27 and MKN45 cells were treated with 5-FU in the absence and presence of Z-VAD-FMK for 24 h. Total cell lysates were immunoblotted with CAD and cleaved PARP antibody. **d, e** HEK-293T cells were co-transfected with Flag-CAD and HA-caspase-3 (**d**) or HA-caspase-6 (**e**). Flag, cleaved PARP and HA expression were detected by immuno-blotting. **f** Western blot of in vitro cleavage reaction containing purified Flag-CAD by immunoprecipitation and recombinant active caspase-3. **g** HGC27 and MKN45 cells were treated with 5-FU in the presence of various indicated caspase inhibitors. Cells were collected and lysates were subjected to immunoblot using CAD and cleaved PARP antibody. **h** *casp3-/-*, *casp6-/-* and *casp3-/-casp6-/-* HGC27 and MKN45 cells were stimulated for 24 h with 5-FU, followed by immunoblot analysis. **i** *casp3-/-*

HGC27 and MKN45 cells were treated with 5-FU in the presence or absence of recovery of HA-caspase-3. The levels of indicated proteins were determined. **j** Western blot of in vitro cleavage reaction containing purified Flag-CAD by immunoprecipitation and recombinant inactive mutant caspase-3-C163A. **k, l** Whole-mount images (**k**) and the statistical analysis (**l**) of stomach from untreated or 5-FU-treated Cldn18-ATK *casp3-/-* mice. Tumors were marked with black dotted lines. *n* = 8 biologically independent samples per group. **m** Protein expression of CAD, PARP, and P53 were determined in tumor samples. **n** Representative images of H&E and IHC staining of stomach sections. Scale bar, 50 μm. **o** Quantification and statistical analysis of Ki67 staining images in (**n**). *n* = 8 biologically independent samples per group. Data in (**a**, **b**, **f**, **h–j**, **m**) derive from the same biological experiment but were processed on parallel gels (see Source Data for gel-specific details). Arrowhead highlights band corresponding to cleavage. Data were expressed as means ± SD, two-tailed Student's *t*-test. ns not significant.

scenario, where the cleaved site, which is Asp, is the only essential feature. For instance, human cyclic GMP-AMP synthase (h-cGAS) contains a motif, ISVD, in which Asp319 is cleaved by caspase-3[37], while gasdermin D (GSDMD) has Asp87 that is cleaved by caspase-3, which is surrounded by D (-3), A (-2), and M (-1)[38]. We, therefore, generated a series of truncated, Flag-tagged CAD mutants, individually transfected these mutants into the HEK-293T cells in which caspase-3 is ectopically expressed, and determined the cleavage of these mutants. As shown in Fig. 3a, b, we found that the CAD mutant consisting of amino acid 395–1444 (CAD$^{395-1444}$) was the only one that could be cleaved by caspase-3, whereas CAD$^{1-394}$, CAD$^{1445-1914}$, or CAD$^{1813-2224}$ could not be cleaved. Further truncation on the CAD$^{395-1444}$ showed that CAD$^{952-1512}$, but not CAD$^{591-1136}$ or CAD$^{744-1350}$, could be cleaved by caspase-3 (Fig. 3c, d). These results indicate that the amino acids between 1350 and 1445 comprise the preferred substrate motif of caspase-3. Within the region of 1350–1445, there are four Asp residues, namely Asp1363, Asp1371, Asp1421, and Asp1429 (Fig. 3e). Through individually mutating these sites to alanine (A) on the CAD$^{952-1512}$ construct, we found that Asp1371, but not the other three sites, blocked the caspase-3-mediated cleavage of CAD (Fig. 3f). Mutation of Asp1371 on full-length Flag-tagged CAD also blocked the cleavage of CAD by caspase-3, indicating that Asp1371 is the unique site on CAD to be cleaved by caspase-3 (Fig. 3g, h). As an additional validation, we performed peptide mass fingerprinting analysis on the cleaved, C-terminal product of CAD. As shown in Supplementary Fig. 3a, since the Flag tag is positioned at the C-terminus of the protein, we performed mass spectrometric analysis on a tryptic digest of the excised 90-kDa fragment. The analysis identified a pile of peptides with the N-terminal residue of Asp1371 was hit, confirming that CAD is cleaved at the Asp1371 site. We further generated an antibody that specifically recognizes the cleavage product of CAD, both the N-terminus (i.e., CAD$^{1-1371}$) and C-terminus (i.e., CAD$^{\Delta N1371}$) (Supplementary Fig. 3b; see also validation data in Supplementary Fig. 3c, d by using the ectopically expressed CAD$^{1-1371}$ and CAD$^{\Delta N1371}$ mutants). By utilizing this antibody, we confirmed that 5-FU efficiently led to the cleavage of endogenous CAD (Fig. 3i, j and Supplementary Fig. 3e, f). Together, we identified Asp1371 of CAD as the cleavage site of caspase-3. A study from over 20 years ago conducted its research under conditions of Interleukin 3 (IL-3) deprivation to identify CAD cleavage sites. As a result, the cleavage sites identified by them differ from those observed in our work. For instance, they reported that the D1143 residue, along with D1371, is necessary for caspase-3 cleavage of CAD[39]. In contrast, we found that mutation of D1143 (CAD-D1143A) does not inhibit CAD cleavage when treated with 5-FU (Supplementary Fig. 3g).

Intriguingly, we found that the cleavage of CAD promotes its degradation, as CAD$^{1-1371}$ and CAD$^{\Delta N1371}$ both showed a significantly shortened haft-life (around 8 h) than that of full-length CAD (around 15 h), when expressed in the HEK-293T cells treated with cycloheximide (CHX), an inhibitor of protein synthesis (Supplementary Fig. 3h, i). We also found that the addition of MG-132 largely blocked the degradation of cleaved CAD (both CAD$^{1-1371}$ and CAD$^{\Delta N1371}$; Fig. 3i, j and Supplementary Fig. 3e, f).

## CAD cleavage is required for the 5-FU-induced apoptosis

We next determined whether the cleavage of CAD contributes to the shortage of pyrimidine nucleotides during the 5-FU treatment. As shown in Supplementary Fig. 3b, the Asp1371 site is located near the C-terminus of the carbamoyl phosphate synthetase (CPSase) domain of CAD, which initiates de novo pyrimidine synthesis. We found that overexpression of neither CAD$^{1-1371}$ nor CAD$^{\Delta N1371}$ could restore the levels of CTP, UTP, dCTP, and dTTP under the 5-FU treatment, while full-length CAD could (Supplementary Fig. 4a), suggesting that CAD cleavage indeed inhibits de novo pyrimidine synthesis. The compartmentalization of metabolic reactions caused by supramolecular complexes formed by metabolic enzymes can significantly enhance

metabolic reactions[22,40]. To investigate the biological function of CAD-D1371A mutation, we found that CAD-D1371A has no difference in binding affinity to glutamate oxaloacetate transaminase 1 (GOT1), voltage-dependent anion-selective channel protein 3 (VDAC3), and UMPS compared to wildtype CAD (Supplementary Fig. 4b). Consistently, overexpression of CAD-D1371A significantly restored the levels of CTP, UTP, dCTP, and dTTP, and such an effect was more pronounced under 5-FU treatment compared to wildtype CAD overexpression, although the mutation does not affect the levels of metabolites under non-drug treatment with overexpression (Fig. 4a). As a result, after overexpression of CAD-D1371A, we observed more significant protection of GC and CRC cell death compared to wildtype CAD, as assessed by the cell viability, morphological changes, and H2A.X phosphorylation staining, under the treatment of 5-FU or IR (Fig. 4b–f and Supplementary Fig. 4c–h).

We next investigated the impact of CAD cleavage in chemotherapy in vivo. Through utilizing the CRISPR/Cas9-mediated targeted knockin strategy, we generated the site-directed mutant MKN45 and HCT116 cells with homozygous CAD-D1371A integrated into the genome. We found that the knockin does not alter CAD expression levels. However, the CAD-D1371A knockin significantly inhibited the cleavage of PARP during 5-FU treatment, while the cleavage of caspase-3 remained unaffected (Supplementary Fig. 4i–k). After transplanting these cells subcutaneously into nude mice which resulted in the formation of xenografts, we found that although the growth rate of the xenografts was similar in both types of CAD, the effect of 5-FU in inhibiting xenografts was significantly reduced in the xenografts formed with CAD-D1371A knockin cells (Fig. 4g, h and Supplementary Fig. 4l, m). Xenograft tumors with the heterozygous CAD-D1371A mutation demonstrated approximately 25% increased resistance to 5-FU compared to wildtype CAD (evidenced by the tumor volume); however, this resistance was less pronounced than that observed in homozygous mutations, which exhibited about 47% resistance (Fig. 4g, h and Supplementary Fig. 4n, o). We also investigated how CAD cleavage impacts chemotherapy in the Cldn18-ATK mouse model. As shown in Supplementary Fig. 4p, the results of the sequence alignment analysis showed that the D1371 site of CAD (colored red) and its flanking sequences are highly conserved in both mice and humans. We therefore generated $Cad^{D1371A/D1371A}$ knockin mice (Supplementary Fig. 4q, r), which were then crossed with the Cldn18-ATK mice. Similar to the xenograft model, we found that the $Cad^{D1371A/D1371A}$ knockin mice had similar levels of hyperplastic gastric tumor formation compared to controls. However, the effects of 5-FU interventions on CAD cleavage (Fig. 4i), as well as the extension of the lifespan of tumor-bearing mice (Fig. 4j), were all attenuated in the mice with $Cad^{D1371A/D1371A}$ knockin. These findings demonstrate that CAD cleavage plays a critical role in mediating the effects of chemotherapy.

## CAD mutations predict chemoresistance in cancer patients

As chemoresistance is a common occurrence during the treatment for GC and CRC[41], we examined whether the failure of CAD cleavage occurs in patients and contributes to their chemoresistance. We have recently obtained some GC tissue samples (formalin-fixed and paraffin-embedded, FFPE) from 16 patients who showed no benefits after neoadjuvant or adjuvant chemotherapy and agreed to surgical excision (details of each patient are provided in Supplementary Table 1). Through performing whole-exome sequencing (WES), we found that samples from two patients—patient #9 and patient #14—had mutations on the Asp1371 site of CAD. Samples from patient #9 had a mutation to Tyr (Y), with the genomic sequence changing from GAT to TAT, and samples from patient #14 had a mutation to Glu (E), with the genomic sequence changing to GAA (Supplementary Fig. 5a–h). We further performed crystal digital droplet PCR (ddPCR; which exhibited good sensitivity and specificity to monitor the amount and proportion of mutations across a wide range, as validated in Fig. 5a, b), and

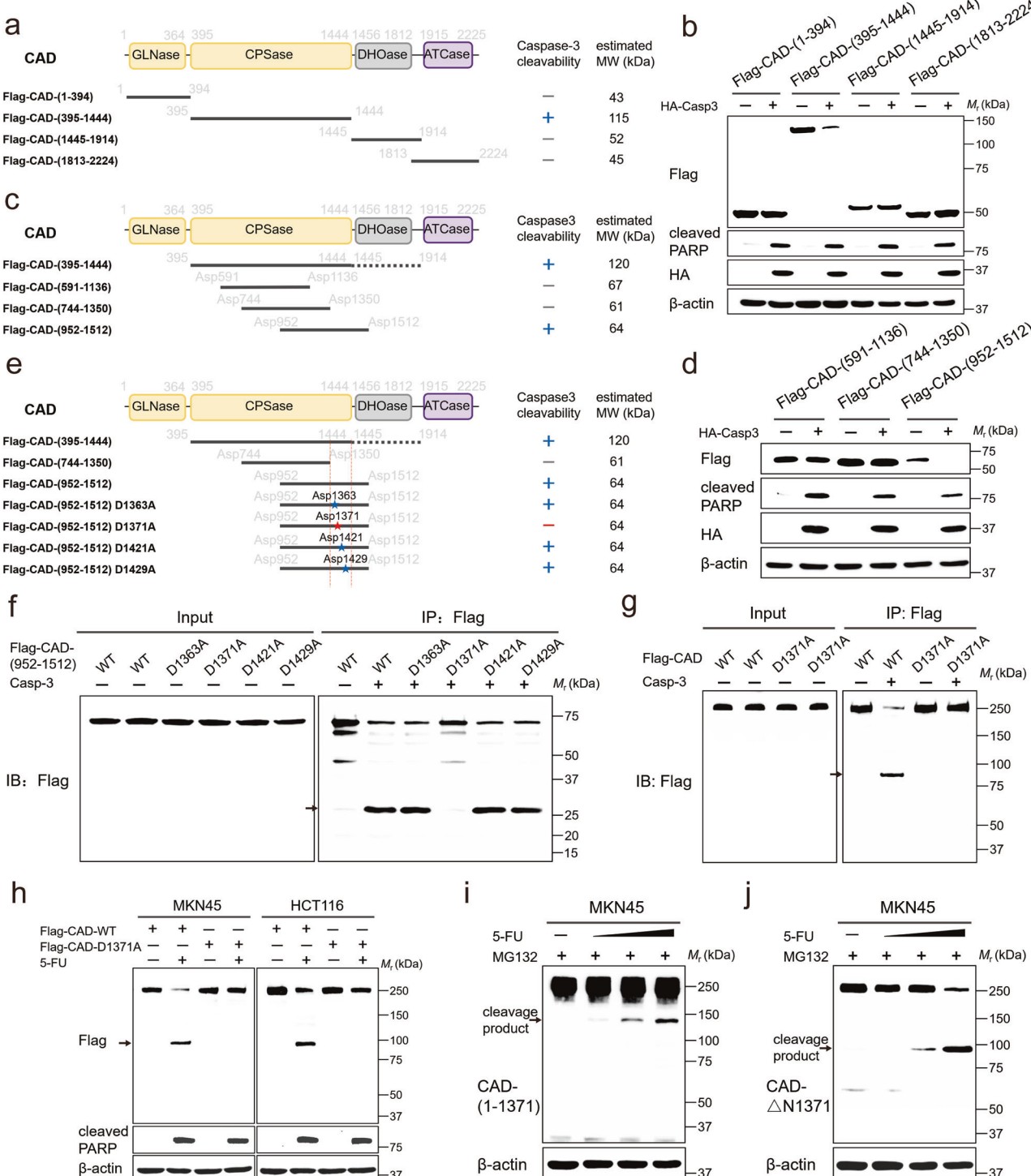

**Fig. 3 | Identification of a cleavage site in CAD. a** Schematic representation of the CAD domains. The cleavability of each construct is listed after it. "+" stands for that it is cleavable; "−" stands for that it is uncleavable. **b** HEK-293T cells were transfected with Flag-tagged CAD mutants and HA-caspase-3 for mapping the CAD cleavage site. 24 h after transfection, total cell lysates were prepared for analysis of Flag, cleaved PARP and HA by Western blot. **c** Schematic diagram of CAD protein truncation. **d** HEK-293T cells were transfected with the indicated plasmids and the expression of the three further truncated CAD were detected by anti-Flag antibody. **e** Schematic diagrams of Flag-CAD-(952-1512) and the different CAD mutants for mapping the cleavage site of CAD. Numbers indicate the position of amino acid in CAD. The pentagrams indicate the putative cleavage sites in CAD. **f, g** Western blot of in vitro cleavage reaction containing purified Flag-tagged CAD-(952-1512) point mutants of putative cleavage sites (**f**) or full-length Flag-CAD-D1371A (**g**) by immunoprecipitation and recombinant active human caspase-3. **h** Western blot of MKN45 and HCT116 cells transfected with the indicated CAD construct and treated for 24 h with 5-FU. **i, j** MKN45 cells were co-treated MG-132 and concentration gradient of 5-FU for 24 h. Cell lysates were subjected to immunoblotting using anti-CAD-(1-1371) (**i**) and anti-CAD-ΔN1371 (**j**) antibodies. Data in (**b, d, f–j**) derive from the same biological experiment but were processed on parallel gels (see Source Data for gel-specific details). Arrowhead highlights band corresponding to cleavage.

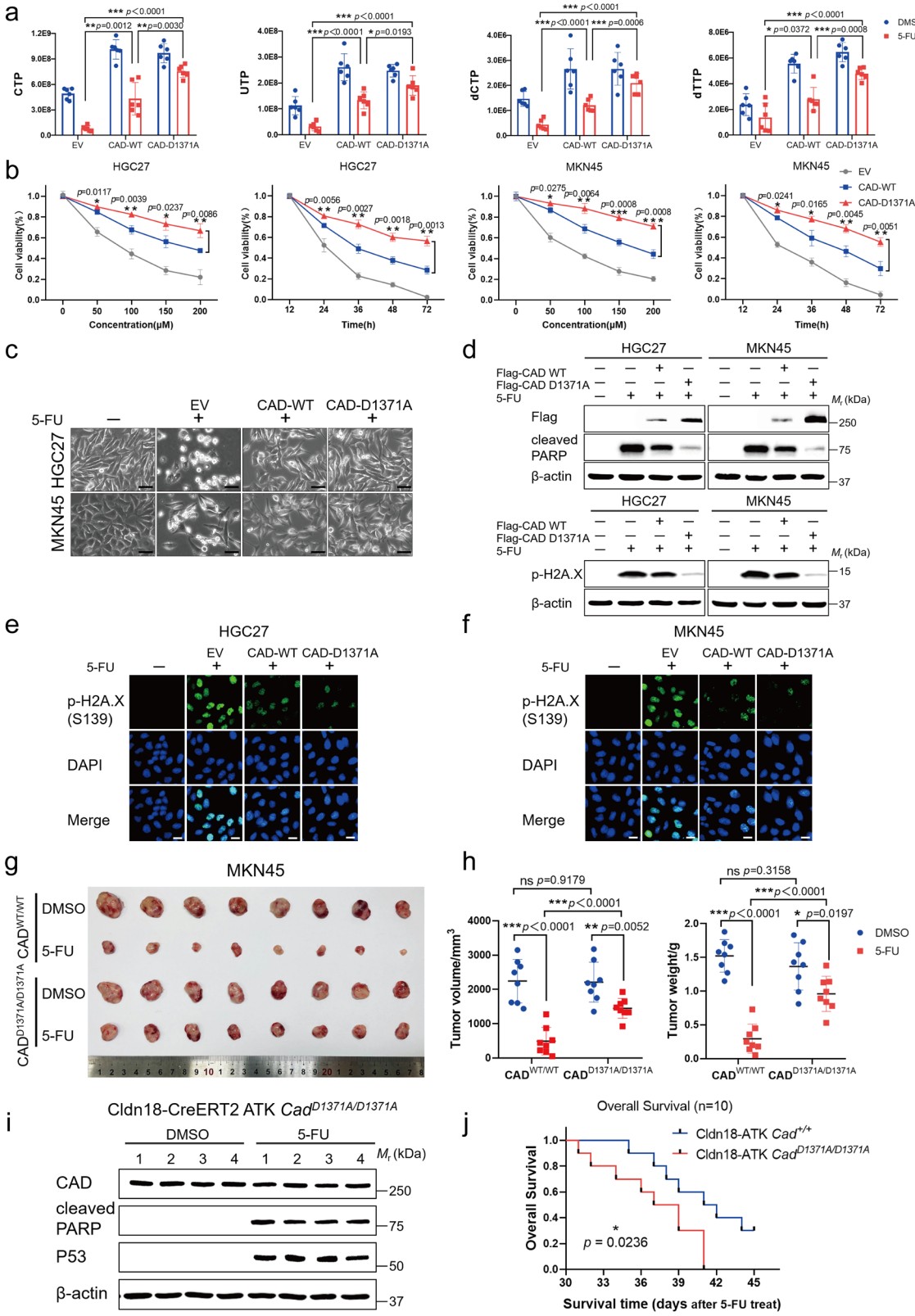

confirmed the existence of Asp1371 mutations in these chemoresistant tissues. These mutations were quantified to be approximately 15%, which conforms to previous findings that showed that chemoresistant GC tissues are highly heterogeneous[7,8]. Furthermore, we performed immunohistochemical staining of CAD levels in each patient sample and found an inverse correlation between the objective response rate (ORR) to neoadjuvant chemotherapy and the levels of CAD. As shown in Supplementary Fig. 5i, a significantly high level of CAD in GC tissues from patients showing chemoresistance (progressive disease, PD), a moderately high level from stable disease (SD), and the lowest level in partial response (PR) could be observed. As an additional control, we examined the existence and rates of Asp1371 mutations in GC tissues from chemosensitive patients. We found that among the 103 tissue samples showing positive CAD expression,

**Fig. 4 | CAD cleavage by caspase-3 is associated with chemosensitivity.**
**a** Relative quantification of CTP, UTP, dCTP, and dTTP levels in MKN45-EV, MKN45-CAD-WT and MKN45-CAD-D1371A cells treated with 5-FU or DMSO for 24 h. $n = 6$ biologically independent samples per group. **b** Cell death assays of HGC27-EV, HGC27-CAD-WT or HGC27-CAD-D1371A cells treated with different concentrations and time period of 5-FU as indicated. Repeat the same treatment and experimental design in MKN45 cells. $n = 3$ biologically independent samples per group. **c** Representative photomicrographs of HGC27 and MKN45 cells expressing EV, CAD-WT, or CAD-D1371A, then treated with 5-FU. Scale bar, 50 μm. **d–f** Representative Immunoblot analysis (**d**) with Flag, cleaved PARP, and p-H2A.X antibody and Immunofluorescence (**e**, **f**) with p-H2A.X antibody of HGC27 and

MKN45 cells as described in (**c**). Scale bar, 20 μm. **g**, **h** Size of tumors formed by injected subcutaneously with $CAD^{WT/WT}$ or homozygous $CAD^{D1371A/D1371A}$ MKN45 cells in xenograft mouse models were treated with DMSO and 5-FU separately and statistical analysis. $n = 8$ biologically independent samples per group. **i** Protein expression of CAD, cleaved PARP and P53 were determined by Western blot in tumor samples from untreated or 5-FU-treated Cldn18-ATK $Cad^{D1371A/D1371A}$ mice. **j** Kaplan–Meier survival analysis for the Cldn18-ATK $Cad^{+/+}$ and Cldn18-ATK $Cad^{D1371A/D1371A}$ mice treated with 5-FU ($n = 10$; Kaplan–Meier). $p = 0.0236$. Data in (**d**, **i**) derive from the same biological experiment but were processed on parallel gels (see Source Data for gel-specific details). Data were expressed as means ± SD, two-tailed Student's $t$-test. ns not significant, $^*p < 0.05$, $^{**}p < 0.01$, $^{***}p < 0.001$.

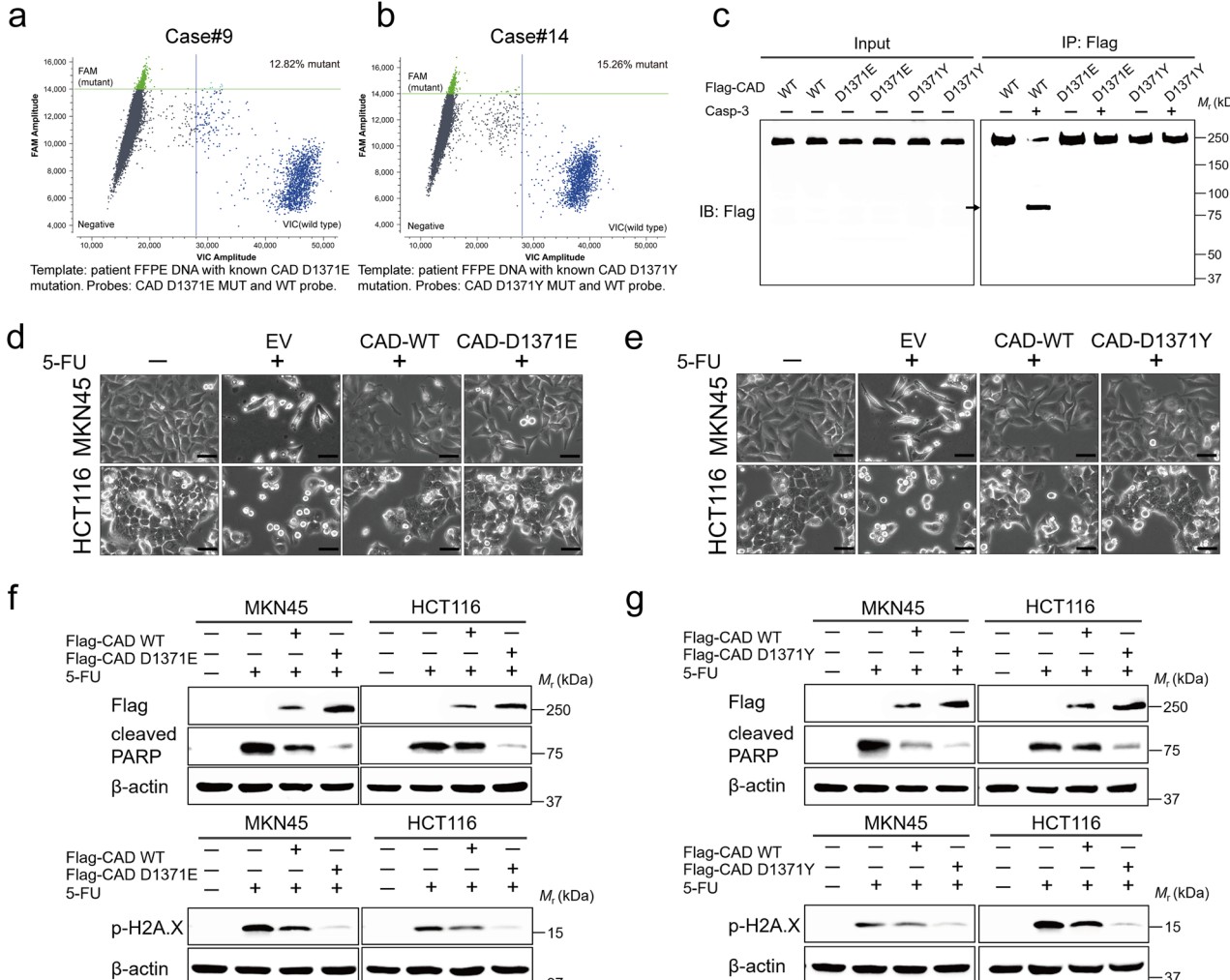

**Fig. 5 | Codon-specific CAD mutations serve as a predictive biomarker for chemotherapy resistance. a**, **b** Two-dimensional ddPCR amplitude plot showing that the assay detects the wildtype CAD gene ($x$-axis) and the D1371 mutation ($y$-axis) in patient FFPE DNA. **c** Western blot of in vitro cleavage reaction containing purified Flag-CAD, Flag-CAD-D1371E and Flag-CAD-D1371Y by immunoprecipitation and recombinant active human caspase-3. **d**, **e** Representative photomicrographs

of MKN45 and HCT116 cells expressing EV, CAD-WT, CAD-D1371E (**d**) or CAD-D1371Y (**e**), then treated with 5-FU. Scale bar, 50 μm. **f**, **g** Representative Immunoblot analysis with Flag, cleaved PARP, and p-H2A.X antibody as described in (**d**, **e**). Data in (**c**, **f**, **g**) derive from the same biological experiment but were processed on parallel gels (see Source Data for gel-specific details). Arrowhead highlights band corresponding to cleavage.

no Asp1371 mutation could be found. Therefore, CAD-D1371 mutation indeed exists at clinical levels and may be correlated with the chemosensitivity of patients.

We therefore explored the roles of CAD-D1371 mutations identified in clinical samples, i.e., the CAD-D1371E and CAD-D1371Y, on protecting the cleavage by caspase-3. As shown in Fig. 5c, we found that these two mutations indeed blocked the cleavage by caspase-3 in a cell-

free system. Furthermore, when expressed in GC and CRC cells, the CAD-D1371E or CAD-D1371Y mutants demonstrated significant protection against cell death and inhibited H2A.X phosphorylation when treated with 5-FU, compared to the wildtype CAD (Fig. 5d–g and Supplementary Fig. 5j, k). These results indicate a functional role of the CAD-D1371 mutations observed in clinical samples in preventing cancer cell death during chemotherapy.

## Development of RMY-186 for the degradation of CAD mutants

We, therefore, set out to develop a way that could downregulate the expression of CAD-D1371 mutants in order to overcome chemotherapy resistance due to their inability to be cleaved by caspase-3. We generated the MKN45 and HCT116 cells that expressed Flag-tagged CAD-D1371A, and utilized these cells to screen a house compound library. Among all compounds screened, we found that 24 compounds showed significant suppression of CAD-D1371A protein levels, as determined by immunoblotting (Supplementary Table 2 and Supplementary Fig. 6a, b). From these compounds, we identified RMY-186 (2-(4-(3-bromobenzyl)-piperazin-1-yl)-3-hydroxy-4H-chromen-4-one) as the most potent hit for suppressing CAD-D1371A expression (Fig. 6a). RMY-186 also demonstrates a general suppression of CAD, as it significantly down-regulated the protein levels of wildtype CAD, the CAD-D1371E and CAD-D1371Y mutants, in addition to the CAD-D1371A (Fig. 6b and Supplementary Fig. 6c, d). Microscale thermophoresis (MST) assays confirmed a direct interaction between RMY-186 and CAD-D1371A protein in vitro, with a $K_d$ of $6.78 \times 10^{-6}$ mol/L (Fig. 6c), which is close to the concentrations of RMY-186 used in cells that efficiently down-regulate CAD. We also determined how RMY-186 downregulates CAD. We found that treatment with MG-132 could restore the protein levels of ectopically expressed CAD, both the wildtype and its Asp1371 mutants, in HEK-293T, GC, and CRC cells under the treatment of RMY-186. In addition, we observed increased ubiquitination of the CAD protein in these cells (Fig. 6d, e and Supplementary Fig. 6e–h), suggesting that RMY-186 promotes the degradation of CAD.

We have also identified the interacting interface between CAD and RMY-186 using the Alphafold2 and CavityPlus server[42,43]. After analyzing the surface feature matching, binding energy, and potential hydrogen bonds and non-covalent interactions, we identified four potential binding pockets on CAD: Pocket 1 (Glu(E)513 and Arg(R)515), Pocket 2 (Asp(D)622 and Glu(E)630), Pocket 3 (Pro(P)741 and Trp(W)743), and Pocket 4 (Glu(E)1214 and Arg(R)1221). We then individually mutated these pockets to determine the effects of RMY-186 on the promotion of CAD ubiquitination. We found that a double mutation of CAD-P741 and CAD-W743 (CAD-P741/W743A), which disrupts Pocket 3, almost completely blocked the effects of RMY-186 on CAD ubiquitination (Fig. 6f and Supplementary Fig. 6i), these data further indicate that RMY-186 directly and specifically binds to CAD, with P741 and W743 serving as key residues in this interaction. As depicted, the active pocket of CAD protein containing the Pro741 (P741) and Trp743 (W743) residues plays a crucial role in facilitating the binding of RMY-186. Pro741 forms the structural foundation for docking, and Trp743 forms two hydrogen bonds with the hydroxyl group of RMY-186, with lengths of 1.9 Å and 2.4 Å, through its carboxyl and amine groups, respectively. In addition, the ketone group of RMY-186 also forms a 2.5 Å hydrogen bond with the carboxyl group (Fig. 6g and Supplementary Fig. 6j). As Pocket 3 is located away from the Asp1371 residue (Supplementary Fig. 6k), these data may explain the indiscriminate suppression of RMY-186 toward CAD and its Asp1371 mutants.

Moreover, we also attempted to identify the E3 ubiquitin ligase mediating the ubiquitination of CAD induced by RMY-186. The proteomics mass spectrometry data of Flag-CAD immunoprecipitates indicate that Dual E2 ubiquitin-conjugating enzyme/E3 ubiquitin-protein ligase BIRC6 or ubiquitin-protein ligase E3C (UBE3C) were most likely involved in the ubiquitination of CAD (Fig. 6h, i and Supplementary Data). Consistently, RMY-186 significantly reduced the half-life of CAD: CAD-D1371A from 13 h to 8 h (Fig. 6j, k), CAD-D1371E from 12 h to 8 h, and CAD-D1371Y from 12 h to 6 h when expressed in the HEK-293T cells (Supplementary Fig. 6l–o). Therefore, RMY-186 directly promotes the degradation of CAD, regardless of its mutation.

## RMY-186 enhances chemosensitivity

Finally, we assessed the impact of RMY-186 on chemotherapy. In line with its role in promoting the degradation of ectopically expressed

CAD, we observed a significant decrease in endogenous CAD in both GC and CRC cells treated with RMY-186. RMY-186 alone did not promote the cell death of GC and CRC cells. However, when used in combination with 5-FU, it significantly increased apoptosis. Such effects could be recapitulated by knocking down CAD in these cells, suggesting that the acceleration of CAD degradation promotes cell death during chemotherapy. These observations are also consistent with our findings that CAD-mediated de novo pyrimidine synthesis is crucial for the survival of GC and CRC cells, particularly under the treatment of chemotherapeutic drugs that result in a shortage of pyrimidine nucleotides (Fig. 1 and Supplementary Fig. 1). Importantly, we found that RMY-186, when used in combination with 5-FU or Oxa, could readily induce apoptosis in the MKN45 and HCT116 cells carrying the CAD-D1371A knock-in mutation, which displayed resistance to cell death when treated with chemotherapy drug alone (Fig. 7a, b and Supplementary Fig. 7a–c; evidenced by the levels of c-PARP, P53, and the phosphorylation of H2A.X). Interestingly, we observed that these chemosensitized cells (treatment with RMY-186 or knockdown of CAD) underwent evident swelling with characteristic large bubbles from the plasma membrane (Fig. 7c, d and Supplementary Fig. 7d–f), resembling the occurrence of pyroptosis in addition to apoptosis[44,45]. Indeed, various GC and CRC cell lines, including the HGC27, MKN45, HCT116, and SW480 analyzed in this study, showed positivity for gasdermin-E (GSDME) as determined by immunoblotting (Supplementary Fig. 7g, where the non-small cell lung cancer (NSCLC) cell line NCI-H1299, known to express high levels of GSDME, was used as a positive control). In addition, we observed significant cleavage of GSDME in these chemosensitized GC and CRC cells, resulting in an increase of GSDME-N, a marker of pyroptosis, after treatment with 5-FU (Fig. 7e, f and Supplementary Fig. 7h, i). Moreover, the release of lactate dehydrogenase (LDH) and the signal of propidium iodide (PI) staining, as other markers of pyroptosis[46,47], could also be upregulated in the chemosensitized GC and CRC cells treated with 5-FU (Fig. 7g–j and Supplementary Fig. 7j–m). Therefore, we found that the promotion of CAD degradation could enhance the effects of chemotherapy by promoting GC and CRC cells death.

We also determined the chemosensitizing effects of RMY-186 in vivo. RMY-186 exhibits low acute toxic, as its $LD_{50}$ is approximately 1391 mg/kg as determined by a single intraperitoneal injection (formulated by dissolving in saline containing 5% DMSO and 1% carboxymethylcellulose, details in "Methods" section; Supplementary Fig. 7n). Repeated exposure of mice with RMY-186 by injecting RMY-186 at 500 mg/kg once a day for 7 days did not show any signs of toxicity, such as lethargy, weight loss, or other physical indications of illness. Furthermore, there was no discernible tissue damage at the later stages of the RMY-186 administration (Supplementary Fig. 7o, after a 14-day washout). We therefore treated the $Cad^{D1371A/D1371A}$;Cldn18-ATK mice with 100 mg/kg RMY-186, and found that they became sensitive to the 5-FU administration, similar to their wildtype littermates. In particular, 5-FU resulted in a 70% suppression of tumor size, along with a significant promotion of cell deaths (both apoptosis and pyroptosis) and a suppression of proliferation in tumor tissues (Fig. 7k–p).

Studies have shown that the combination of various antitumor agents with immune checkpoint inhibitors (ICIs) can enhance the immune response against specific types of tumors[48,49]. Therefore, we also investigated the effects of RMY-186 and 5-FU on the efficacy of the anti-PD-1 antibody, a classical ICI, in suppressing GC in Cldn18-ATK $Cad^{D1371A/D1371A}$ mice. We found that the combination of RMY-186 and 5-FU indeed enhances the effects of the anti-PD-1 antibody on GC suppression. This triple treatment also significantly increased $CD3^+$ and $CD8^+$ T-cell infiltration compared to other treatments (assessed by staining for CD3 and CD8a), indicating a strong enhancement of immune responses in these tumors (Supplementary Fig. 7p–r). Together, RMY-186 could serve as a promising therapy to overcome GC and

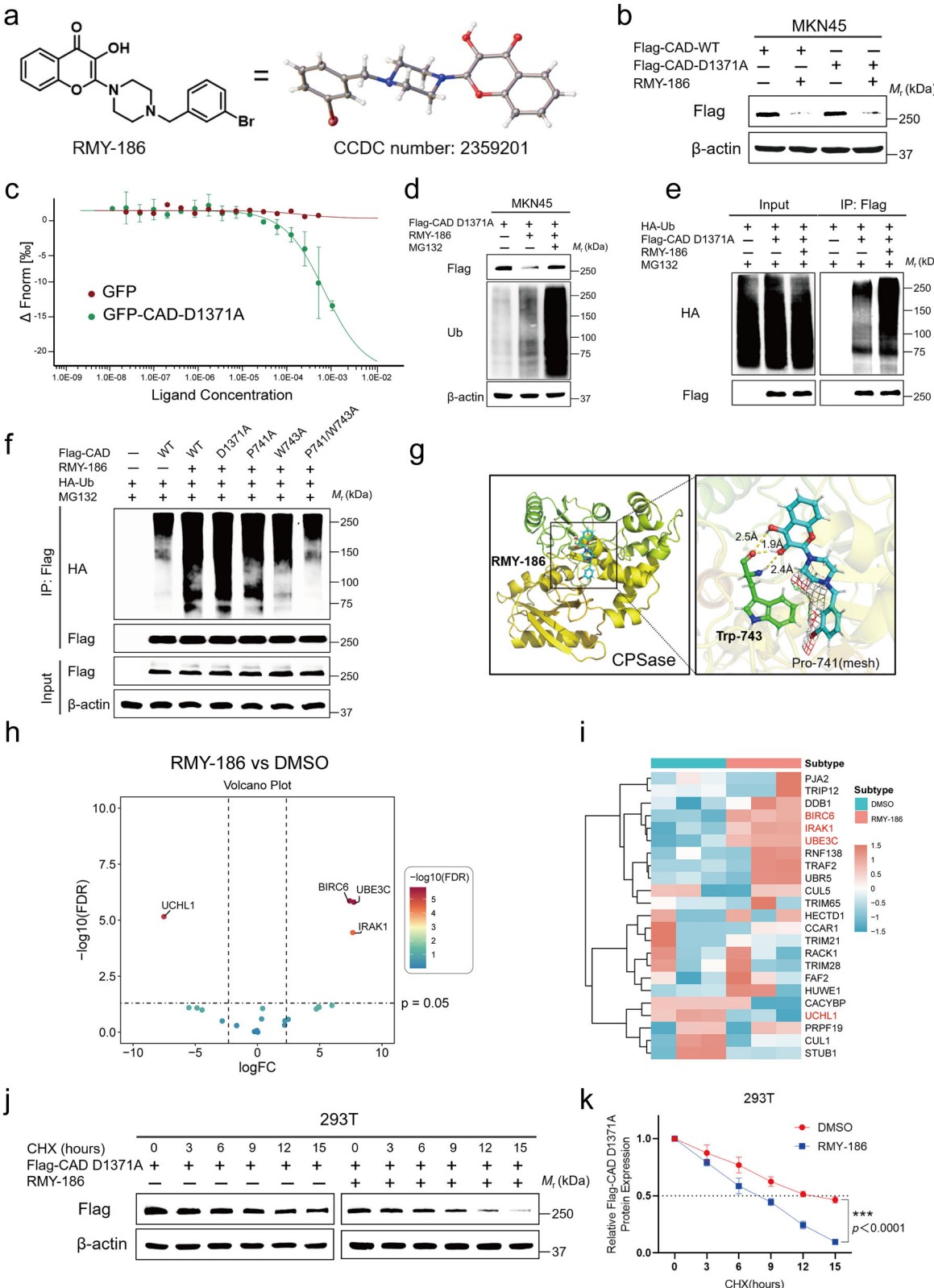

CRC resistance to chemotherapy caused by CAD mutation at its cleavage site (Supplementary Fig. 8).

## Discussion

In our research, we found that cleavage of CAD in GC and CRC cells by caspase-3 at the Asp1371 site leads to inhibition of de novo pyrimidine synthesis, which is crucial for their death when treated with different chemotherapeutic drugs. Preventing caspase-3 cleavage by introducing CAD-Asp1371 mutants also prevents the cytotoxic effects of chemotherapeutic drugs in these cells and inhibits tumor suppression in mouse models. Notably, we observed CAD-Asp1371 mutations in GC tissues from human patients; the mutation abundance accounts for approximately 15% in patients who failed neoadjuvant or adjuvant chemotherapy, potentially explaining the absence of CAD D1371

**Fig. 6 | Discovery of a potent CAD inhibitor RMY-186. a** Single-crystal structure of RMY-186. **b** Flag-CAD-WT or Flag-CAD-D1371A was transfected into MKN45 cells and then treated with RMY-186 (10 μM) for 24 h. **c** MST assay with RMY-186 and its differential selectivity toward GFP-CAD-D1371A versus GFP. *n* = 3 biologically independent samples per group. **d** Flag-CAD-D1371A overexpressed MKN45 cells were pre-treated with MG-132 (5 μM) then treated with RMY-186 (10 μM) for 24 h. Overexpressed protein of Flag-tag and ubiquitination were detected by immunoblotting. **e** Immunoprecipitation showing ubiquitination of CAD-D1371A protein in HA-Ub and Flag-CAD-D1371A overexpressed MKN45 cells treated with RMY-186. **f** Immunoprecipitation showing ubiquitination of CAD protein in MKN45 cells transfected with the indicated plasmids. **g** Computational molecular docking analysis to investigate the interaction of RMY-186 binding to CAD-D1371A (AlphaFold2). CPSase represents the local polypeptides shown. **h**, **i** Volcano plots (**h**) and Heat-map (**i**) showing the differentially E3 ubiquitination ligases in proteomics MS data. **j** HEK-293T cells were transfected with Flag-CAD-D1371A and CHX (20 μg/mL) was added to cells 24 h after transfection with and without RMY-186 (10 μM) treatment. Cells were collected after indicated time (0, 3, 6, 9, 12, or 15 h) and cell lystes were immunoblotted with anti-Flag antibody to detect the protein level change. **k** Scanning densitometry was performed for each Western blot indicated in (**j**) using ImageJ. The best-fit exponential decay lines were plotted using GraphPad Prism and the half-life can be read from it (marked by dot-dash line). *n* = 3 biologically independent samples per group. Data in (**d**–**f**, **j**) derive from the same biological experiment but were processed on parallel gels (see Source Data for gel-specific details). Data were expressed as means ± SD, two-tailed Student's *t*-test. ***$p < 0.001$.

mutation in the Catalogue of Somatic Mutations in Cancer (COSMIC) database. In comparison, these mutations were not detected in patients who showed sensitivity to chemotherapy. Regrettably, due to limited sample size and loss of some clinical data, comprehensive prognostic information remained elusive. Future research will expand the sample size to better understand the mutation's prevalence and its link to treatment outcomes. Therefore, although CAD mutations were present in only a very small percentage of all patients and were difficult to detect through unbiased, universal, and large-scale Sanger sequencing, they may dominantly determine the chemosensitivity of tumor tissue. In addition, these data remind us of the high degree of heterogeneity in GC and CRC[6,7,50,51], emphasizing the need for increased attention in future clinical practices.

Although the CAD with the Asp1371 mutation did not enhance the binding to previously reported interacting proteins, nor did it increase the de novo synthesis of pyrimidine nucleotides, whether the mutated form of CAD exhibits any other functions in cancer cells remains unknown to us. The presence of the CAD-Asp1371 mutation may help explain the failure of immunotherapy (resistance to immune checkpoint inhibitors) that often occurs during the treatment of GC and CRC[52-54]. Despite containing a high abundance of immune cells, including CD8+ T, CD4+ T, and natural killer (NK) cells inside, and being considered "immune-inflamed" or "hot" tumors, many GC and CRC patients show resistance to ICIs, similar to immunosuppressive tumor types[55-58]. It is now understood that the blockage of caspase-3-dependent cleavage of CAD would also block the immune cell-mediated, extrinsic pathway of apoptosis, because caspase-3 is required for the extrinsic pathway-mediated apoptosis by acting as the downstream factor of caspase-8 triggered by these immune cells[59-61]. In other words, the CAD-Asp1371 mutation blocks both the cell death caused by chemotherapeutic drugs through the intrinsic pathway of apoptosis and the cell death caused by immunotherapy through the extrinsic pathway of apoptosis, both of which act through caspase-3[62,63]. Consistently, it has been reported that a certain portion of patients who show chemoresistance also exhibit a failure toward immunotherapy[64]. Priming the immune response is a potential method for sensitizing the effect of ICIs[65,66]. Our chemotherapy-resistant animal model experiments demonstrated that co-treatment of 5-FU, RMY-186 and anti-PD-1 antibody recruited more CD8+ T cells to the tumor microenvironment. Therefore, the RMY-186 compound that can efficiently downregulate the Asp1371 mutant of CAD and overcome the chemoresistance may also help address the failure of GC and CRC patients to respond to immunotherapy.

Our study again emphasizes the importance of de novo pyrimidine synthesis in tumor cell survival. In addition to its cleavage, it has been previously shown that the activity of CAD is tightly regulated. For instance, a decrease in growth factors can inhibit CAD by affecting the phosphorylation of CAD through the inhibition of phosphatidylinositide 3-kinases (PI3K)-AKT2 signaling, mitogen-activated protein kinases (MAPK)-extracellular regulated protein kinase (ERK) signaling, or mTORC1-S6K signaling[67-69]. In addition, UMPS that interacts with and forms pyrimidinosome together with CAD to support the de novo pyrimidine synthesis, can be phosphorylated and inhibited by the AMP-activated protein kinase (AMPK) in low glucose[22]. Since reduced growth factors and blood glucose levels can be observed in various dietary regimes such as caloric restriction or ketogenic diet[70,71], it would be interesting to investigate whether these regimes may enhance chemosensitivity, particularly in patients who can tolerate them. Moreover, although it has not yet been tested in GC or CRC or in CAD mutant-induced chemoresistant conditions, studies have suggested several compounds that can target de novo pyrimidine synthesis. For instance, combined treatment with doxorubicin and leflunomide, which is a clinically approved inhibitor of de novo pyrimidine synthesis, has been shown to effectively suppress triple-negative breast cancer (TNBC) xenografts[20]. Additionally, BAY2402234 functions as a DHODH inhibitor and has been used to treat gliomas by causing sustained DNA damage[72,73]. Therefore, it would be valuable to develop more strategies that combine pyrimidine synthesis inhibition and cytotoxic induction to treat chemoresistant GC and CRC.

## Methods

This study complied with all relevant ethical regulations. The research protocol was approved by the Institutional Ethics Review Board of the First Affiliated Hospital of China Medical University (No. [2022]366). The export of the relevant genetic information has been registered with and approved by China's Ministry of Science and Technology (No. BF2025051116673), with the First Affiliated Hospital of China Medical University serving as the authorized institution. All participating patients provided written informed consent, including consent for the publication of individual-level data. All the animal experiments below were approved by the Animal Ethics Committee of Xiamen University (No. XMULAC20200080). The ethics protocol explicitly defined humane endpoints, stipulating a maximal permissible tumor burden not exceeding 20 mm in diameter or strictly limiting maximal tumor volume to ≤10% of the animal's body weight to ensure animal welfare. Animals were housed under a standardized 12-h light/12-h dark cycle and provided with ad libitum access to food and water throughout the experimental period.

### Cell lines

All the cell lines employed in this research were purchased from the Shanghai Institute for Biological Science (China). HEK-293T cells were cultured in DMEM. GC cell lines (AGS, HGC27, MKN28, MKN45, MKN74, and NCI-N87), CRC cell lines (HCT116, HT29, RKO, and SW480) and NSCLC cell line (NCI-H1299) were cultured in RPMI 1640. All cells were supplemented with 10% FBS, 100 U/mL penicillin, and 100 mg/mL streptomycin and cultured at 37 °C in 5% $CO_2$. As for mycoplasma contamination, the exercise check revealed that there was none. Before the study, both the HEK-293T and cancer cell lines were confirmed via short tandem-repeat DNA profiling.

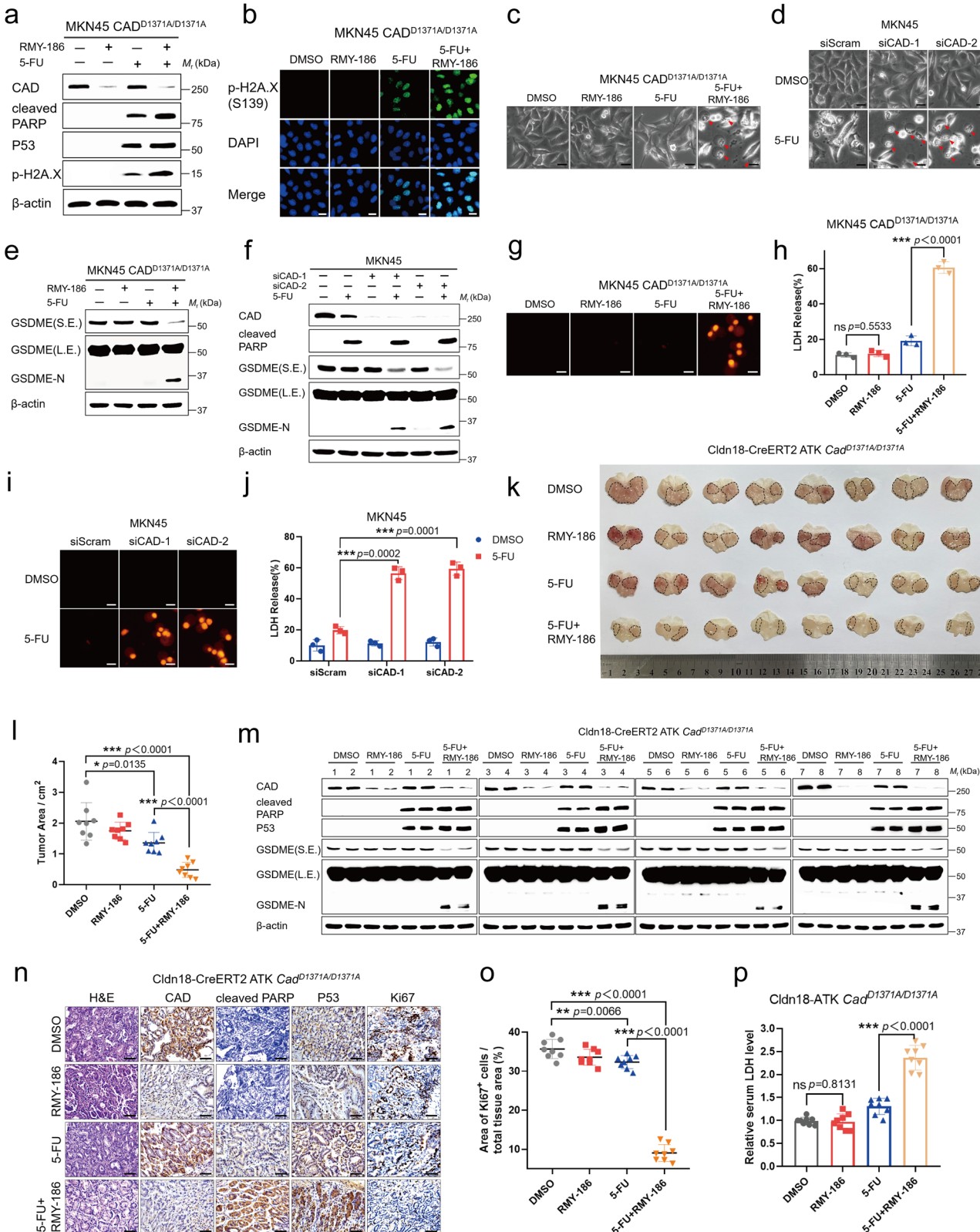

## Human specimens

Tumor tissues and adjacent normal tissues from all GC patients ($n = 119$) were collected at the First Affiliated Hospital of China Medical University between September 16, 2008, and May 22, 2022. Written informed consent was obtained from all donors, and each patient had undergone a pathology-based diagnosis of GC before surgery. Among these patients, samples from 16 chemoresistant patients were subjected to WES and ddPCR. Eight of these samples came from patients whose disease had progressed following adjuvant chemotherapy. The remaining eight samples were from patients clinically diagnosed with advanced GC, as indicated by CT staging, who had received neoadjuvant chemotherapy prior to surgery, but still showed disease progression. In these cases, samples were collected either through subsequent pathological biopsy or palliative surgery.

**Fig. 7 | RMY-186 as a promising therapy for overcoming chemotherapy resistance. a** Protein expression of CAD, cleaved PARP, P53 and p-H2A.X was determined after the MKN45 CAD$^{D1371A/D1371A}$ cells were treated with DMSO, RMY-186 (10 μM), 5-FU (50 μM), or the combination of RMY-186 and 5-FU for 24 h. **b** p-H2A.X was determined by Immunofluorescence as described in (**a**). Scale bar, 20 μm. **c**, **d** Microscopic images of pyroptotic cells. Arrowheads indicate ballooned cell membrane characteristic of pyroptotic cells. Scale bar, 25 μm. **e** Western blot was performed to detect the expression levels of GSDME in MKN45 CAD$^{D1371A/D1371A}$ cells with treatment as described in (**a**). **f** Western blot was performed to detect the expression levels of GSDME in CAD-KD MKN45 cells with 5-FU treatment. **g** PI (50 μg/mL) staining of pyroptotic cells in (**a**). Scale bar, 25 μm. **h** Culture supernatants in (**a**) were collected to measure the percentage of LDH release. $n = 3$ biologically independent samples per group. **i** PI (50 μg/mL) staining of pyroptotic cells in (**f**). Scale bar, 25 μm. **j** Culture supernatants in (**f**) were collected to measure

the percentage of LDH release. $n = 3$ biologically independent samples per group. **k**, **l** Whole-mount images (**k**) and the statistical analysis (**l**) of stomach from untreated, RMY-186-treated, 5-FU-treated or the combination of RMY-186 and 5-FU treated Cldn18-ATK Cad$^{D1371A/D1371A}$ mice. Tumors were marked with black dotted lines. $n = 8$ biologically independent samples per group. **m** Protein expression of CAD, cleaved PARP, GSDME and P53 were determined in tumor samples. **n** Representative images of H&E and IHC staining of stomach sections. Scale bar, 50 μm. **o** Quantification and statistical analysis of Ki67 staining images in (**n**). $n = 8$ biologically independent samples per group. **p** ELISA analyses of the concentrations of LDH in serum. $n = 8$ biologically independent samples per group. Data in (**a**, **e**, **f**, **m**) derive from the same biological experiment but were processed on parallel gels (see Source Data for gel-specific details). Data were expressed as means ± SD, two-tailed Student's $t$-test. ns not significant, $^*p < 0.05$, $^{***}p < 0.001$.

## Mouse models

Cldn18-CreERT2; Apc$^{fl/fl}$; Trp53$^{fl/fl}$; Kras$^{G12D}$ (Cldn18-ATK) mice were a generous gift from Dr. Yoshiaki Ito (National University of Singapore), and the mice developed obvious hyperplastic gastric tumors after Tamoxifen induction[25]. caspase-3 KO mice were a generous gift from Dr. Jianfeng Wu (Xiamen University). Cad$^{D1371A/D1371A}$ mice were constructed by the Xiamen University Laboratory Animal Center. Eight-week-old Cldn18-ATK mice, Cldn18-ATK caspase-3 KO mice and Cldn18-ATK Cad$^{D1371A/D1371A}$ mice were induced with a single intraperitoneal injection of 100 mg/kg tamoxifen (#T5648, Sigma) dissolved in corn oil (#HY-Y1888, MedChemExpress). Two weeks after Tamoxifen induction, 5-FU (#F6627, Sigma) was injected intraperitoneally at a dosage of 25 mg/kg once weekly for a month.

For RMY-186 treatment, eight-week-old C57BL/6 mice were used to evaluate the acute toxicity (LD$_{50}$) of RMY-186 in the living body. 100, 250, 500, 1000, and 2500 mg/kg of RMY-186 were dissolved in normal saline containing 5% DMSO and 1% carboxymethylcellulose. These mice were randomly divided into six groups and received vehicle control or different concentrations of RMY-186 (six mice in each group). The test substance was given through intraperitoneal injection, and mortality over 24 h was observed. RMY-186 at a concentration of 100 mg/kg is administered to Cldn18-ATK Cad$^{D1371A/D1371A}$ mice, whether or not it is combined with 5-FU.

To evaluate the therapeutic effects of combination treatment with RMY-186 and anti-PD-1 antibodies under chemotherapy. Cldn18-ATK mice were randomly assigned to four groups (six mice in each group) after Tamoxifen induction. Mice were administrated with 5-FU and RMY-186 in the presence or absence of IgG2a (#BE0085, Bio X Cell, AB_1107771) or anti-PD-1 antibodies(CD279) (#BE0146, Bio X Cell, AB_10949053). Anti-PD-1 antibodies (200 μg/mouse) were injected intraperitoneally twice a week for a month.

At the end of the experiment, all the mice were euthanized and several internal organs and tissues were taken for histopathological examination. The specimens were cautiously collected and washed with ice-cold 1× PBS; then homogenized in 2× SDS loading buffer using a homogenizer. Samples obtained were boiled for 15 min prior to western blotting.

## Western blotting

Protein samples were separated using SDS-PAGE gels prepared with the One-Step PAGE Gel Fast Preparation Kit (#E302-E305, Vazyme) and electrophoresed at 100 V for 2.5 h. Transfer to PVDF membranes (#IPVH00010, Merck Millipore) was followed by blocking with Protein Free Rapid Blocking Buffer (#PS108P, Epizyme; 30 min, room temperature). Primary antibodies were diluted in Universal Antibody Diluent (#WB500D, NCM Biotech). Membranes were incubated with antibodies overnight (4 °C), then with secondary antibodies for 1 h (room temperature). The complete information for all primary and secondary antibodies, including commercial sources, catalog numbers and dilution ratios, is detailed in the "Antibodies" section. Signals were

developed with Clarity Western ECL Substrate (#1705061, Bio-Rad) and imaged using an Azure C300 system (Azure Biosystems).

## RNA extraction and RT-PCR

Total RNA was isolated from cells by TRIzol reagent (#15596026, Invitrogen Life Technologies). The reverse transcription was conducted using the PrimeScript RT Master Mix following the manufacturer's instructions (#RR036A, TakaRa). RT-PCR was carried out on a 7500 Real-time PCR system (Applied Biosystems) using SYBR Premix Ex Taq reagents (#RR420A, TakaRa). Human β-actin was used as an internal control. The following primer sequences were used for RT-PCR:

 CAD-Forward: 5'-GAAGAATATCCTGCTGACCATTGG-3'
 CAD-Reverse: 5'-GAGACTGGCATAGAGGCTGTAG-3'
 β-actin-Forward: 5'-CATGTACGTTGCTATCCAGGC-3'
 β-actin-Reverse: 5'-CTCCTTAATGTCACGCACGAT-3'.

## Ionizing radiation

X-ray irradiation of cells were carried out in a RS-2000Pro Biological Irradiator located in Xiamen University Laboratory Animal Center. It was purchased from Rad Source Corporation (Atlanta, GA). The dose rate for the machine is 0.1–16 Gy/min.

## mRNA stability assays

CAD mRNA stability was determined using actinomycin D (ActD) mRNA stability assays in HGC27, MKN45, HCT116, and SW480 cells were treated with 5-FU (50 μM). CAD mRNAs were quantified relative to 18S rRNA mRNA, at various times after addition of Actinomycin D (ActD) (6.5 μg/mL) to the cell culture medium. mRNA degradation was analyzed by quantitative RT-PCR as described above. The 18S rRNA primer sequences were used for RT-PCR:

 18S rRNA-Forward: 5'-GTAACCCGTTGAACCCCATT-3'
 18S rRNA-Reverse: 5'-CCATCCAATCGGTAGTAGCG-3'.

The relative amount of CAD mRNA at each time point was determined in three independent experiments by cross-normalization using the internal controls and setting the input amount of CAD mRNA to 100%.

## Antibodies

Antibodies against the following proteins were purchased from Cell Signaling Technology (Beverly, Massachusetts, USA): CAD (#11933), cleaved PARP (Asp214) (#9541), caspase-3 (#9662), caspase-6 (#9762), caspase-7 (#9492), LC3 I/II (#4108), Phospho-Histone H2A.X (Ser139) (#9718), FLAG (#8146), HA (#3724), β-actin (#3700), and DAPI (#4083). Antibodies against DHODH (#sc-166348), UMPS (#sc-398086), p53 (#sc-126) and Ubiquitin (#sc-8017) were obtained from Santa Cruz Biotechnology (Dallas, Texas, USA). The antibody against CAD (#A6849) for immunohistochemistry was purchased from ABclonal (Wuhan, China). The antibody against GSDME (#ab215191) was obtained from Abcam (Cambridge, UK). Antibodies against GOT1

(#14886-1-AP) and VDAC3 (#55260-1-AP) were obtained from Proteintech (Rosemont, USA). The dilution ratio for all of these antibodies was 1:1000. Secondary antibodies used in this study were Goat Anti-Rabbit IgG H&L (#ab6721), Goat Anti-Mouse IgG H&L (#ab205719), and Rat Anti-mouse IgG for IP (HRP) (#ab131368, optimized secondary antibodies that do not detect denatured heavy and/or light chains during Western blot), which were obtained from Abcam. Secondary antibodies were used at a 1:5000 dilution.

Custom rabbit anti-CAD-1-1371 (antigenic determinant: aa 358–KEATAGNPGGQTVR-371) and anti-CAD-ΔN1371 (antigenic determinant: aa 1831–TTTPERPRRGIPG-1843) polyclonal antibodies were custom synthesized against corresponding short peptides by GenScript (GenScript Biotech, Piscataway, NJ, USA)[74].

## Reagents
The chemotherapy drugs used were 5-FU (#F6627, Sigma), Oxaliplatin (#HY-17371, MedChemExpress), Doxorubicin (#HY-15142A, MedChemExpress) and Paclitaxel (#T7191, Sigma). Z-VAD-FMK (#550377), Z-DEVD-FMK (#550378), Z-IETD-FMK (#550411), and Z-LEHD-FMK (#550381) were purchased from BD Biosciences Pharmingen (San Diego, CA, USA). Z-VEID-FMK (#1146-1) was ordered from Biovision (Palo Alto, CA, USA) and caspase-3/7 Inhibitor (Ac-DNLD-CHO, #HY-136733) was from MedChemExpress. Cell culture media also contained the following additives: Uridine (#U3750, Sigma), deoxythymidine (#D1413, Sigma), deoxycytidine (#D3897, Sigma), dihydroorotate (#D7128, Sigma) and Actinomycin D (#SBR00013, Sigma). MG-132 (#HY-13259), Chloroquine (CQ, #HY-17589A), Cycloheximide (CHX, #HY-12320) and Propidium Iodide (PI, #HY-D0815) were obtained from MedChemExpress (New Jersey, USA).

## Immunohistochemistry (IHC)
Five-μm-thick serial sections of the tumor were prepared from 10% FFPE tissue blocks for IHC. In brief, the sections were rinsed in distilled water for 10 min for all the slides and treated with freshly prepared 3% hydrogen peroxide for 15 min to get rid of the endogenous peroxidase activity. Tumor sections were incubated with rabbit monoclonal CAD (#A6849, ABclonal), cleaved PARP1 (#ab32064, Abcam), mouse monoclonal p53 (#sc-126, Santa), and rabbit monoclonal Ki67 (#ab245113, Abcam) antibodies in a humidified chamber at 4 °C overnight, after which the section was washed and then successively incubated with biotin-labeled anti-mouse or anti-rabbit IgG and peroxidase-conjugated streptavidin. The dilution ratio for all of these antibodies was 1:250. Both positive controls and negative controls (without the primary antibody) were also performed in each run of the experiment.

## Plasmid construction
The cDNA encoding human CAD (GenBank accession number: D78586) was cloned into pCMV-C-3xFlag (Public Protein/Plasmid Library, China) as described before[75]. All the CAD point mutants were generated using the QuikChange site-directed mutagenesis kit (Stratagene, La Jolla, California, USA) and subcloned into pCMV-C-3xFlag using the same enzyme sites (EcoR I/Xho I) as wildtype CAD. The cDNA fragment coding for deletion mutant CAD-1-1371 and CAD-ΔN1371 was digested with EcoR I/Xho I and subsequently subcloned into EcoR I/Sal I sites of pCMV-C-3xFlag to generate pCMV-C-3xFlag-CAD-1-1371 and pCMV-C-3xFlag-CAD-ΔN1371. All the CAD deletion mutants were subcloned into pCMV-C-3xFlag using the same enzyme sites as CAD-ΔN1371. CAD was subcloned into Not I/Bgl II sites of pEGFP-N1-His expression vector (Public Protein/Plasmid Library) to generate GFP-CAD. The cDNA encoding human PARP1 (M18112), RIPK1 (U50062), HDAC3 (U66914), BID (AF042083) and caspase-3 (U13737) were subcloned into pCMV-C-3xFlag (Public Protein/Plasmid Library). The cDNA encoding human caspases was cloned into pET-22b. All the primers used in this study and construction details are available on request.

## Lentivirus production and generation of CRISPR/Cas9 knockout cell lines
Lentiviruses were produced by transfecting HEK-293T packaging cells with lentiviral backbone constructs, the packaging plasmid psPAX2 (#PE081, Public Protein/Plasmid Library) and the envelope plasmid pMD2.G (#PE082, Public Protein/Plasmid Library) using the calcium phosphate transfection method. Consequently, following 10 h of transfection, the medium was aspirated, and then it was replaced with fresh warm medium supplemented with non-essential amino acid (NEAA). Lentiviral supernatants were harvested from the culture media of HEK-293T cells 48 h post-transfection. Target cells were transduced with lentiviruses, and 10 μg/mL polybrene (#H9268, Sigma) was added. After 24 h, the cells were treated with puromycin (#ant-pr-1, InvivoGen) for 2–3 days. The efficiencies of generating knockout were established via Western blotting. Two independent puromycin-resistant cell pools from each transduction with elevated knockout efficiencies (more than 80%) were preserved for later tests.

## Immunoprecipitation (IP) assay
HEK-293T cells were grown for 36 h after transient transfection with plasmid and collected by lysis in ice-cold immunoprecipitation buffer (20 mM Tris-HCl, pH 7.4, 150 mM NaCl, 1 mM EDTA, 1% Triton X-100) containing protease inhibitor cocktail tablets (#04693132001, Roche). Anti-FLAG M2 magnetic beads (#M8823, Sigma) were used for the cell lysates incubation at 4 °C overnight. The immunocomplex was analyzed for its virtue of immunoblotting with the help of the specific primary antibody (Anti-FLAG, #8146, CST) as well as the corresponding secondary antibodies (Rat Anti-mouse IgG for IP (HRP), #ab131368, Abcam, optimized secondary antibodies that do not detect denatured heavy and/or light chains during Western blot).

## Recombinant caspases production and in vitro cleavage reaction
Recombinant caspases production was performed according to the procedures described by Roschitzki-Voser et al.[27]. Human CAD proteins were immunopurified from HEK-293T cells as described above. The immunoprecipitates were washed five times with ice-cold lysis buffer, and the beads were incubated with either active human recombinant caspase (0.1 μg) or PBS in a reaction solution (50 mM HEPEs, 50 mM NaCl, 5% glycerol, 0.1% CHAPS, 10 mM EDTA and 10 mM DL-Dithiothreitol (DTT), pH 8.0) at 37 °C for 2 h. Special attention should be paid to the fact that DTT needs to be prepared and used now due to its special physical and chemical properties. The beads were re-sedimented, washed and boiled in 2× SDS loading buffer. The specimens were separated on an SDS-PAGE gel and subjected to western blotting as described above.

## Protein mass spectrometry
To detect potential sites where CAD may be cleaved by caspase-3, after electrophoresis of the in vitro cleavage reaction samples, the gels were cut down according to the lanes containing different samples (without or with recombinant caspase-3), and all molecular weights of the gels were included. Then the gels were analyzed using the timsTOF™ Pro Mass Spectrometer (Bruker, Germany). TimsTOF™ Pro uses Trapped ion mobility spectrometry (TIMS) technology and parallel accumulation serial fragmentation (PASEF) to achieve shotgun proteomic analysis. It has single peptide and protein identification performance, which can obtain accurate protein sequences and assist in determining the amino acid cleavage position of unstable proteins.

## Metabolite quantification via GC-MS
The change in the level of pyrimidine nucleotide-related metabolites was assessed under chemotherapy by the GC-MS (Agilent 5977B, US). In brief, for the measurement of the levels of metabolites in GC cells,

the cells were first plated in 6-well plates and then, after cell treatment, washed with ice-cold saline and lysed immediately in liquid nitrogen. For the extraction of the metabolites, 350 μL of methanol and water (70:30; v/v) kept at −20 °C was added to each well, and the GC adherent cells in each well were trypsinized and resuspended in the methanol-water mixture, which was then transferred to Eppendorf tubes. Chloroform with a temperature of −20 °C and 150 μL was added; after that, the sample pipettes were mixed at 4 °C for 20 min, followed by centrifugation at 17,000×g for 10 min at 4 °C. The upper phase with polar metabolites in methanol was then evaporated with a vacuum concentrator (Labconco Corporation, US) at 4 °C for 12 h and stored at −80 °C until the test or GC-MS was carried out.

Before applying to GC-MS, specimens were further centrifuged for 10 min at 15,000×g, followed by the transfer of 60 μL of the supernatant to the respective injection vial. GC was done employing a Hypersil-HP-5MS capillary column (30 m × 0.25 mm i.d., 0.25 μm film thickness) interfaced to a GC/MSD apparatus (7890B GC system, 5977B inert XL MSD, Agilent). The specified injector temperature was 260 °C. The column oven temperature was set initially at 70 °C and maintained for 2 min, then ramped to 180 °C at a rate of 7 °C/min and to 250 °C at a rate of 5 °C/min, then to 310 °C at a rate of 25 °C and kept at this temperature for 15 min, while the MSD transfer temperature was maintained at 280 °C[76]. The ion source temperature was set at 150 °C, and the temperature of the metastable supplementary quadrupole was set at 230 °C. All chemometric data were processed employing the MassHunter software version B.07.01 SP1, which was developed by the Agilent company. The Metran software tool was used to perform the peak integration. For relative metabolite abundance in the specimens, relative intensity and ion counts were expressed in terms of the total ion count of the sample.

## LC-MS/MS analysis
Ubiquitinated Flag-CAD samples were prepared as described in the section describing the immunoprecipitation (IP) assay. The Ubiquitinated Flag-CAD protein was subjected to SDS-PAGE and stained with Coomassie blue. The specific band was excised and digested with chymotrypsin before being analyzed by mass spectrometry. The peptides were subjected to NSI source followed by tandem mass spectrometry (MS) in Q Exactive™ Plus (Thermo) coupled online to the UPLC.

## Immunofluorescence analysis
Cultures were prepared from each group, and these were cultured on coverslips in 6-well plate culture dishes. Afterward, the cells were immersed in 4% paraformaldehyde (PFA) for 15 min and permeabilized with 0.5% Triton X-100 for 15 min. Slices were then blocked employing 5% bovine serum albumin (BSA) in PBS at room temperature for 1 h before incubation with the primary antibodies against DAPI (#4083, CST) and Phospho-Histone H2A.X (Ser139) (#9718, CST) at 4 °C overnight.

After dewaxing and receiving antigen repairing, tissue sections were incubated with primary antibodies against CD3 (#ab16669, Abcam) or CD8 alpha (#ab 217344, Abcam) overnight at 4 °C. Then they were incubated with fluorescent secondary antibodies appropriately respond to primary antibody in species for 1 h in dark condition and DAPI at room temperature for 10 min. The images were acquired using a fluorescence microscope (Nikon Eclipse Ti) and digital image analysis software (NIS-Elements, Nikon).

## Cell cycle analysis
Cell cycle distribution was documented employing a flow cytometer (Fortessa X20; Becton, Dickinson and Company; US) with a Cell Cycle and Apoptosis Analysis Kit (#40301ES; Yeasen) according to the instructions of the manufacturer. The cells were then trypsinized with trypsin-EDTA and then fixed employing 70% alcohol for 24 h at 4 °C. Thereafter, the cells were washed once or twice with 1× PBS, freshly prepared, and then resuspended in the staining solution and then combined with the PI and RNase A solution mixture. The specimens were placed at 37 °C in the darkness with a duration of 30 min, and then flow cytometry was performed.

## Transfection of siRNA
siRNAs targeting CAD (siRNA-1: GCACCAAGCAGGAGGAAUUTT, siRNA-2: UGGCCAAGCUGGAGAAUUUTT), and negative control (siScramble: UUCUCCGAACGUGUCACGUTT) were synthesized by Public Protein/Plasmid Library (Nanjing, China). The transfection was carried out using Lipofectamine 3000 Reagent (#L3000015, Invitrogen) according to the manufacturer's protocol.

## LDH release assay
CytoTox 96(R) Non-Radioactive Cytotoxicity Assay Kit (#G1780; Promega) was used to measure LDH released from pyroptosis cells at 450 nm following the manufacturer's guide. Released LDH in culture supernatants is measured with 30 min coupled enzymatic assay, which results in conversion of a tetrazolium salt (INT) into a red formazan product. Quantity of color developed is directly related to the extent of sample lysation without any complete cell membranes. Visible wavelength absorbance data were recorded by a standard 96-well plate reader (Thermo Fisher).

## Cell-counting kit 8 (CCK-8) cell viability assay
Cells were seeded in a 96-well cell culture plate at 1000 cells per well and incubated in a normal culture medium for 24 h to allow them to adhere to the plate. Next, the cells were treated with the indicated drugs based on the experimental design of the study. Cell viability was assessed employing the CCK-8 kit (#K1018; ApexBio Technology) based on the producer's user manual. Absorbance at 450 nm was determined in a microplate reader (Multiskan TM, Thermo Fisher).

## RNA sequencing and data analysis
Many GC cells were transfected to 6 cm plates, then treated with DMSO or 5-FU for 48 h, and the total RNA extraction was then performed. The following experiments were carried out as mentioned before in our previous study[77]. For instance, GSNAP (version 2013-11-10) was utilized to align the raw FASTQ reads to the mouse reference genome (UCSC: GRCm38/mm10). Specificity in this case involved filtering of the reads that would only contain the unique reads that mapped on the reference sequence. For statistical analysis, the R language was used with limma for differential expression analysis. Pathway enrichment was performed employing the online edition of GSEA employing the 'Reactome gene sets'.

## Homozygously wildtype CAD (WT/WT) or CAD-D1371A (D1371A/D1371A) knock-in cell lines
CRISPR-Cas9-mediated ablation of the CAD gene was achieved with CRISPR-Cas9 RNP and OligO (provided by Haixing Bioscience, Suzhou) containing expression cassettes for hSpCas9 and chimeric guide RNA. Two single-guide RNAs (sgRNAs: 5′-GAAGACGTATGACCGATTACCAT-3′ and 5′-TGGTACGTCTACGAGTTCAATAC-3′) targeting exon 26 of CAD were designed using the CRISPR Design Tool (http://crispr.mit.edu). Ribonucleoprotein (RNP) complexes containing Cas9 protein and synthesized sgRNAs, along with donor oligonucleotides, were delivered into cells via electroporation using the Neon™ Transfection System (#MPK10096; Thermo Fisher Scientific) following the manufacturer's protocol[78]. At 72 h post-transfection, single-cell clones were isolated through limiting dilution in 96-well plates. Initial screening for CAD knockout efficiency was performed through endpoint PCR. For mutation validation, genomic DNA extracted with the Quick-DNA Miniprep Kit (#D3025; Zymo Research) underwent PCR amplification

using Taq DNA polymerase (2 U, New England Biolabs), dNTPs (100 μM) and Target-flanking primers (250 nM):

Forward: 5'-TGTGCAGTGAGTGCACCACT-3'

Reverse: 5'-GTGCACTTGATATCGATGAT-3'.

Sanger sequencing of the amplified region confirmed biallelic frameshift mutations in selected clones, which were subsequently expanded under identical culture conditions to parental cell lines.

## Purification of genomic DNA from FFPE tissues

The QIAamp(R) DNA FFPE Tissue Kit (#56404; QIAGEN, Hilden, Germany) was used to extract DNA from FFPE tissues from patients who derived no clinical benefit from neoadjuvant or adjuvant chemotherapy.

## Digital droplet PCR (ddPCR)

Identification of the mutation of interest in CAD (reference sequence NM_004341.5) in ctDNA was performed employing the Naica crystal digital PCR system obtained from Stilla Technologies, France. Primers and probes for the detection in exon 26 are as follows:

CAD-GAT-Forward: 5'-TTGACTCCGGGTTGGCA-3'

CAD-GAT-Reverse: 5'-CAGCTAGCTGCTCCAGGATG-3'

CAD-GAT-GAT-Probe: 5'-VIC-TCACCATCCACAGCC-MGB-3'

CAD-GAT-TAT-Probe: 5'-FAM-TCACCATACACAGCCT-MGB-3'

CAD-GAT-GAA-Probe: 5'-FAM-TCACCTTCCACAGCC-MGB-3'.

These digital PCR reactions were set up employing PerFecTa Multiplex qPCR ToughMix reagents, which were obtained from Quanta Biosciences, Gaithersburg, MD, US; 40 nM FITC obtained from Saint Louis, MO, US; 1 μL of the primer and probe multiplex mix; and 3 μL of the DNA template. The total volume of the PCR reaction is 20 μL. Cycling conditions were 95 °C for 10 min, followed by 45 cycles of 95 °C for 10 s and 62 °C for 15 s. After the sample plate is successfully set on the computer, it is put into the QX200 Droplet Microdroplet Analyzer for droplet analysis and detection. The data can be uploaded to the computer for final analysis. After the reaction procedure is completed, adjust the threshold line to the appropriate position according to the specific reaction situation, so as to carry out the next step of result interpretation. For the negative controls, normal wild-type DNA was used, while for the positive controls, mutated DNA was employed.

## Whole-exome sequencing (WES)

Genomic DNA isolated from clinical specimens was fragmented to a peak size of 230 ± 50 bp using a Covaris LE220R-plus system (duty cycle: 10%, intensity: 5.0) under optimized acoustic shearing conditions. DNA fragments underwent enzymatic processing with T4 DNA polymerase (Thermo Fisher Scientific) and Klenow fragment (NEB) at 20 °C for 30 min to generate blunt ends, followed by 3'-adenylation using Klenow enzyme (65 °C, 30 min). Illumina-compatible duplex adapters (Integrated DNA Technologies, 15 μM) containing unique dual indexes (UDI) were ligated to facilitate multiplexed sequencing of 384 samples per run. Libraries were size-selected via AMPure XP beads (Beckman Coulter; 0.9 × ratio) to enrich 300–500 bp fragments and amplified with KAPA HiFi HotStart polymerase (Roche) for 8 cycles to minimize PCR duplicates.

Target enrichment was performed using Agilent SureSelect Human All Exon v8 probes (v8.1.0 design, 35.13 Mb coverage) through 24 h hybridization at 65 °C with Cot-1 DNA (Invitrogen) blocking. Captured DNA was recovered using streptavidin-coated Dynabeads M-270 (Thermo Fisher) and further amplified for 12 cycles to incorporate sample-specific indices. Final libraries were quantified via Qubit 4.0 fluorometry (Thermo Fisher) and assessed for fragment distribution on an Agilent 5400 Fragment Analyzer (DV200 > 85%). Sequencing was conducted on an Illumina NovaSeq 6000 platform (PE150) at Novogene (Beijing), achieving a median coverage depth of 100 × across 95.4% of the target regions. Raw data were processed through a modified GATK4 pipeline incorporating unique molecular identifiers (UMIs) for duplicate removal and variant calling[79].

## Synthesis of RMY-186

The synthesis route of compound RMY-186 (2-(4-(3-bromo-benzyl)-piperazin-1-yl)-3-hydroxy-4H-chromen-4-one) was shown above; the Chemdraw files for all compound structures have been provided as supplementary material. Reagents and conditions: (a) dry DMF, 80 °C, 4 h, 60% yield; (b) CH₂Cl₂, HCl, reflux, 1 h, 89% yield; (c) H₂O₂, NaOH, CHCl₃, −5 °C, 8 h, 80% yield; (d) HCl, CHCl₃, 70 °C, 1 h, 32% yield; (e) NBS, vazo, MeCN, 85 °C, 8 h, 34% yield; (f) piperazine, CHCl₃, 80 °C, 3 h, 82% yield; (g) CH₂Cl₂, Et₃N, C₇H₆Br₂, r.t., 12 h, 19.7% yield.

All reagents were purchased and used without further purification unless otherwise noted. Reactions were monitored by thin-layer chromatography (TLC) on YanTai silica gel GF-254 thin-layer plates. ¹H NMR and ¹³C NMR spectra were determined with a Bruker Avance III 600 MHz spectrometer (600 MHz for ¹H and 150 MHz for ¹³C). Chemical shifts are expressed in $\delta$ values (ppm), using tetramethylsilane (TMS) as the internal standard; coupling constants ($J$) are given in Hz. Signal multiplicities are characterized as s (singlet), d (doublet), t (triplet), m (multiplet). High-Resolution Mass Spectrometry (HRMS) was acquired on a Q-Exactive series MS instrument with UV detection at 254 nm in low-resonance electrospray mode (ESI)[80]. The melting point was determined using an INESA WRS-3A micromelting point tester. The infrared spectrum was determined using a Bruker Alpha infrared spectrometer.

### (E)-3-(Dimethylamino)-1-(2-hydroxy-phenyl) prop-2-en-1-one (3)

To a solution of 1-(2-hydroxy-phenyl) ethan-1-one (1) (200 mg, 1.47 mmol) and 1,1-dimethoxy-N, N-dimethylmethanamine (2) (210 mg, 1.76 mmol) in anhydrous N, N-dimethylformamide (DMF) (8 mL) was added, heated to 80 °C and stirred thoroughly for 4 h. After confirming the completion of the reaction with TLC, the resulting solution was extracted with ethyl acetate, washed with water and brine, dried over Na₂SO₄ and concentrated in vacuo to give 169 mg (60%) of the crude (E)-3-(dimethylamino)-1-(2-hydroxy-phenyl) prop-2-en-1-one (3), which was directly used for the next step.

### 4H-Chromen-4-one (4)

To a solution of intermediate 3 (100 mg, 0.52 mmol) in CH₂Cl₂ (8 mL) was added concentrated hydrochloric acid (1 mL) dropwise and refluxed for 1 h. After the reaction, a large amount of water was added, extracted with dichloromethane and washed with saturated brine, concentrated to obtain a red solid (intermediate 4), which was directly used next without purification (yield 89%). ¹H NMR (600 MHz, CDCl3): $\delta$ 8.22 (dd, $J$ = 1.4, 7.9 Hz, 1H), 7.86 (d, $J$ = 6.0 Hz, 1H), 7.71–7.64 (m, 1H), 7.46 (d, $J$ = 8.4 Hz, 1H), 7.41 (t, $J$ = 7.6 Hz, 1H), 6.35 (d, $J$ = 6.0 Hz, 1H).

### 3-Hydroxy-4H-chromen-4-one (6)

To a solution of intermediate 4 (975 mg, 6.67 mmol) was dissolved in CHCl₃ (20 mL), NaOH (50 mmol) was added to stir at −5 °C for 5 min, H₂O₂ solution (20%, 10 mL) was added, and the reaction was monitored by TLC to the end after stirring at 0 °C. The organic phase was rapidly separated with a separatory funnel, washed with saturated brine, and concentrated to obtain a red solid (intermediate 5). Concentrated hydrochloric acid (1 mL) was dropped into intermediate 5 to

completely extract it into the liquid, heated up to 70 °C and stirred for 1 h. A yellowish solid precipitated upon the addition of a large volume of water. Filtered to get intermediate **6** (yield 32%). [1]H NMR (600 MHz, CDCl$_3$): $\delta$ 8.27 (dd, $J$ = 1.6, 8.0 Hz, 1H), 8.02 (s, 1H), 7.68 (ddd, $J$ = 1.6, 7.0, 8.5 Hz, 1H), 7.52–7.48 (m, 1H), 7.44–7.39 (m, 1H), 6.38 (br. s., 1H).

### 2-Bromo-3-hydroxy-4H-chromen-4-one (7)
To a solution of intermediate **6** (340 mg, 2.10 mmol) was dissolved in MeCN (20 mL), $N$-bromosuccinimid (NBS) (550 mg, 3.09 mmol) and azobisisobutyronitrile (Vazo) (70 mg, 0.43 mmol) were weighed and added, protected from light, stirred at 85 °C for 8 h. Vazo was added every 2 h. After the reaction was complete, it was concentrated and purified by silica gel column chromatography to provide **7** (172 mg, 34%) as a white solid. [1]H NMR (600 MHz, CDCl$_3$): $\delta$ 8.28 (dd, $J$ = 1.6, 8.0 Hz, 1H), 8.24 (s, 1H), 7.72 (ddd, $J$ = 1.6, 7.0, 8.5 Hz, 1H), 7.51–7.45 (m, 2H).

### 3-Hydroxy-2-(piperazin-1-yl)-4H-chromen-4-one (8)
To a solution of piperazine (121 mg, 1.4 mmol) was dissolved in CHCl$_3$ (10 mL), heated to 80–90 °C, and intermediate **7** (241 mg, 1.0 mmol) dissolved in CHCl$_3$ was added dropwise with stirring and stirred by heating for 4 h. Concentrated to dryness to obtain yellowish brown solid crude intermediate **8** (202 mg, 82%), which was directly plunged into the next step without purification[81].

### 2-(4-(3-bromobenzyl)-piperazin-1-yl)-3-hydroxy-4H-chromen-4-one (9, RMY-186)
To a solution of intermediate **8** (246 mg, 1.0 mmol) in CH$_2$Cl$_2$ (10 mL) was added Et$_3$N (101 mg, 1.0 mmol) dropwise, and then 1-bromo-3-(bromomethyl)-benzene (500 mg, 2 mmol) was added while stirring at room temperature. After 12 h of stirring, TLC monitoring showed that intermediate **8** had been completely consumed. The mixture was washed with brine (3 × 30 mL), and then dried with Na$_2$SO$_4$, and concentrated in vacuo. The residue was purified by silica gel column chromatography (EtOAc/n-hexane = 1:10 → 1:1) to afford RMY-186 as a yellowish solid (81.8 mg, yield 19.7%). Mp = 162–163 °C. [1]H NMR (600 MHz, CDCl$_3$): $\delta$ 8.15 (dd, $J$ = 7.9, 1.1 Hz, 1 H), 7.49–7.56 (m, 2 H), 7.41 (d, $J$ = 7.9 Hz, 1 H), 7.30–7.36 (m, 2 H), 7.26–7.29 (m, 1 H), 7.18–7.22 (m, 1 H), 3.75–3.90 (m, 4 H), 3.53 (s, 2 H), 2.59 (t, $J$ = 4.9 Hz, 4 H); [13]C NMR (151 MHz, CDCl$_3$): $\delta$ 169.7, 151.9, 150.3, 140.2, 132.0, 131.2, 130.4, 129.9, 127.6, 125.0, 124.7, 124.4, 122.6, 121.1, 116.5, 62.3, 52.9, 46.2; IR (neat) cm$^{-1}$: 3203, 2921, 1603, 1533, 1427, 1274; HRMS (ESI) $m/z$ calcd for C$_{20}$H$_{19}$BrN$_2$O$_3$ [M + H]$^+$ 415.0652, found 415.0655.

### Microscale thermophoresis (MST)
HEK-293T cells expressing GFP-CAD-D1371A were used as a source of fluorescently labeled CAD-D1371A for the binding assay. Cell lysates were prepared using RIPA buffer (20 × 10$^6$ cells/mL). For binding studies, the lysates were diluted 150× with MST binding buffer (10 mM HEPES, pH 7.4; 1 mM MgCl$_2$; 20 mM NaCl; 0.01% NP-40; 10 mM Tris-HCl, pH 7.4; 30 mM NaCl; 2 mM MgCl$_2$; 0.05% Tween-20) to provide the optimal level of the fluorescent protein in the binding reaction[82]. pEGFP-N1-His expression vector transfected HEK-293T cells have been used to evaluate background fluorescence. Background fluorescence can be more significant and thus has to be monitored. Titration series consisting of 16 binding mixtures and lysate sample without the ligand have been prepared. Each sample contained 15 μL of diluted cell lysate and 15 μL of RMY-186 solutions of varying concentrations. The measurements were taken in standard treated capillaries on Monolith NT.115 (NanoTemper Technologies GmbH, Germany) instrument using 50% IR-laser power and LED excitation source with $\lambda$ = 470 nm at ambient temperature[83]. RMY-186 binding led to a change in the mobility of CAD-D1371A in the temperature gradient. Thermophoretic signal is expressed on the $y$-axis, with RMY-186 concentration on the $x$-axis. The average of three measurements is plotted as a single data point.

### Molecular docking
The CAD-D1371A protein structure was obtained from AlphaFold Protein Structure Database (https://alphafold.ebi.ac.uk/)[43]. Maestro 11.1 software was employed to dock. Protein was considered rigid, and the small molecules were considered flexible during the entire docking process. Commonly, molecular docking results are drawn in a two-dimensional format, and the resultant structures and interactions are visualized employing the program LigPlot.

### Statistics and reproducibility
All experiments were conducted through at least three independent experimental replicates to ensure biological reproducibility. Trends were similar in all the replicates. All attempts at replication were successful. All data are representative of three independent experiments.

No statistical methods were used to pre-determine sample sizes, but the sample size was chosen in advance based on the common practice of the described experiment in the literature and is specified for each experiment. For cell culture experiments, at least three replicates per group and for animal studies, at least eight animals per group were used unless specified otherwise, taking into account the variability within a cage and experimental group as well as between individual experimental repetitions to set appropriate sample numbers to allow for sound interpretation of experimental results. No data were excluded from our analyses. Mice were randomly divided into different groups. No randomization was performed for other experiments as control group and treatment group in these experiments were defined. Data collection and analysis were not performed blind to the conditions of the experiments. Investigators were not blinded in cells; investigators performing experiments analyses were blinded to patients and animal genotyping information. All results were depicted as figures in terms of mean ± SD, two-tailed Student's $t$-test. $p$-values < 0.05 were considered significant.

### Reporting summary
Further information on research design is available in the Nature Portfolio Reporting Summary linked to this article.

## Data availability
Raw sequencing data for all samples have been uploaded to the Sequence Read Archive (SRA) public database with accession number PRJNA1108914. RNA sequencing data are available under accession number SRX24491504. The small-molecule crystallographic data for RMY-186 have been submitted to the Cambridge Structural Database (CSD) with deposition number 2359201, and relevant information is deposited to PubChem (preview ID: 1553371761). The mass spectrometry proteomics data have been deposited in the ProteomeXchange Consortium via the iProX partner repository[84] with the dataset identifier PXD059365. Source data are provided with this paper.

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

## Acknowledgements

We thank Dr. Chensong Zhang (Xiamen University) and Dr. Qiao Wu (Xiamen University) for assistance with experimental design and manuscript writing. We thank Dr. Yoshiaki Ito (National University of Singapore) for kindly providing *Cldn18*-CreERT2; *Apc*fl/fl; *Trp53*fl/fl; Kras^G12D mice, and Dr. Jianfeng Wu (Xiamen University) for *caspase-3* KO mice, and Dr. Wenbin Hong (Xiamen University) for technical support of molecular docking. This work was financially supported by the National Natural Science Foundation of China (X.H., 82122046; J.Z., 82103160; Y.S., U22A20324), the National Science and Technology Major Project of the Ministry of Science and Technology of China (X.H., 2024ZD0525003), the Young Top Talents of Fujian Young Eagle Project (X.H.), the Fujian Health Youth Scientific Research Project (X.H., 2021ZQNZD018) and, the International Science and Technology Cooperation Project of Liaoning Province (Z.W., 2023JH2/10700005).

## Author contributions

X.H. and Z.W. conceived the project, generated hypotheses, and designed the studies. Y.S. provided clinical samples from GC patients. F.L., C.C. and Y.D. designed and synthesized the compounds. J.M. and J.Z. were the key contributors in designing and conducting most of the experiments. C.Z. provided the constructional suggestions and help for this project. J.M., A.C. and G.P. performed biological experiments and mouse experiments. J.T. carried out bioinformatics analyses. Z.L. identified the histological characteristics of GC model mice. M.Z., Y.Z. and Y.X. conducted metabolite detection and analysis. Z.N. collected all clinical samples and clinical characteristics. H.Z., S.Z., M.X. and W.Y. performed the experiments, provided reagents, and analyzed data. J.M. and C.Z. wrote the manuscript. X.H., Y.S. and Z.W. commented on and edited the manuscript.

## Competing interests

The authors declare no competing interests.
