## [Transparent Peer Review file · Nature Communications]

Cleavage of CAD by caspase-3 determines the cancer cell fate during chemotherapy

Corresponding Author: Professor Xuehui Hong

Version 0:

Reviewer comments:

Reviewer #1

(Remarks to the Author)

In this study, Ma et al. describe the role of the de novo pyrimidine synthesis pathway in determining the chemosensitivity of gastric and colon cancer cells. The study demonstrates that chemotherapeutic drugs can induce the degradation of CAD, a key enzyme in pyrimidine synthesis, leading to apoptosis in chemosensitive cells, and identifies the critical role of caspase-3-mediated cleavage of CAD at the Asp1371 residue, with mutations at this site causing chemoresistance. The authors further identify RMY-186 as a potential therapeutic agent to degrade mutant CAD and overcome chemoresistance. Overall, this study provides valuable insights into the molecular mechanisms underlying chemotherapy resistance and suggests a potential therapeutic strategy to improve treatment outcomes in cancer patients. The scope of the work is comprehensive, and this study could have an impact on the field of cancer therapy, particularly in understanding and addressing chemoresistance. The impact of the study can be improved by addressing the comments below.

Major comments:

- The authors did not provide any validation for caspase-3 KO mice. IHC/IF staining for caspase-3 expression in tissues from these mice should be included.
- Fig. 2t – Authors claim that the caspase-3 KO mice are not significantly affected by 5-FU treatment, but the Ki67 expression seems to be decreased in the 5-FU-treated sample. Authors should provide quantification for Ki67 and examine if the difference is significant. If there is a significant difference, authors should elaborate more on that result.
- Fig. 2t - How representative is the c-PARP image shown? Is the cell death completely attenuated in the caspase-3 KO samples? If yes, is the reason for the demonstrated slight reduction in tumor growth (Fig. 2r) due to a reduction in proliferation? How is the expression of cleaved caspase-7 affected in these mice?
- The focus of the study is CAD degradation induced by caspase-3 cleavage; however, the authors did not include any results showing their distribution/co-localization in vivo. It would be helpful to co-stain for cleaved caspase-3 and CAD using an IF assay to show their distribution in tissues and provide further evidence of the relationship between cleaved-caspase-3+ cells and CAD degradation. Is the expression of cleaved caspase-3 distributed differently in samples with and without CAD mutations? Co-IP assays could also show the interaction between the proteins.
- The authors established a link between CAD mutations at the Asp1371 site and chemoresistance. Have they explored whether the mutated form of CAD has any other role in cancer cells? Does it affect clinical outcomes such as overall survival or progression-free survival and how does the mutation frequency in chemoresistant patients compares to a larger cohort.
- The authors present data showing that RMY-186 promotes the degradation of CAD and its mutants through increased ubiquitination. Do they have any insight about the mechanism through which RMY-186 enhances ubiquitination and degradation (the authors mention the use of MG-132 to restore the protein levels of CAD but not the potential mechanism)? Additionally, is this effect specific to CAD, or could RMY-186 also impact other proteins? Expanding on the specificity of RMY-186 would help to ensure that potential therapeutic effects are not accompanied by unintended consequences, which would be important for potential future clinical application. Have the authors explored whether RMY-186 enhances the efficacy of other standard treatments in resistant cancer models?
- There is again no assay provided to show Caspase-3/CAD distribution in 5-FU/RMY-186 mouse samples.

Minor comments:

- The authors discuss multiple pathways in the introduction section, but the complexity may be challenging for readers to follow. Including a schematic illustration would greatly enhance clarity and understanding of the mechanisms involved.
- Authors should include the cell line information in all figure plots, not only in figure legends (Fig. 1h-l, Fig. 4a-d, etc.).
- Fig. 2 – Authors should improve the arrangement of panels; they are not formatted in a proper order logical to follow (j-k-p?). This applies to other figures as well, due to the large amount of data included, at times they are very difficult to follow.

Authors should consider adjusting them for clarity.

- Fig. 2a-b - It is unclear where the CQ and MG-132 treated samples are shown.
- Line 311 – Typo: “haft-life” instead of “half-life”

Reviewer #2

(Remarks to the Author)

In their manuscript, Ma and Zhao et al describe the caspase-3 mediated cleavage of the pyrimidine biosynthetic enzyme CAD during apoptosis. They demonstrate that caspase-3 is necessary and sufficient for CAD cleavage. They map the caspase-3 cleavage site to D1371 within the CPSase domain of CAD. Overexpression of CAD, especially a CAD mutant that cannot be cleaved by caspase 3, increases pyrimidine pools and survival upon 5-FU treatment, while blunting H2AX phosphorylation. Expression of the CAD D1371A mutant renders tumors somewhat resistant to 5-FU treatment. The authors identify tumors with the CAD D1371 mutation and demonstrate that cells expressing these variants are also relatively 5-FU resistant. Finally, the authors identify a small molecule that induces CAD proteasomal degradation. Combined inhibition of CAD and 5-FU hastens cell death, which is attributed to increased pyroptosis, and this combination further reduces tumor burden. Overall, the authors provide a thorough assessment of the impact of apoptosis on CAD and the consequent effect on 5-FU sensitivity.

The experiments are well-controlled and technically well-done. My main comments concern the conclusions that the authors have drawn, which at times tend to be over-interpretations of the data presented. In particular, the authors frequently conflate 5-FU with all chemotherapy, when the protective effect of the CAD D1371 mutant may be highly specific to pyrimidine analogs. These comments can largely be addressed by substantially modifying the text.

Major Points:

1. In the abstract the authors state “we found that the de novo pyrimidine synthesis pathway, which we previously identified as crucial for the proliferation of gastric and colon cancer cells, determines the chemosensitivity of these cancer types” and “conferred tumor chemoresistance”, “resulting in a significant improvement in the effectiveness of chemotherapy”, and refer to “chemotherapy” in several other places, including the body of the manuscript. These statements inappropriately imply that pyrimidine synthesis impacts chemosensitivity broadly, when the authors have only really focused on 5-FU sensitivity.
2. The authors interpret the results upon expression of CAD as evidence that CAD degradation is necessary for apoptosis induction. However, the authors do not report the degree of CAD protein expression in Figures 1, 4a-m, and 5g-h. Therefore, it is difficult to know whether CAD is being over-expressed and driving UMP/thymidine synthesis, diluting the effect of FdUMP. Moreover, the relative over-expression of CAD versus the CAD mutant may explain the data presented in Figure 4. Understanding the relative expression in these models is needed.
3. “we generated the site directed mutant MKN45 and HCT116 cells with homozygous CAD-D1371A integrated into the genome”. The authors need to provide genotyping data to support this statement, should indicate the total level of CAD protein in these engineered cells, and determine whether PARP and caspase-3 cleavage is affected upon 5FU treatment.

Minor Points:

1. It would be helpful for the interpretation of the data to understand whether another method of inducing apoptosis that does not involve a nucleotide or nucleotide analog also is impacted by CAD D1371A expression. Is this phenotype specific for 5-FU and other pyrimidine analogs, or is CAD cleavage truly hastening apoptosis when stimulated in other ways? While in my opinion this experiment is not needed for publication, it would substantially increase impact if CAD cleavage is required for apoptosis induction beyond nucleotide analogs (e.g. venetoclax).
2. Authors state that pyrimidine analogs “prevented the 5-FU-induced apoptosis”. It would be more accurate to say that they suppressed H2AX phosphorylation as this figure does not measure specific apoptosis markers.
3. Authors state, “we observed that caspase-9, which is specifically involved in the intrinsic apoptosis pathway, was responsible for the CAD cleavage”. This wording implies that caspase-9 is directly cleaving CAD, which I don’t think was the authors’ intent.
4. “a pile of peptides with the N-terminal residue of Asp1371 was hit”. Could the authors be more precise about what these data show?

Reviewer #3

(Remarks to the Author)

- What are the noteworthy results?
 - o The authors discovered a correlation between a reduction in CAD cleavage and resistance to 5-FU treatment in cells and in mouse models. The native cleavage process was found to be performed by caspase-3 within the apoptosis pathway. CAD mutates to D1371A in a small population of resistant cancer patients, preventing caspase-3 cleavage and maintaining full length CAD. A small molecule RMY-186 was found to increase apoptosis and pyroptosis in cancer cells and tumors.
- Will the work be of significance to the field and related fields? How does it compare to the established literature? If the work is not original, please provide relevant references.
 - o The work is definitely of interest to the field as chemoresistance is a large problem in cancer therapeutics. This work was very detailed in determining the mechanism of the pathway of chemoresistance and the mode of chemosensitization.

- Does the work support the conclusions and claims, or is additional evidence needed?
 - o Is there any experimental evidence to support the docking of RMY-186 into CAD-D1371A in cells? The correlation between RMY-186 and ubiquitination/degradation of CAD is clear, and the authors calculated a Kd for RMY-186 with CAD-D1371A-GFP but no in cell experiments were done to confirm this was the target. A Kd of 6.78 μ M is not terribly strong binding and the cellular environment would likely have competing enzymes.
 - o The link of RMY-186 to increasing patient response to immunotherapy is unclear. (line 549-551) What evidence specifically supports that conclusion? The authors show that RMY-186 causes high levels of ubiquitination on CAD, leading to its degradation. They also showed that RMY-186 there was an increase in GSDME cleavage and subsequent pyroptosis. More data would be helpful to provide a stronger link between RMY-186 and its potential to provide immunotherapy sensitivity.
- Are there any flaws in the data analysis, interpretation and conclusions? Do these prohibit publication or require revision?
 - o Many conclusions are drawn from an extremely large number of experiments. There is very little explanation as to how they got to those conclusions. Some sections did have enough explanations (i.e. Figure 2 had more text regarding the rationale and conclusions drawn than many other figures)
 - o The article is written like a communication with minimal explanations yet is already as long as a full research article. It almost feels like this paper is too big with the amount of figures and subsections of figures. If they authors plan to just state a conclusion that is made with data in Figure X parts h-k with no other substantiation, those figures may be better suited for supplemental information. If there are no page limitations, I suggest the authors expand more on their data analysis and interpretation.
- Is the methodology sound? Does the work meet the expected standards in your field?
 - o Yes to both.
- Is there enough detail provided in the methods for the work to be reproduced?
 - o Specifics of fluorescence microscopy are not given in much detail. The methods describe the imaging itself, but does not define clearly the different variables and the conditions used. The figure captions list the conditions using abbreviations (i.e. EV, CAD, dT/dC) but the figures have more conditions in the results (i.e. OE-CAD, DMSO, Vector). It is unclear if vector is the same as empty vector. Sometimes it appears to be and other times it appears to be the transfected group. These comments apply to Figure 1 specifically, but more generally to all of the figures. More details should be presented either in the figure captions or in the methods section.
 - o Figure 1 (d,e) and (f,g) have their labels swapped in the figure caption. Parts d and e are immunofluorescence and f and g are immunoblots.
 - o Many abbreviations are not defined such as OE-CAD. It would be useful to have an extensive abbreviations section where all are defined.
- General comments:
 - o The abstract is much longer than the 200 words stated by the journal
 - o There are some typographical errors throughout the manuscript

Reviewer #4

(Remarks to the Author)

Prior observations on the extensive heterogeneity within gastric (GC) and colorectal (CRC) cancers attributed the heterogeneity to dysregulation of metabolic reprogramming as key to resistance to chemotherapy-mediated DNA damage . Ma. J. et al, in their quest to further the mechanism of resistance in GC and CRC and following a continuum in their work which identified UHMK1, Dyrk3, and FOXK2 as modulators of nucleotide synthesis with direct influence on de novo reprogramming of nucleotide synthesis, has led them to CAD. CAD is the first rate-limiting enzyme in the de novo pyrimidine synthesis pathway, and also a caspase-3 substrate. In prokaryotes de novo pyrimidine synthesis is composed of several sequential steps each of which is encoded by a separate enzyme. However, in mammalian system, four enzymes in the pathway, namely glutaminase (GLN), glutaminase-dependent carbamoyl phosphate synthase (CPS-2), aspartate transcarbamoylase (ATC), and dihydroorotase (DHO) are fused into one single protein, called CAD.

In a novel aspect of current work, the authors have found that activated caspase-3 cleaves CAD at Asp-1371. This results in decreased pyrimidine synthesis and the ensuing imbalance in nucleotide pool increases DNA damage which triggers apoptosis. Tumors with elevated CAD expression can resist apoptosis by increasing pyrimidine synthesis. Further, mutations at caspase-3 cleavage site emerge resistant to treatment. This is an added perk to current study which may screen responders to therapy prior to treatment.

In another novel addition, the authors have found through a screen of 24 compounds -structures not shown, except for RMY-186- in HEK-293T cells transfected with CAD-D1371A mutant, that one molecule among the small library, named RMY-186 is active in both wild type and mutant CAD, and acts through promotion of degradation by ubiquitination of CAD. This part seems to be a repurposing of a previous scaffold which was recently reported by the authors (reference 72) to be active against pulmonary fibrosis in SARC-CoV-2 nucleocapsid protein. Mechanism not provided.

Chemistry:

The synthesis route is a rather linear 7-step procedure. The last step is reported at a non-optimized low yield of 19.7%. In

that particular step, the N-alkylation of the secondary amine with a benzylic halide would have propensity to form a quaternary ammonium rather than stop at the tertiary amine level, which may explain the low yield. Therefore, it could have been preferably accomplished through a reductive amination using either sodium cyanoborohydride or sodium triacetoxyborohydride reaction on 3-bromo benzaldehyde and piperazine intermediate 8. Both reducing agents are compatible with the ketone in intermediate 8. Most needed aldehydes are readily available.

The analytical profiling and structural assessment of RMY-186 are adequate.

The authors have used microscale thermophoresis, in what is considered a primary assay, to test the library for activity against CAD. In addition, they mentioned in passing that RMY-186 is not active alone. It is rather active in combination with 5-FU. This, along with previously reported activity against nucleocapsid protein, leave a lot to validate with respect to actual target(s) given that RMY-186 is only active at the micromolar level. Nevertheless, RMY-186 offers a good tool to work with for the time being.

In another entry, the authors used alphafold2 to predict the CAD structure and have docked RMY-186 therein. The resulting structure that is shown has plenty of unfolded hinge areas and looked more globular than a single entity with cooperative units. That may hint at the need for additional binding partners for CAD to be fully structured. That too may put into question the functionality of the presented fold. Absent an X-Ray structure of a co-crystal, or affinity pulldown, one should not venture too far based on just modeling. Moreover, the binding pose as suggested has the hydroxyflavone making all relevant contacts and benzyl piperazine facing the solvent area, should predict that nearly all the scaffold members equally are active since they all share the same hydroxyflavone moiety.

One can then wonder about the inclusion of both the modeling work and the chemistry in this manuscript and how much value do they add. I don't think they add any value to the biological findings.

The synthesis description is nearly identical (verbatim with no effort to even correct prior typos) in this manuscript as well as in reference 72. In the current manuscript the synthesis route is described rather succinctly with analytical data reported only for the final compound RMY-186.

The text has a lot of typos and at times, RMY-186 was misspelled as RNY-186 and the number indicated at RMY-183 in one instance.

References 7, 14, 23, 49, 54, 56, 58, 63, 68, 73, and 74 are incomplete.

Reviewer #5

(Remarks to the Author)

Ma et al. reported that CDA degradation is an important step in chemotherapy-induced apoptosis in gastric and colon cancer cells. They reported that caspase-3 is a major protease that targets Asp1371 for CAD degradation. Expression of Asp1371 mutant CAD showed treatment resistance in xenografts and a mouse model of gastric cancer. Sequencing of treatment-resistant primary tumors identified two cases harboring somatic missense mutations at Asp1371, and these mutants also showed drug resistance in cell models. The authors identified a compound that induces CAD degradation and showed that this compound can overcome chemoresistance in vivo.

Duplication of results from a previous study

1. This is an interesting study showing that CAD degradation is essential for chemotherapy-induced apoptosis of cancer cells. However, the novelty of their finding that CAD is cleaved by caspase 3 is questionable because a previous paper has already reported this (Caspase-dependent cleavage of carbamoyl phosphate synthetase II during apoptosis. Huang M, Kozlowski P, Collins M, Wang Y, Haystead TA, Graves LM. *Mol Pharmacol*. 2002;61(3):569-77.). Therefore, the authors should cite this work and delete duplicate results (Fig2e~o and Figure 3) from the previous report.
2. The previous paper identified two cleavage sites by Caspase 3, which is inconsistent with the results of the current study. The authors should test another cleavage site and confirm the difference and discuss about the results.

Sequencing of patient samples

It is quite important but also should be carefully evaluated that the authors found CAD1371 mutations in primary treatment-resistant samples.

1. Figure 5a shows that mutant reads are ~50% of the total reads. This is not consistent with the percentage of mutant alleles in Figure 5b/c (12~15%).
2. Are these mutants heterozygous? Did the authors detect copy number loss (LOH) of this gene in the two patient samples?
3. If the mutants are heterozygous in the patient samples, the authors should test whether the heterozygous mutant allele is sufficient to acquire drug resistance in cell line/mouse models.
4. Searching the public mutation database (COSMIC) does not reveal any mutation at Asp1371 in any type of cancer samples ever sequenced, suggesting that these mutants are so rare. This should be mentioned in the Discussion section as the cohort analyzed in this study is too small.
5. Did the authors find gene amplification of the CAD genes in cases without somatic mutations?

Reviewer #6

(Remarks to the Author)

In this study, Ma et al. investigated the role of the pyrimidine synthesis pathway in the chemosensitivity of gastric tumor models. They found that chemotherapeutic drugs, such as 5-fluorouracil (5-FU), can promote caspase-3-mediated cleavage and subsequent degradation of CAD (carbamoyl-phosphate synthetase 2, aspartate transcarbamylase, and dihydroorotase), an enzyme involved pyrimidine biosynthesis. They go on to report that over-expression of CAD (or mutations in the Asp1371 cleavage site) can lead to “tumor resistance”. Moreover, they found mutations “related” to Asp1371 in tumor patients who underwent unsuccessful neoadjuvant chemotherapy. Finally, they identified a compound, RMY-186, that appears to enhance chemosensitivity. This study contains many interesting observations and, if correct, may ultimately lead to improved treatment options for GC and GRC. However, the presentation is at times confusing, and several of the main novel conclusions need stronger experimental support.

Specific points:

As a general comment, this manuscript contains a lot of data, but the photomicrographs of cells are often too small and hard to evaluate. Therefore, I was not able to fully evaluate some of the relevant experiments.

1. Based on the results shown in Fig. 1, the authors conclude that supplementation of cells rU, dT and dC prevents 5-FU-induced apoptosis. It would be important to show specificity by examining effects on apoptosis induced by ionizing radiation.
2. I have some questions regarding the specificity of RMY-186 and conclusions derived from using this compound. The authors show that RMY-186 can promote the degradation of CAD, including in cells over-expressing CAD-D1371A (and other mutations in the cleavage site). Somewhat surprisingly treatment of GR and CRC cells with RMY-186 did not induce apoptosis in cells with significantly reduced levels of CAD. Moreover, RMY-186 enhanced sensitivity of both WT CAD and CAD-D1371A knock-in mutant cells to 5-FU. Why is this? According to the authors model, it would be expected that CAD-D1371A knock-in cells are (considerably) more resistant to combination treatment than CAD WT cells.
2. The authors state that CAD cleavage is required for 5-FU-induced apoptosis (line 320). This statement is too strong and not supported by the data presented (see point #2). They show a reduced effect of 5-FU on the size of xenografts, but they do not demonstrate that this is mediated by resistance towards apoptosis, and they most definitely fail to show a strict requirement of CAD for 5-FU-induced apoptosis.
3. Along those lines, the effect of 5-FU is complex and has been investigated for decades. For example, incorporation into RNA is responsible for the gastrointestinal toxicity of 5-FU in mice (Houghton JA, Houghton PJ, Wooten RS. Mechanism of induction of gastrointestinal toxicity in the mouse by 5-fluorouracil, 5-fluorouridine, and 5-fluoro-2'-deoxyuridine. *Cancer Res.* 1979 Jul;39(7 Pt 1):2406-13. PMID: 156065), and it has been reported that 5-FU incorporation into RNA but not DNA was associated with cell death (Geoffroy FJ, Allegra CJ, Sinha B, Grem JL. Enhanced cytotoxicity with interleukin-1 alpha and 5-fluorouracil in HCT116 colon cancer cells. *Oncol Res.* 1994;6(12):581-91. PMID: 7787251). The current study focuses on DNA damage (H2A.X), and it would be important to also investigate effects on RNA biology.
4. The results from over-expression of CAD (both WT CAD-1371 mutants) need to be taken with a grain of salt since over-expression of caspase substrates can inhibit caspase activity.
5. The authors' model predicts that mutating the caspase-3 cleavage site in CAD should lead to specific protection against 5-FU-induced apoptosis. This should be tested, for example, by examining the response of these cells to ionizing radiation (evaluating H2A.X, PARP cleavage, etc.).
6. “CAD cleavage by caspase-3 is necessary and sufficient for chemosensitivity” (Fig. 4). This is a gross overstatement that should be toned down – at best the data show an effect, but it is not as strict as implied.
7. It is stated (for example, in line 464) that RMY-186 when used in combination with 5-FU induces apoptosis in cells carrying the CAD-D1371A knock-in mutation, yet later on observations are cited that chemosensitized cells appear to undergo pyroptosis. This is confusing and needs better documentation and explanation/discussion.
8. Along these lines, the model (Fig. 8) is not clear at all and overstates what can be concluded from the available data. First, it implies a far greater role of CAD-cleavage in the execution of apoptosis than what the data support. Second, the proposed mechanism of action of RMY-186 is only depicted for “high heterogenous tumors” – why? I suggest to present a model that focuses on the difference between cells with WT and cleavage-resistant CAD. Finally, why does caspase-3 cleavage and degradation of CAD cause apoptosis and not pyroptosis in the left panel, whereas RMY-186-induced degradation of CAD supposedly causes pyroptosis? This is particularly confusing in light of the authors' data supporting induction of apoptosis (Fig. 7: cleaved PARP, p53, pH2A.X). There are clearly many other mechanisms at play to account for the observations and the current illustration is not very helpful.

Minor comment:

The English is often a bit rough and should be corrected throughout the ms. For example, in the abstract the sentence “By searching for a compound that capable for degrading the CAD-Asp1371 mutant, we discovered, ...” should be corrected to: “By searching for a compound that is capable of degrading the CAD-Asp1371 mutant, we discovered ...”

Version 1:

Reviewer comments:

Reviewer #1

(Remarks to the Author)

Major critique has been satisfactorily addressed.

Reviewer #2

(Remarks to the Author)

The authors have responded thoroughly to all of my critiques. I have no further comments and congratulate them on an interesting set of observations.

Reviewer #3

(Remarks to the Author)

After reading through the comments and updates provided by the authors, I am satisfied with their explanations and changes to their manuscript. The manuscript is formatted in a much tidier way with the additional figures in the supplemental information section.

Reviewer #4

(Remarks to the Author)

The heterogeneity within gastric cancer (GC) and colorectal cancer (CRC) is well established and is attributed to among other factors, the dysregulation of metabolic reprogramming as key to resistance to chemotherapy-mediated DNA damage. In their manuscript, Ma. J. et al, showed that CAD, a fusion of glutaminase (GLN), glutaminase-dependent carbamoyl phosphate synthase (CPS-2), aspartate transcarbamoylase (ATC), and dihydroorotase (DHO), is the first rate-limiting enzyme in the de novo pyrimidine synthesis pathway, and importantly a caspase-3 substrate. CAD needs to be cleaved by caspase-3 at Asp1371 prior to ubiquitin-dependent degradation. In patient samples, mutations at Asp1371 confer resistance to several chemotherapeutic agents. Through a screening campaign and a modest SAR they identified RMY-186 as an inhibitor of CAD which, when combined with 5-FU overcomes resistance to chemotherapy. These findings and others offer CAD as a promising therapeutic target and possibly a marker for therapeutic resistance.

Looking at the reviews including mine, and I am only the Organic Chemist, I feel like the authors have done a great job addressing all issues raised by our colleagues and myself. This is very apparent in the extensive number of experiments and figures that were subsequently added.

Looking at all the reviews including mine, and I am only the Organic Chemist, I feel like the authors have done a great job addressing all issues raised by our colleagues. This is very apparent in the extensive number of experiments and figures that were subsequently added.

As pointed out by my colleagues, the manuscript size is enormous, and the number of figures goes beyond the space allowed. Consolidating figures, discussions, and deciding what goes to supporting section will be a daunting task.

There are still some typos as well as rough expressions which should be addressed.

Reference 82 is incomplete and Page number should be added J Vis Exp. (78): 50541 (2013)

As far as I am concerned, the chemistry is sound and well controlled.

We are all aware of metabolic reprogramming and the major role that it plays in cancer resistance and evasiveness. This paper is another addition to the field in which colleagues working on cancer therapeutics should know about. The addition of a small molecule inhibitor of CAD will offer a new tool with which to further study the pathway.

The overall merit for publication in this prestigious journal is met.

Reviewer #5

(Remarks to the Author)

The authors have responded appropriately to all of the reviewer's comments. I have no further comments on this revision.

Reviewer #6

(Remarks to the Author)

The authors have responded very well to the criticism raised during the previous review, provided important clarification and

extensive new data supporting their conclusions. Therefore I recommend rapid publication of this interesting manuscript at this time.

Responses to Reviewer #1

We are extremely grateful for your constructive suggestions, which have helped substantiate the conclusions. We have now carefully addressed all of the suggestions made by you, either by providing clarification or by performing new experiments. In particular, we have investigated the potential co-localization of CAD and cleaved caspase-3 during the cleavage of CAD by caspase-3, and found that they did not co-localize. We have also identified the sites of CAD necessary for RMY-186 binding, which is also required for the subsequent ubiquitination and degradation of CAD. Point-by-point responses to your comments are given below, with your comments given verbatim in plain italic type, and our responses in bold:

Reviewer #1 (Remarks to the Author):

In this study, Ma et al. describe the role of the de novo pyrimidine synthesis pathway in determining the chemosensitivity of gastric and colon cancer cells. The study demonstrates that chemotherapeutic drugs can induce the degradation of CAD, a key enzyme in pyrimidine synthesis, leading to apoptosis in chemosensitive cells, and identifies the critical role of caspase-3-mediated cleavage of CAD at the Asp1371 residue, with mutations at this site causing chemoresistance. The authors further identify RMY-186 as a potential therapeutic agent to degrade mutant CAD and overcome chemoresistance. Overall, this study provides valuable insights into the molecular mechanisms underlying chemotherapy resistance and suggests a potential therapeutic strategy to improve treatment outcomes in cancer patients. The scope of the work is comprehensive, and this study could have an impact on the field of cancer therapy, particularly in understanding and addressing chemoresistance. The impact of the study can be improved by addressing the comments below.

We thank you for the positive evaluation.

Major comments:

• The authors did not provide any validation for caspase-3 KO mice. IHC/IF staining for caspase-3 expression in tissues from these mice should be included.

Thank you for your kind suggestion. This validation was in fact conducted during the generation of the ATK mice model, and we apologize for not including this data earlier. As shown below, the caspase-3 protein was not detected in the GC tissue of knockout Cldn18-ATK mice, indicating that caspase-3 was successfully knocked out. These data are now also included in the current Fig. 2n.

• Fig. 2t – Authors claim that the caspase-3 KO mice are not significantly affected by 5-FU treatment, but the Ki67 expression seems to be decreased in the 5-FU-treated sample. Authors should provide quantification for Ki67 and examine if the difference is significant. If there is a significant difference, authors should elaborate more on that result.

We thank you for pointing out the quantification of Ki67 expression in the tumor samples from caspase-3 KO mice treated with 5-FU. We have now performed this quantification and, as shown below (also listed as current Fig. 1o and Fig. 2o), we found that the decrease in Ki67 expression induced by 5-FU is no longer present in the caspase-3 knockout Cldn18-ATK mice (right) when compared to the wild-type mice (left).

Intrigued by your suggestion, we also quantified the Ki67 expression in the tumor samples from the Cldn18-ATK *Cad*^{D1371A/D1371A} mice treated with RMY-186 and/or 5-FU. As shown below (current Fig. 7o), we found that 5-FU alone did not affect Ki67 expression in these tissues, and the combination of RMY-186 with 5-FU significantly decreased Ki67 expression. These results support our conclusion that RMY-186 enhances chemosensitivity in the 5-FU-resistant GC model.

• Fig. 2t - How representative is the c-PARP image shown? Is the cell death completely attenuated in the caspase-3 KO samples? If yes, is the reason for the demonstrated slight reduction in tumor growth (Fig. 2r) due to a reduction in proliferation? How is the expression of cleaved caspase-7 affected in these mice?

Thank you for your insightful comment. We totally agree that 5-FU-mediated cell death can occur independently of caspase-3. In fact, as shown in current Fig. 2l and Fig. 2o of our manuscript (also shown below), we observed that 5-FU slightly decreased the average tumor area and Ki67 expression in the GC tissues from caspase-3-KO, Cldn18-ATK mice, although these changes were not statistically significant.

In addition, it was previously reported that 5-FU can induce cell death by causing damage to DNA and RNA after being converted into nucleotide analogs such as FdUMP, FdUTP, and FUTP (PMID 6170079, PMID 2656050). Furthermore, it has been demonstrated that impaired ribosome biogenesis is a major mechanism for 5-FU-mediated direct cellular effect, which results in cell cycle arrest (PMID 39378883, PMID 34265272). We have now incorporated these discussions into our revised manuscript (lines 256-258, pages 12).

Following your suggestion, we have now determined the expression of cleaved caspase-7 in the GC tissues from the caspase-3 knockout Cldn18-ATK mice. We found that 5-FU effectively promotes the cleavage of caspase-7 (shown below). These results support the conclusion that 5-FU can still induce cell death even in the absence of caspase-3, as discussed above. However, we would like to point out that we did not observe any cleavage of PARP in these tissues, which is consistent with the representative data for PARP cleavage in caspase-3 knockout Cldn18-ATK mice GC tissues presented in the original Fig. 2s (current Fig. 2m). It is possible that in these tissues, caspase-3, rather than caspase-7, plays a dominant role in PARP cleavage. Again, we appreciate your insightful comment, which has greatly improved our understanding of the cell death mechanisms induced by 5-FU.

- *The focus of the study is CAD degradation induced by caspase-3 cleavage; however, the authors did not include any results showing their distribution/co-localization in vivo. It would be helpful to co-stain for cleaved caspase-3 and CAD using an IF assay to show their distribution in tissues and provide further evidence of the relationship between cleaved-caspase-3+ cells and CAD degradation. Is the expression of cleaved caspase-3 distributed differently in samples with and without CAD mutations? Co-IP assays could also show the interaction between the proteins.*

Thank you for pointing out the subcellular co-localization of cleaved caspase-3 and CAD. Following your suggestion, we have now determined their co-localization by staining these two proteins in MKN45 cells treated with 5-FU for varying durations. To ensure the consistency of our results, we began treating the cells with 5-FU at different starting times, but fixed (collected) them with formaldehyde at the same time. We also kept all imaging parameters, including "PMT voltage," "Offset," "Pinhole," and "Gain," unchanged for each image captured.

As shown below, we observed an increase in the cleaved caspase-3 signal alongside a decrease in CAD signal as the duration of 5-FU treatment was extended, which is consistent with the immunoblotting data presented in current Fig. 2b and Supplementary Fig. 2g. However, we did not observe any significant co-localization between the two proteins, except within the apoptotic bodies, which are not intact cell structures.

We also performed immunoprecipitation (IP) assays to determine the interaction between cleaved caspase-3 and CAD, as you suggested. As shown below on the left panel (current Supplementary Fig. 4b), we did not observe any interaction between endogenous cleaved caspase-3 and CAD in either MKN45 or HCT 116 cells, regardless of 5-FU treatment, using an antibody against endogenous CAD. As a control, this antibody successfully co-immunoprecipitated known CAD-interacting proteins, such as glutamate oxaloacetate transaminase 1 (GOT1), voltage-dependent anion-selective channel protein 3 (VDAC3), and uridine 5'-monophosphate synthase (UMPS) (PMID 37291265). In addition, we found no interaction between ectopically expressed CAD and caspase-3 (shown below on the right panel). Together with the immunostaining data shown above, these results indicate that cleaved caspase-3 does not form a stable complex with CAD. Such a lack of interaction might be attributed to the way cleaved (activated) caspase-3 engages with its substrates, potentially in a "kiss-and-run" manner, as also seen in other proteinases (PMID 37777960, PMID 38851188). This hypothesis is further supported by the strong interaction observed between the enzymatically inactive mutant of caspase-3, Casp3^{C163A}-p17, and CAD (shown below, right panel).

We also determined the expression levels of CAD and the cleavage of caspase-3 in cells with the CAD-D1371A mutation knock-in (also raised by Reviewer #2 in point #3). As shown below (current Supplementary Fig. 4j), the results indicate that the knock-in does not alter CAD expression levels. However, we found that the CAD-D1371A knock-in significantly inhibited the cleavage of PARP during 5-FU treatment, while the cleavage of caspase-3 remained unaffected (shown below, current Supplementary Fig. 4k). These findings are consistent with the results obtained from CAD ectopic expression experiments shown in our initial submission (current Fig. 1h and Supplementary Fig. 1l).

Following your suggestion, we also determined the distribution of cleaved caspase-3 and CAD in the GC tissues of Cldn18-ATK mice after treatment with 5-FU using immunohistochemistry staining. As shown below (upper panel), we found that in the cells positive for cleaved caspase-3 staining, CAD signals were barely detectable, implying that CAD in these cleaved caspase-3-positive cells may have already been cleaved and degraded. Supporting this finding, in the GC tissues from the uncleavable CAD-D1371A knock-in Cldn18-ATK mice, we observed a portion of cells with both cleaved caspase-3 and CAD signals positive. We also attempted to stain for cleaved CAD in these tissues; however, we found that the antibody for cleaved CAD was not applicable to immunohistochemistry. Together, our results suggest that cleaved caspase-3 and CAD may not form a stable complex.

- *The authors established a link between CAD mutations at the Asp1371 site and chemoresistance. Have they explored whether the mutated form of CAD has any other role in cancer cells? Does it affect clinical outcomes such as overall survival or progression-free survival and how does the mutation frequency in chemoresistant patients compares to a larger cohort.*

We totally agree with your opinion that CAD and its D1371A mutation has other roles irrelevant to chemoresistance. In fact, during the revision, we identified the role of CAD cleavage in influencing the radiosensitivity of GC cells. As suggested by Reviewer #6, we analyzed the roles of the CAD-D1371A mutation on ionizing radiation (IR)-induced apoptosis in GC cells. The results, shown below in the upper panels (current Supplementary Fig. 1j-k), show a radiation-dose-dependent reduction of CAD, accompanied by an increase in p53 and cleaved PARP in MKN45 and HCT 116 cells. This induction of apoptosis can be suppressed when CAD levels are restored through ectopic expression, similar to the effects observed under the 5-FU treatment (shown below on the middle and lower panels). Importantly, we discovered that CAD-D1371A renders these cells resistant to IR-induced apoptosis (shown below on the lower panels, current Supplementary Fig. 4h). We have now incorporated this data into our revised manuscript.

Following your suggestions, we have also determined the roles of CAD and its

D1371A mutation in regulating nucleotide levels in GC cells, because it affects both proliferation and prognosis in GC patients following chemotherapy or radiotherapy (PMID 36682222, PMID 37291265 — also listed as reference #17, #22 in our manuscript). As shown below on the upper panel (for information only), we found that expression of both wildtype CAD and CAD-D1371A significantly increases the levels of CTP, UTP, dCTP, and dTTP, with no significant difference between the two expressions. In addition, we investigated whether CAD-D1371A influences the formation of pyrimidinosomes, which are supramolecular complexes essential for de novo pyrimidine synthesis (PMID 37291265). As shown below on the lower panel (current Supplementary Fig. 4b), we found that both wildtype CAD and CAD-D1371A exhibit similar binding affinities with GOT1, VDAC3, and UMPS. We have now included these data and discussions (lines 321-327, page 15; lines 541-544, page 25) in our manuscript.

Regarding the larger cohort of chemoresistant patients to evaluate clinical outcomes and their relationship with CAD mutation frequency (also raised by Reviewer #5), we admit that this is crucial for a comprehensive understanding of CAD mutations as a potential prognostic biomarker for chemosensitivity. However, we would like to point out that obtaining tissue samples (from GC patients who have failed neoadjuvant chemotherapy) is extremely challenging. This difficulty arises because these patients typically do not undergo further surgery, making tissue collection

unfeasible. GC patients requiring neoadjuvant therapy are typically locally advanced cases deemed unresectable by surgical evaluation. Once neoadjuvant chemotherapy fails, it implies these patients cannot undergo surgery according to surgical principles and can only continue treatment with alternative chemotherapy regimens, thus making specimens hard to obtain. These specimens can only be acquired when patients develop complications such as bleeding, obstruction, or perforation necessitating palliative surgery — this being the sole circumstance allowing sufficient specimen collection from such cases.

Additionally, the scarcity of specimens complicates our ability to conduct robust statistical analyses, such as those assessing overall survival (OS) or progression-free survival (PFS). We have now added a paragraph of discussion in our revised manuscript (lines 527-534, page 24).

• *The authors present data showing that RMY-186 promotes the degradation of CAD and its mutants through increased ubiquitination. Do they have any insight about the mechanism through which RMY-186 enhances ubiquitination and degradation (the authors mention the use of MG-132 to restore the protein levels of CAD but not the potential mechanism)?*

We thank you for your insightful comment about the mechanisms of RMY-186 in mediating the ubiquitination and degradation of CAD, a point also raised by Reviewers #3 and #4. We have now identified the potential binding sites of CAD for RMY186 by conducting in-silico docking assays using the AutoDock Vina software, during which the structure of CAD was predicted using the CavityPlus platform (<http://www.pkumdl.cn/cavityplus>; PMID 29750256). After analyzing the surface feature matching, binding energy, and potential hydrogen bonds and non-covalent interactions, we identified four potential binding pockets on CAD:

- Pocket 1: Glu(E)513 and Arg(R)515
- Pocket 2: Asp(D)622 and Glu(E)630
- Pocket 3: Pro(P)741 and Trp(W)743
- Pocket 4: Glu(E)1214 and Arg(R)1221

We then individually mutated these pockets to determine the effects of RMY-186 on the promotion of CAD ubiquitination. As shown in the upper panel (current Fig. 6g and Supplementary Fig. 6k), we found that a double mutation of CAD-P741 and CAD-W743 (CAD-P741/W743A), which disrupts Pocket 3 (illustrated as a ribbon diagram in the lower panel), almost completely blocked the effects of RMY-186 on CAD ubiquitination.

We have also tried to identify the ubiquitination sites of CAD through protein mass spectrometry. As shown below (for information only), a total of 16 lysine sites on CAD were found to be ubiquitinated. These sites can be roughly categorized into three fragments of CAD, as defined during the identification of the caspase-3 cleavage site in our original submission (depicted in the current Fig. 3a and 3c, also shown below):

- K186 corresponds to Flag-CAD-1-394 (fragment #1);
- K696, K739, K747, K754, K760, K778, and K867 correspond to Flag-CAD-365-1118 (fragment #2);
- K1119, K1127, K1211, K1228, K1262, K1313, K1325, and K1411 correspond to Flag-CAD-691-1444 (fragment #3).

Mass spectra of ubiquitination modifications at 16 lysine sites of CAD

In line with the mass spectrometry results, we observed significant ubiquitination of fragments #2 and #3 of CAD when they were individually expressed in HEK293T cells. This ubiquitination was further enhanced by RMY-186, as shown in the upper panel below. Although fragment #1 also underwent ubiquitination, it was not promoted by RMY-186 (also shown in the upper panel), suggesting that the ubiquitination sites of CAD for its degradation by RMY-186 are located in fragments #2 and #3. Unfortunately, after individually mutating the 15 lysine residues in the truncated fragments #2 and #3, we did not observe any impairment in CAD ubiquitination mediated by RMY-186 (shown in the lower panels). Therefore, it appears that a combination of two or more lysine site mutations is necessary to definitively identify the sites required for RMY-186-mediated CAD ubiquitination. Nonetheless, we hope that at this stage, you will agree that the binding of RMY-186 to CAD effectively induces its ubiquitination.

We also attempted to identify potential E3 ligases that might be involved in catalyzing the ubiquitination of CAD. We performed quantitative mass spectrometry to analyze the changes in the interacting proteins of CAD (which were expressed and subsequently immunoprecipitated) in MKN45 cells treated with RMY-186 compared to those treated with DMSO control. This analysis revealed a total of 2052 proteins ($n = 3$ replicates for each treatment). The potential E3 ligases were then predicted using the “DEP” R package (PMID 29446774), which processed the raw data by filtering, correcting background noise, normalizing, and imputing missing values based on the MNAR definition using the “man” method. Subsequent differential

analysis was conducted using the “limma” package (PMID 25605792), referencing the Ubinet 2.0 database (PMID 33693667).

As illustrated in the volcano plots and heatmap below (generated with the “pheatmap” and “ggplot2” R packages; current Fig. 6h-i), we identified the dual E2 ubiquitin-conjugating enzyme/E3 ubiquitin-protein ligase BIRC6 and the ubiquitin-protein ligase E3C (UBE3C) as the potential E3 ligases responsible for the ubiquitination of CAD. We have now incorporated this data into our revised manuscript.

• Additionally, is this effect specific to CAD, or could RMY-186 also impact other proteins? Expanding on the specificity of RMY-186 would help to ensure that potential therapeutic effects are not accompanied by unintended consequences, which would be important for potential future clinical application. Have the authors explored whether RMY-186 enhances the efficacy of other standard treatments in resistant cancer models?

We appreciate your pointing out the requirement to investigate whether RMY-186 can enhance the efficacy of other standard treatments in resistant cancer models. As shown below, we added a combination of the chemotherapy drug oxaliplatin (Oxa) and RMY-186 to treat the MKN45 and HCT116 cells carrying the CAD-D1371A knock-in mutation. The combination of RMY-186 and Oxa similarly caused the resistance of the resistant model to death to disappear (listed as current Supplementary Fig. 7c).

- There is again no assay provided to show Caspase-3/CAD distribution in 5-FU/RMY-186 mouse samples.

As detailed in points #4 above, we also explored the distribution of cleaved caspase-3 and CAD in the gastric tumor tissues of Cldn18-ATK *Cad*^{D1371A/D1371A} mice after 5-FU/RMY-186 treatment. As shown below, the IF results of the tissues still do not clearly demonstrate their distribution/co-localization.

Minor comments:

- The authors discuss multiple pathways in the introduction section, but the complexity may be challenging for readers to follow. Including a schematic illustration would greatly enhance clarity and understanding of the mechanisms involved.

We thank you for the insightful comment and have now revised the introduction section to improve clarity and provide a more straightforward explanation of the mechanisms involved.

- Authors should include the cell line information in all figure plots, not only in figure legends (Fig. 1h-l, Fig. 4a-d, etc.).

We have now included the cell line information in all figure plots of our revised manuscript. Thank you for the suggestion.

- Fig. 2 – Authors should improve the arrangement of panels; they are not formatted in a proper order logical to follow (j-k-p?). This applies to other figures as well, due to the large amount of data included, at times they are very difficult to follow. Authors should consider adjusting them for clarity.

We apologize for the lack of clarity in the panels and have carefully rearranged the figure panels of Fig. 2 and Supplementary Fig. 2, 4, 5 to ensure they are formatted in the correct order. Thanks for the suggestion.

- Fig. 2a-b - It is unclear where the CQ and MG-132 treated samples are shown.

We apologize for the unclear labeling and have reformatted Fig. 2a-b (current Fig. 2c and Supplementary Fig. 2c-d, h) to make the results for the CQ and MG-132 treated group more distinguishable.

- Line 311 – Typo: “haft-life” instead of “half-life”

We apologize for the typo and have corrected it.

Responses to Reviewer #2

We are extremely grateful for your constructive suggestions, which have helped substantiate the conclusions. We have now carefully addressed all of the suggestions made by you, either by providing clarification or by performing new experiments. In particular, we investigated whether the D1371A mutation of CAD affects the protein levels of CAD itself, as well as the cleavage of caspase-3 and PARP induced by 5-FU. In addition, we have explored the effects of other chemotherapy drugs, besides 5-FU, on the cleavage of CAD and the induction of apoptosis. Point-by-point responses to your comments are given below, with your comments given verbatim in plain italic type, and our responses in bold:

Reviewer #2 (Remarks to the Author):

In their manuscript, Ma and Zhao et al describe the caspase-3 mediated cleavage of the pyrimidine biosynthetic enzyme CAD during apoptosis. They demonstrate that caspase-3 is necessary and sufficient for CAD cleavage. They map the caspase-3 cleavage site to D1371 within the CPSase domain of CAD. Overexpression of CAD, especially a CAD mutant that cannot be cleaved by caspase 3, increases pyrimidine pools and survival upon 5-FU treatment, while blunting H2AX phosphorylation. Expression of the CAD D1371A mutant renders tumors somewhat resistant to 5-FU treatment. The authors identify tumors with the CAD D1371 mutation and demonstrate that cells expressing these variants are also relatively 5-FU resistant. Finally, the authors identify a small molecule that induces CAD proteasomal degradation. Combined inhibition of CAD and 5-FU hastens cell death, which is attributed to increased pyroptosis, and this combination further reduces tumor burden. Overall, the authors provide a thorough assessment of the impact of apoptosis on CAD and the consequent effect on 5-FU sensitivity.

We thank you for the positive evaluations.

The experiments are well-controlled and technically well-done. My main comments concern the conclusions that the authors have drawn, which at times tend to be over-interpretations of the data presented. In particular, the authors frequently conflate 5-FU with all chemotherapy, when the protective effect of the CAD D1371 mutant may be highly specific to pyrimidine analogs. These comments can largely be addressed by substantially modifying the text.

Major Points:

1. In the abstract the authors state” we found that the de novo pyrimidine synthesis pathway, which we previously identified as crucial for the proliferation of gastric and colon cancer cells, determines the chemosensitivity of these cancer types” and “conferred tumor chemoresistance”, “resulting in a significant improvement in the

effectiveness of chemotherapy”, and refer to “chemotherapy” in several other places, including the body of the manuscript. These statements inappropriately imply that pyrimidine synthesis impacts chemosensitivity broadly, when the authors have only really focused on 5-FU sensitivity.

We thank you for pointing out the generality of CAD cleavage in determining chemosensitivity. In our initial submission, we did examine the effects of chemotherapy drugs other than 5-FU, including Oxaliplatin (Oxa), Doxorubicin (Dox), and Paclitaxel (PTX), on CAD cleavage. As shown below (current Fig. 1a and Supplementary Fig. 1a), all these drugs, along with 5-FU, can lead to CAD cleavage. During the revision, we also analyzed the effect of Venetoclax (Vene), an apoptosis inducer, on CAD cleavage, as suggested by you (minor point #1), and we observed CAD cleavage as well. We also explored other apoptosis inducers, such as Staurosporine (STS) and Raptinal (Rap), and found that they also induce CAD cleavage (shown below, current Supplementary Fig. 2f). Therefore, CAD cleavage can occur in response to various cytotoxic agents, including chemotherapy drugs. We sincerely hope you will agree that this is closely related to chemosensitivity.

2. The authors interpret the results upon expression of CAD as evidence that CAD degradation is necessary for apoptosis induction. However, the authors do not report the degree of CAD protein expression in Figures 1, 4a-m, and 5g-h. Therefore, it is difficult to know whether CAD is being over-expressed and driving UMP/thymidine synthesis, diluting the effect of FdUMP. Moreover, the relative over-expression of CAD versus the CAD mutant may explain the data presented in Figure 4. Understanding the relative expression in these models is needed.

We thank you for the insightful comment regarding the protein level of CAD when ectopically expressed in the 5-FU-treated cells. In fact, to ensure that the ectopically expressed CAD levels are comparable to those in the untreated state, we titrated the amount of expression plasmid used for transfection in each cell line. For HGC27 cells, we found that 0.3 µg of plasmid per well in a 6-well dish effectively restores CAD expression (shown below on the left, current Fig. 1c and Supplementary Fig. 1c), while for MKN45, HCT116 and SW480 cells, 0.4 µg, 0.35 µg and 0.45 µg of plasmid are needed, respectively. Using this approach, we successfully detected the effects of restored CAD on inhibiting PARP cleavage and p-H2A.X under 5-FU treatment in each cell line.

Similarly, we titrated the plasmid amounts used for restoring CAD mutants, including CAD-D1371A, CAD-D1371E, and CAD-D1371Y, in each cell line (shown below; with CAD-D1371A as the representative mutant). Our findings indicate that these mutants can effectively block 5-FU-induced apoptosis.

We totally agree that the ectopic expression of CAD, even if not at an over-expressed level, will dilute the pool of FdUMP required for 5-FU-mediated apoptosis, thereby impairing cell death. In fact, in our original submission, we demonstrated that the restoration of CAD significantly increases the levels of UTP, CTP, dCTP, and dTTP (also shown below, current Fig. 4a), which can contribute to the dilution of FdUMP, as you suggested.

In addition, we would like to point out that, aside from diluting FdUMP and thereby suppressing 5-FU-mediated apoptosis, the nucleotide pool maintained by CAD can also inhibit apoptosis, as we and others have previously suggested (PMID 38869064, PMID 36682222). This is further supported by data obtained during the revision (suggested by Reviewer #6 in point #5). As shown below, we found that Raptinal, an activator of caspase-3 that does not result in the accumulation of FdUMP, directly leads to the cleavage of CAD. As a control, Raptinal also caused the cleavage of RhoGDI, which is another substrate of caspase-3 (PMID 10381642, PMID 11989976). Importantly, we observed a significant cleavage of PARP during Raptinal treatment, which could be inhibited by the restoration of CAD through ectopic expression. Therefore, CAD can suppress apoptosis in a manner that is independent of 5-FU and FdUMP.

3. “we generated the site directed mutant MKN45 and HCT116 cells with homozygous CAD-D1371A integrated into the genome”. The authors need to provide genotyping data to support this statement, should indicate the total level of CAD protein in these engineered cells, and determine whether PARP and caspase-3 cleavage is affected upon 5FU treatment.

We thank you for pointing out the validation of MKN45 and HCT116 cells with the CAD-D1371A knock-in. We have, in fact, validated these cell lines before the original submission, and we apologize for their omission in the initial submission. The sequencing results for the CAD locus of MKN45 (shown in the left panel) and HCT116 (in the right panel) indicate either homozygously wildtype CAD (WT/WT) or CAD-D1371A (D1371A/D1371A) knock-in (also shown in current Supplementary Fig. 4i). In addition, during the revision, we generated heterozygous CAD-D1371A (D1371A/+) knock-in MKN45 and HCT116 cells, as suggested by Reviewer #5, and we have also included the validation data for these new cell lines (shown below, current Supplementary Fig. 4i). The presence of double, overlapping peaks in the CAD-D1371A/+ cells indicates the coexistence of both the wildtype and CAD-D1371A sequences.

Following your suggestion, we determined the protein levels of CAD in these knock-in cells. The results indicate that the knock-in does not alter CAD expression levels. However, we found that the CAD-D1371A knock-in significantly inhibited the

cleavage of PARP during 5-FU treatment, while the cleavage of caspase-3 remained unaffected (shown below, current Supplementary Fig. 4j-k). These findings are consistent with the results obtained from CAD ectopic expression experiments shown in our initial submission (current Fig. 1h and Supplementary Fig. 1l).

Minor Points:

1. It would be helpful for the interpretation of the data to understand whether another method of inducing apoptosis that does not involve a nucleotide or nucleotide analog also is impacted by CAD D1371A expression. Is this phenotype specific for 5-FU and other pyrimidine analogs, or is CAD cleavage truly hastening apoptosis when stimulated in other ways? While in my opinion this experiment is not needed for publication, it would substantially increase impact if CAD cleavage is required for apoptosis induction beyond nucleotide analogs (e.g. venetoclax).

As discussed in major point #1, we have now determined the effects of Venetoclax, as you suggested, along with other apoptosis-inducing agents such as Oxaliplatin, Staurosporine, and Raptinal on CAD cleavage (as shown in the current Supplementary Fig. 2f). We found that CAD cleavage can occur not only with 5-FU treatment but also with these other apoptotic agents. Again, we thank you for the insightful suggestion, which expands the significance of CAD cleavage in the context of apoptotic cell death.

2. Authors state that pyrimidine analogs “prevented the 5-FU-induced apoptosis”. It would be more accurate to say that they suppressed H2AX phosphorylation as this figure does not measure specific apoptosis markers.

We appreciate your kind suggestion and have revised the description of “5-FU-induced apoptosis” to include more precise terminology, such as “PARP cleavage,” to enhance the accuracy of our experimental descriptions.

3. Authors state, “we observed that caspase-9, which is specifically involved in the intrinsic apoptosis pathway, was responsible for the CAD cleavage”. This wording implies that caspase-9 is directly cleaving CAD, which I don’t think was the authors’ intent.

We have now changed “was responsible for the CAD cleavage” to “was related to the CAD cleavage”. Again, we thank you for such a kind suggestion.

4. “a pile of peptides with the N-terminal residue of Asp1371 was hit”. Could the authors be more precise about what these data show?

We have now indicated this information in our revised manuscript (lines 287-290, pages 14).

Responses to Reviewer #3

We are extremely grateful for your constructive suggestions, which have helped substantiate the conclusions. We have now carefully addressed all of the suggestions made by you, either by providing clarification or by performing new experiments. In particular, we confirmed the direct interaction between RMY-186 and CAD, and identified the sites of CAD required for RMY-186 binding. We also investigated how RMY-186 enhances the ubiquitination of CAD by determining the region of CAD that can be ubiquitinated. Furthermore, we assessed the role of RMY-186 in increasing sensitivity to anti-PD-1 immunotherapy. Point-by-point responses to your comments are given below, with your comments given verbatim in plain italic type, and our responses in bold:

Reviewer #3 (Remarks to the Author):

- *What are the noteworthy results?*
 - o *The authors discovered a correlation between a reduction in CAD cleavage and resistance to 5-FU treatment in cells and in mouse models. The native cleavage process was found to be performed by caspase-3 within the apoptosis pathway. CAD mutates to D1371A in a small population of resistant cancer patients, preventing caspase-3 cleavage and maintaining full length CAD. A small molecule RMY-186 was found to increase apoptosis and pyroptosis in cancer cells and tumors.*
- *Will the work be of significance to the field and related fields? How does it compare to the established literature? If the work is not original, please provide relevant references.*
 - o *The work is definitely of interest to the field as chemoresistance is a large problem in cancer therapeutics. This work was very detailed in determining the mechanism of the pathway of chemoresistance and the mode of chemosensitization.*

We thank you for your positive evaluations.

- *Does the work support the conclusions and claims, or is additional evidence needed?*
 - o *Is there any experimental evidence to support the docking of RMY-186 into CAD-D1371A in cells? The correlation between RMY-186 and ubiquitination/degradation of CAD is clear, and the authors calculated a Kd for RMY-186 with CAD-D1371A-GFP but no in cell experiments were done to confirm this was the target. A Kd of 6.78 μ M is not terribly strong binding and the cellular environment would likely have competing enzymes.*

We thank you for the insightful comment on the interface between RMY-186 and CAD (also raised by Reviewer #1 and Reviewer #4), which we are also intrigued with. We have now identified the potential binding sites of CAD for RMY186 by conducting in-silico docking assays using the AutoDock Vina software, during which the

structure of CAD was predicted using the CavityPlus platform (<http://www.pkumdl.cn/cavityplus>; PMID 29750256). After analyzing the surface feature matching, binding energy, and potential hydrogen bonds and non-covalent interactions, we identified four potential binding pockets on CAD:

- Pocket 1: Glu(E)513 and Arg(R)515
- Pocket 2: Asp(D)622 and Glu(E)630
- Pocket 3: Pro(P)741 and Trp(W)743
- Pocket 4: Glu(E)1214 and Arg(R)1221

We then individually mutated these pockets to determine the effects of RMY-186 on the promotion of CAD ubiquitination. As shown in the upper panel (current Fig. 6g and Supplementary Fig. 6k), we found that a double mutation of CAD-P741 and CAD-W743 (CAD-P741/W743A), which disrupts Pocket 3 (illustrated as a ribbon diagram in the lower panel), almost completely blocked the effects of RMY-186 on CAD ubiquitination.

We have also identified the ubiquitination sites of CAD required for RMY-186-mediated degradation, through protein mass spectrometry. As shown below (for informational only), a total of 16 lysine sites on CAD were found to be ubiquitinated. These sites can be roughly categorized into three fragments of CAD, as defined during the identification of the caspase-3 cleavage site in our original submission (depicted in the current Fig. 3a and 3c, also shown below):

- K186 corresponds to Flag-CAD-1-394 (fragment #1);
- K696, K739, K747, K754, K760, K778, and K867 correspond to Flag-CAD-365-1118 (fragment #2);
- K1119, K1127, K1211, K1228, K1262, K1313, K1325, and K1411 correspond to Flag-CAD-691-1444 (fragment #3).

Mass spectra of ubiquitination modifications at 16 lysine sites of CAD

In line with the mass spectrometry results, we observed significant ubiquitination of fragments #2 and #3 of CAD when they were individually expressed in HEK293T cells. This ubiquitination was further enhanced by RMY-186, as shown in the upper panel below. Although fragment #1 also underwent ubiquitination, it was not promoted by RMY-186 (also shown in the upper panel), suggesting that the ubiquitination sites of CAD are located in fragments #2 and #3. Unfortunately, after individually mutating the 15 lysine residues in the truncated fragments #2 and #3, we did not observe any impairment in CAD ubiquitination mediated by RMY-186 (shown in the lower panels). Therefore, it appears that a combination of two or more lysine site mutations is

necessary to definitively identify the sites required for RMY-186-mediated CAD ubiquitination. Nonetheless, we hope that at this stage, you will agree that the binding of RMY-186 to CAD indeed induces its ubiquitination.

We also attempted to identify potential E3 ligases that might be involved in catalyzing the ubiquitination of CAD. We performed quantitative mass spectrometry to analyze the changes in the interacting proteins of CAD (which were expressed and subsequently immunoprecipitated) in MKN45 cells treated with RMY-186 compared to those treated with DMSO control. This analysis revealed a total of 2052 proteins (n = 3 replicates for each treatment). The potential E3 ligases were then predicted

using the “DEP” R package (PMID 29446774), which processed the raw data by filtering, correcting background noise, normalizing, and imputing missing values based on the MNAR definition using the “man” method. Subsequent differential analysis was conducted using the “limma” package (PMID 25605792), referencing the Ubinet 2.0 database (PMID 33693667).

As illustrated as the volcano plots and heatmap below (generated with the “pheatmap” and “ggplot2” R packages; current Fig. 6h-i), we identified the dual E2 ubiquitin-conjugating enzyme/E3 ubiquitin-protein ligase BIRC6 and the ubiquitin-protein ligase E3C (UBE3C) as the potential E3 ligases responsible for the ubiquitination of CAD. We have now incorporated this data into our revised manuscript.

o The link of RMY-186 to increasing patient response to immunotherapy is unclear. (line 549-551) What evidence specifically supports that conclusion? The authors show that RMY-186 causes high levels of ubiquitination on CAD, leading to its degradation. They also showed that RMY-186 there was an increase in GSDME cleavage and subsequent pyroptosis. More data would be helpful to provide a stronger link between RMY-186 and its potential to provide immunotherapy sensitivity.

We thank you for the insightful comment (also raised by Reviewer #6). We proposed that the combination of RMY-186 and 5-FU treatment may enhance the response to immunotherapy because it can induce pyroptosis in GC cells. This process releases damage-associated molecular patterns (DAMPs), which help activate and increase the infiltration of anti-tumor immune cells. Consistently, it has been suggested that triggering pyroptosis could be an effective strategy to reverse the immunosuppressive microenvironment of GC, thereby enabling more successful immunotherapeutic treatment (PMID 32188940, PMID 32188939).

We have also investigated the effects of RMY-186 and 5-FU on the efficacy of the anti-PD-1 antibody, a classical immune checkpoint inhibitor (ICI), in suppressing GC in *Cldn18-ATK Cad^{D1371A/D1371A}* mice. We found that the combination of RMY-186 and 5-FU indeed enhances the effects of the anti-PD-1 antibody on GC suppression. This

triple treatment also significantly increased CD3⁺ and CD8⁺ T-cell infiltration compared to other treatments (assessed by staining for CD3 and CD8a, respectively; see current Supplementary Fig. 7p-r, also shown below), indicating a strong enhancement of immune responses in these tumors.

• Are there any flaws in the data analysis, interpretation and conclusions? Do these prohibit publication or require revision?

o Many conclusions are drawn from an extremely large number of experiments. There is very little explanation as to how they got to those conclusions. Some sections did have enough explanations (i.e. Figure 2 had more text regarding the rationale and conclusions drawn than many other figures)

Thank you for your kind suggestion. In the revised manuscript, we have added further explanations to clarify how the conclusions were drawn from the experiments.

o The article is written like a communication with minimal explanations yet is already as long as a full research article. It almost feels like this paper is too big with the

amount of figures and subsections of figures. If they authors plan to just state a conclusion that is made with data in Figure X parts h-k with no other substantiation, those figures may be better suited for supplemental information. If there are no page limitations, I suggest the authors expand more on their data analysis and interpretation.

Again, we thank you for the kind suggestion. We admit that the description of our results in this study is concise and lacks detailed explanation due to space limitations. Following your suggestion, we have moved some of the figures to the supplementary information section and expanded the data interpretation for better clarity.

- *Is the methodology sound? Does the work meet the expected standards in your field?*
 - o *Yes to both.*

We thank you for the positive evaluations.

- *Is there enough detail provided in the methods for the work to be reproduced?*
 - o *Specifics of fluorescence microscopy are not given in much detail. The methods describe the imaging itself, but does not define clearly the different variables and the conditions used. The figure captions list the conditions using abbreviations (i.e. EV, CAD, dT/dC) but the figures have more conditions in the results (i.e. OE-CAD, DMSO, Vector). It is unclear if vector is the same as empty vector. Sometimes it appears to be and other times it appears to be the transfected group. These comments apply to Figure 1 specifically, but more generally to all of the figures. More details should be presented either in the figure captions or in the methods section.*

We thank you for your kind suggestion, and have now included a detailed description of the fluorescence microscopy conditions and variables used in the methods section (lines 843-849, page 38-39). We have also defined all abbreviations used in this paper and ensured consistency throughout the manuscript.

o Figure 1 (d,e) and (f,g) have their labels swapped in the figure caption. Parts d and e are immunofluorescence and f and g are immunoblots.

We thank you for the comments and have corrected these mistakes.

o Many abbreviations are not defined such as OE-CAD. It would be useful to have an extensive abbreviations section where all are defined.

Again, we appreciate your kind suggestion and apologize for not clearly defining the abbreviations. All of them have now been defined in our revised manuscript.

- *General comments:*
 - o *The abstract is much longer than the 200 words stated by the journal*

We have now shortened the abstract to less than 200 words to comply with the guidelines of Nature Communications.

o There are some typographical errors throughout the manuscript

We have now carefully reviewed the manuscript and corrected all typos. Thanks for the comment.

Responses to Reviewer #4

We are extremely grateful for your constructive suggestions, which have helped substantiate the conclusions. We have now carefully addressed all of the suggestions made by you, either by providing clarification or by performing new experiments. In particular, we confirmed the direct interaction between RMY-186 and CAD, and identified the sites of CAD required for RMY-186 binding. We also investigated how RMY-186 enhances the ubiquitination of CAD by pinpointing the region of CAD that can be ubiquitinated. We also provided optimized procedures for the synthesis of RMY-186. Point-by-point responses to your comments are given below, with your comments given verbatim in plain italic type, and our responses in bold:

Reviewer #4 (Remarks to the Author):

Prior observations on the extensive heterogeneity within gastric (GC) and colorectal (CRC) cancers attributed the heterogeneity to dysregulation of metabolic reprogramming as key to resistance to chemotherapy-mediated DNA damage. Ma. J. et al, in their quest to further the mechanism of resistance in GC and CRC and following a continuum in their work which identified UHMK1, Dyrk3, and FOXK2 as modulators of nucleotide synthesis with direct influence on de novo reprogramming of nucleotide synthesis, has led them to CAD. CAD is the first rate-limiting enzyme in the de novo pyrimidine synthesis pathway, and also a caspase-3 substrate. In prokaryotes de novo pyrimidine synthesis is composed of several sequential steps each of which is encoded by a separate enzyme. However, in mammalian system, four enzymes in the pathway, namely glutaminase (GLN), glutaminase-dependent carbamoyl phosphate synthase (CPS-2), aspartate transcarbamoylase (ATC), and dihydroorotase (DHO) are fused into one single protein, called CAD.

In a novel aspect of current work, the authors have found that activated caspase-3 cleaves CAD at Asp-1371. This results in decreased pyrimidine synthesis and the ensuing imbalance in nucleotide pool increases DNA damage which triggers apoptosis. Tumors with elevated CAD expression can resist apoptosis by increasing pyrimidine synthesis. Further, mutations at caspase-3 cleavage site emerge resistant to treatment. This is an added perk to current study which may screen responders to therapy prior to treatment.

We thank you for the positive evaluation.

In another novel addition, the authors have found through a screen of 24 compounds -structures not shown, except for RMY-186- in HEK-293T cells transfected with CAD-D1371A mutant, that one molecule among the small library, named RMY-186 is active in both wild type and mutant CAD, and acts through promotion of degradation by ubiquitination of CAD.

This part seems to be a repurposing of a previous scaffold which was recently reported by the authors (reference 72) to be active against pulmonary fibrosis in SARS-CoV-2 nucleocapsid protein. Mechanism not provided.

We thank you for your insightful comments. As you noted, both RMY-186, identified here, and RMY-205, which we screened and used to treat SARS-CoV-2-induced pulmonary fibrosis (current reference #81, PMID 36889311), are derived from the same core structure. Both chemicals are among the 57 flavonoid derivatives that we designed and synthesized. This includes 32 hydroxyl-free chromogenic ketone compounds at position 3, and 25 hydroxyl-substituted chromogenic ketone compounds at position 3. In our previous screening for treating SARS-CoV-2-induced pulmonary fibrosis, we conducted structure-activity relationship assays on 17 out of the 57 compounds, identifying RMY-205 as the most effective one. In this study, we further screened an additional 24 compounds (shown below, current Supplementary Table 2) and identified RMY-186 as the most effective for CAD degradation. Furthermore, we found that RMY-186 can directly bind to CAD (as discussed below). It is important to note that RMY-205 has little effect on CAD degradation, suggesting that RMY-186 and RMY-205 exhibit different biological activities, despite being derived from the same core structure.

Supplementary Table 2. Structures of small molecule compounds.

Compound	R1	R2	Compound	R1	R2	Compound	R1	R2
RMY-179	H		RMY-202	H		WBR-6	Br	RMY-183	H		RMY-203	H		WBR-7	H	RMY-186	H		RMY-205	H		WBR-8	H	RMY-188	H		WBR-1	F		WBR-9	H	RMY-189	H		WBR-2	F		WBR-10	H	RMY-190	H		WBR-3	H		WBR-11	H	RMY-195	H		WBR-4	H		WBR-12	H	RMY-199	H		WBR-5	Br		WBR-14	H	
Chemistry:

The synthesis route is a rather linear 7-step procedure. The last step is reported at a non-optimized low yield of 19.7%. In that particular step, the N-alkylation of the secondary amine with a benzylic halide would have propensity to form a quaternary ammonium rather than stop at the tertiary amine level, which may explain the low yield. Therefore, it could have been preferably accomplished through a reductive amination using either sodium cyanoborohydride or sodium triacetoxyborohydride reaction on 3-bromo benzaldehyde and piperazine intermediate 8. Both reducing agents are compatible with the ketone in intermediate 8. Most needed aldehydes are readily available.

Thank you for your kind and thoughtful suggestion. We indeed considered a synthetic strategy similar to yours, initially. Unfortunately, we found that many of the reagents and derivatives required for this approach, especially those needed to replace the benzaldehyde, are difficult to obtain. This is the reason we did not pursue the strategy involving the reductive amination reaction on intermediate 8 of piperazine. We will certainly adopt the aldehyde reduction and amination method, as you suggested, whenever these reagents become available in the future.

The analytical profiling and structural assessment of RMY-186 are adequate.

The authors have used microscale thermophoresis, in what is considered a primary assay, to test the library for activity against CAD. In addition, they mentioned in passing that RMY-186 is not active alone. It is rather active in combination with 5-FU. This, along with previously reported activity against nucleocapsid protein, leave a lot to validate with respect to actual target(s) given that RMY-186 is only active at the micromolar level. Nevertheless, RMY-186 offers a good tool to work with for the time being.

Thank you for your insightful comment. As you mentioned, RMY-186 does not induce apoptosis on its own, unlike 5-FU and other chemotherapeutic drugs. This can be attributed to the fact that the increased demand for nucleotides in GC cells occurs only in the presence of genotoxic stress, not under normal conditions (current Fig. 7a and Supplementary Fig. 7a). In fact, our previous studies have demonstrated that knocking down CAD in GC cells does not lead to their death, although impairs their growth. In addition, during the revision process, we performed new experiments (in response to the suggestion raised by Reviewer #1 in major point #6) and obtained new results that show how the pro-apoptotic effects of other chemotherapy drugs, such as oxaliplatin (Oxa), can be enhanced by RMY-186 through the degradation of CAD (shown below, current Supplementary Fig. 7c). Therefore, we hope you agree that by targeting CAD, RMY-186 increases the chemosensitivity.

We completely agree that RMY-186 may have targets other than CAD, as you pointed out. However, we hope you agree that RMY-186 does bind CAD and leads to its degradation, as discussed below.

In another entry, the authors used alphafold2 to predict the CAD structure and have docked RMY-186 therein. The resulting structure that is shown has plenty of unfolded hinge areas and looked more globular than a single entity with cooperative units. That may hint at the need for additional binding partners for CAD to be fully structured. That too may put into question the functionality of the presented fold. Absent an X-Ray structure of a co-crystal, or affinity pulldown, one should not venture too far based on just modeling. Moreover, the binding pose as suggested has the hydroxyflavone making all relevant contacts and benzyl piperazine facing the solvent area, should predict that nearly all the scaffold members equally are active since they all share the same hydroxyflavone moiety.

Thank you very much for your insightful suggestions regarding the structural modelling (also raised by Reviewer #3). We have now uploaded the validated single-crystal structure data to the CCDC database and have optimized the docking model (listed as current Fig. 6a, also shown below).

According to your suggestion, we have now identified the potential binding sites of CAD for RMY186 by conducting in-silico docking assays using the AutoDock Vina software, during which the structure of CAD was predicted using the CavityPlus platform (<http://www.pkumdl.cn/cavityplus>; PMID 29750256). After analyzing the surface feature matching, binding energy, and potential hydrogen bonds and non-covalent interactions, we identified four potential binding pockets on CAD:

- Pocket 1: Glu(E)513 and Arg(R)515
- Pocket 2: Asp(D)622 and Glu(E)630
- Pocket 3: Pro(P)741 and Trp(W)743
- Pocket 4: Glu(E)1214 and Arg(R)1221

We then individually mutated these pockets to determine the effects of RMY-186 on the promotion of CAD ubiquitination. As shown in the upper panel (current Fig. 6g and Supplementary Fig. 6k), we found that a double mutation of CAD-P741 and CAD-W743 (CAD-P741/W743A), which disrupts Pocket 3 (illustrated as a ribbon diagram in the lower panel), almost completely blocked the effects of RMY-186 on CAD ubiquitination. Therefore, through more detailed molecular docking studies, we identified an additional residue, P741, alongside W743, which we initially documented. Both of these residues are critical for the binding of RMY-186 to CAD. We hope you will agree that identifying these interaction sites provides sufficient evidence to conclude that RMY-186 specifically binds to CAD, at least at this stage.

One can then wonder about the inclusion of both the modeling work and the chemistry in this manuscript and how much value do they add. I don't think they add any value to the biological findings.

Thank you for your comment. As discussed above, we plan to utilize the modeling work to identify the binding sites of RMY-186 on CAD. This will help confirm the direct interaction between the two, allowing us to further investigate the detailed mechanism by which RMY-186 enhances CAD ubiquitination. We hope you agree that these findings support the biological results.

The synthesis description is nearly identical (verbatim with no effort to even correct prior typos) in this manuscript as well as in reference 72. In the current manuscript the synthesis route is described rather succinctly with analytical data reported only for the final compound RMY-186.

Again, we thank you for pointing out the deficiencies in the method details for compound synthesis. We have corrected the spelling errors and included the structural characterization data for the intermediate compounds necessary for the synthesis of the final compound RMY-186 (lines 977-1037, pages 45-47).

The text has a lot of typos and at times, RMY-186 was misspelled as RNY-186 and the number indicated at RMY-183 in one instance.

We have corrected these typos and apologize for any inconvenience caused.

References 7, 14, 23, 49, 54, 56, 58, 63, 68, 73, and 74 are incomplete.

We have corrected the formatting errors in the reference section.

Responses to Reviewer #5

We are extremely grateful for your constructive suggestions, which have helped substantiate the conclusions. We have now carefully addressed all of the suggestions made by you, either by providing clarification or by performing new experiments. In particular, we have discussed the differences between previous studies on caspase-3-mediated CAD cleavage and our findings, validating conclusions on the distinct cleavage site. We also evaluated clinical samples with CAD-D1371 mutations from chemotherapy-resistant patients, confirming that heterozygous allele mutation cells show chemoresistance. Point-by-point responses to your comments are given below, with your comments given verbatim in plain italic type, and our responses in bold:

Reviewer #5 (Remarks to the Author):

Ma et al. reported that CDA degradation is an important step in chemotherapy-induced apoptosis in gastric and colon cancer cells. They reported that caspase-3 is a major protease that targets Asp1371 for CAD degradation. Expression of Asp1371 mutant CAD showed treatment resistance in xenografts and a mouse model of gastric cancer. Sequencing of treatment-resistant primary tumors identified two cases harboring somatic missense mutations at Asp1371, and these mutants also showed drug resistance in cell models. The authors identified a compound that induces CAD degradation and showed that this compound can overcome chemoresistance in vivo.

Duplication of results from a previous study

1. This is an interesting study showing that CAD degradation is essential for chemotherapy-induced apoptosis of cancer cells. However, the novelty of their finding that CAD is cleaved by caspase 3 is questionable because a previous paper has already reported this (Caspase-dependent cleavage of carbamoyl phosphate synthetase II during apoptosis. Huang M, Kozlowski P, Collins M, Wang Y, Haystead TA, Graves LM. Mol Pharmacol. 2002;61(3):569-77.). Therefore, the authors should cite this work and delete duplicate results (Fig2e~o and Figure 3) from the previous report.

We thank you for bringing the paper by Huang M et al. to our attention, as we were previously unaware of it. In fact, we also utilized staurosporine (STS) and doxorubicin (Dox) treatments they used to induce apoptosis during our revision (as suggested by Reviewer #2). We have also reproduced their findings, demonstrating that these two compounds can indeed induce the cleavage of CAD. In our revised manuscript, we have now cited their paper as a significant reference (current reference #39), where we describe their observations in detail (lines 297-303, page 14).

However, we would like to point out that Huang M et al. conducted their research

under conditions of IL-3 deprivation to identify CAD cleavage sites, which is fundamentally different from our study that used chemotherapy drugs such as 5-FU. As a result, the cleavage sites identified by them differ from those observed in our work. For instance, they reported that the D1143 residue, along with D1371, is necessary for caspase-3 cleavage of CAD. In contrast, we found that mutation of D1143 (CAD-D1143A) does not inhibit CAD cleavage when treated with 5-FU (as shown in the current Supplementary Fig. 3g). Moreover, Huang M et al. also suggest that varying apoptotic stimuli may lead to different degradation patterns of CAD.

In addition, the amount of caspase-3 protein used by Huang M et al. for in vitro cleavage of CAD was 0.25 μg , which is higher than the 0.1 μg we used. At the lower concentration of 0.1 μg , we were only able to observe cleavage of CAD at D1371 (Fig. 2f). It is noteworthy that we and others have suggested that higher concentrations of caspase-3 can result in non-specific recognition and cleavage of co-incubated protein substrates (PMID 32971525). This possibility of non-specific cleavage is also supported by the findings of Huang M et al., where they observed that the cleaved band of CAD does not align with those seen in vivo under IL-3 deprivation.

In summary, we have elucidated the mechanisms underlying CAD degradation during chemotherapy-induced apoptosis and highlighted its clinical significance. We sincerely hope that you will agree that our identification of cleavage sites should not be regarded as a duplicate of Huang M et al.'s findings.

2. The previous paper identified two cleavage sites by Caspase 3, which is inconsistent with the results of the current study. The authors should test another cleavage site and confirm the difference and discuss about the results.

As discussed above, the different sites identified by Huang M et al. may be due to the differing experimental conditions they employed. Again, we thank you for referencing the study by Huang M et al., as it significantly improves the quality of our paper.

Sequencing of patient samples

It is quite important but also should be carefully evaluated that the authors found CAD1371 mutations in primary treatment-resistant samples.

1. Figure 5a shows that mutant reads are ~50% of the total reads. This is not consistent with the percentage of mutant alleles in Figure 5b/c (12~15%).

Thank you for your comments. In fact, the approximately 50% mutant reads shown in Figure 5a of our initial submission do not represent the overall mutation allele frequency within the entire sample. This panel was created from a partial screenshot of the Integrative Genomics Viewer (IGV) software, specifically for one of the positive samples. To avoid any confusion, we have now provided the full IGV screenshot of the Whole Exome Sequencing (WES) data in Supplementary Fig. 5g-h, which are also shown below. We are also happy to provide any additional information you need (as stated in the "Data Availability" section).

Regarding the percentage of mutation, it is derived from the results of droplet digital PCR (ddPCR), which accurately measures mutation frequency at specific targeted sites. We chose to perform this ddPCR assay in addition to WES because WES may have certain inaccuracies in precisely quantifying specific loci.

Case#9

Case#14

2. Are these mutants heterozygous? Did the authors detect copy number loss (LOH) of this gene in the two patient samples?

Thank you for your insightful comment. Through WES and ddPCR analyses, we confirmed that the mutations are present in a heterozygous state in both patient samples (Case #9 and Case #14). In our analysis of the coverage depth in the CAD gene region (human reference genome hg19_chr2:242,742,132-242,783,918), we found no copy number variations between the samples (the sequencing depth was 20 GB).

3. If the mutants are heterozygous in the patient samples, the authors should test whether the heterozygous mutant allele is sufficient to acquire drug resistance in cell line/mouse models.

We appreciate your constructive comment regarding the investigation of the effects of the heterozygous CAD-D1371A mutation on the sensitivity of chemotherapy. We have now generated heterozygous CAD-D1371A (D1371A/+) knock-in MKN45 cells. As shown below (current Supplementary Fig. 4i), the sequencing results for the CAD locus in these cells display double, overlapping peaks, indicating the presence of both the wildtype and CAD-D1371A sequences. In contrast, MKN45 cells with either homozygous wildtype CAD (WT/WT) or homozygous CAD-D1371A (D1371A/D1371A) knock-in show a single peak at the CAD locus.

We next implanted these cells into nude mice and treated them with 5-FU. As shown below (Fig. 4g-h and Supplementary Fig. 4n-o), we found that xenograft tumors with the heterozygous CAD-D1371A mutation demonstrated approximately 25% increased resistance to 5-FU compared to wild-type CAD (evidenced by the tumor volume); however, this resistance was less pronounced than that observed in homozygous mutations, which exhibited about 47% resistance. Therefore, the CAD mutation exhibits haploinsufficiency, which helps to explain the chemoresistance observed in patients with the heterozygous CAD-D1371A mutation.

4. Searching the public mutation database (COSMIC) does not reveal any mutation at Asp1371 in any type of cancer samples ever sequenced, suggesting that these mutants are so rare. This should be mentioned in the Discussion section as the cohort analyzed in this study is too small.

We thank you for your insightful comment regarding the mutation frequency and cohort size, which was also noted by Reviewer #1. We admit that the mutation frequency of CAD-D1371A is rare, and have included a discussion on this in our revised manuscript (lines 527-534, page 24).

We would also like to point out that the low frequency of CAD-D1371A and the lack of records in the COSMIC database may be attributed to its presence in tissue samples from GC patients who have experienced failure of neoadjuvant chemotherapy. These samples are particularly difficult to obtain, as these patients typically do not undergo further surgery, making tissue collection challenging. The reason why it is difficult to obtain specimens from these patients who failed neoadjuvant chemotherapy is: GC patients requiring neoadjuvant therapy are typically locally advanced cases deemed unresectable by surgical evaluation. Once neoadjuvant chemotherapy fails, it implies these patients cannot undergo surgery according to surgical principles and can only continue treatment with alternative chemotherapy regimens, thus making specimens hard to obtain. These specimens can only be acquired when patients develop complications such as bleeding, obstruction, or perforation necessitating palliative surgery — this being the sole circumstance allowing sufficient specimen collection from such cases.

The scarcity of specimens further complicates our ability to perform robust statistical analyses, such as those evaluating overall survival (OS) or progression-free

survival (PFS). We have also added this discussion in our revised manuscript as well.

5. Did the authors find gene amplification of the CAD genes in cases without somatic mutations?

We appreciate your comment. We must acknowledge that, aside from identifying mutations with a certain abundance at the CAD-D1371 site, we are currently unable to determine the presence of CAD gene amplification. This limitation arises because all of our sequencing samples were obtained from formalin-fixed, paraffin-embedded (FFPE) tissues of patients who experienced failure of neoadjuvant chemotherapy. As mentioned earlier, collecting these samples in a short timeframe is quite challenging. In future studies, we plan to expand our WES sample size. For instance, we aim to compare CAD gene amplification between samples from patients who had successful versus failed neoadjuvant chemotherapy to address this issue.

Responses to Reviewer #6

We are extremely grateful for your constructive suggestions, which have helped substantiate the conclusions. We have now carefully addressed all of the suggestions made by you, either by providing clarification or by performing new experiments. In particular, we have clarified the impact of pyrimidine nucleotide metabolism and CAD cleavage site mutation on radiation-induced apoptosis. We also explored the relevance of RNA damage from chemotherapy, and elaborated on the pyroptosis induced by RMY-186 and 5-FU. Moreover, we confirmed that RMY-186 can enhance sensitivity to anti-PD-1-mediated immunotherapy. Point-by-point responses to your comments are given below, with your comments given verbatim in plain italic type, and our responses in bold:

Reviewer #6 (Remarks to the Author):

In this study, Ma et al. investigated the role of the pyrimidine synthesis pathway in the chemosensitivity of gastric tumor models. They found that chemotherapeutic drugs, such as 5-fluorouracil (5-FU), can promote caspase-3-mediated cleavage and subsequent degradation of CAD (carbamoyl-phosphate synthetase 2, aspartate transcarbamylase, and dihydroorotase), an enzyme involved pyrimidine biosynthesis. They go on to report that over-expression of CAD (or mutations in the Asp1371 cleavage site) can lead to “tumor resistance”. Moreover, they found mutations “related” to Asp1371 in tumor patients who underwent unsuccessful neoadjuvant chemotherapy. Finally, they identified a compound, RMY-186, that appears to enhance chemosensitivity. This study contains many interesting observations and, if correct, may ultimately lead to improved treatment options for GC and GRC. However, the presentation is at times confusing, and several of the main novel conclusions need stronger experimental support.

Specific points:

As a general comment, this manuscript contains a lot of data, but the photomicrographs of cells are often too small and hard to evaluate. Therefore, I was not able to fully evaluate some of the relevant experiments.

We thank you for your comment and have resized the photomicrographs of cells, including Fig. 1b, 4c, 5d-e and Supplementary Fig. 1b, 4d, in the revised manuscript to enhance clarity and facilitate evaluation.

1. Based on the results shown in Fig. 1, the authors conclude that supplementation of cells rU, dT and dC prevents 5-FU-induced apoptosis. It would be important to show specificity by examining effects on apoptosis induced by ionizing radiation.

We thank you for the suggestion regarding the impact of pyrimidine nucleotide

metabolism on apoptosis induced by ionizing radiation (IR). Following your advice, we first titrated the IR dosage on GC cells. As shown below (current Supplementary Fig. 1j), we found that 8 Gy of IR is sufficient to induce the cleavage of CAD. Similar to observations made with 5-FU treatment, the cleavage of CAD, along with increases in p53 and cleaved PARP, exhibited a dose-dependent response to IR.

Moreover, we discovered that restoring CAD protein levels through ectopic expression can rescue cells from IR-induced apoptosis (shown below, current Supplementary Fig. 1k). These results indicate that the *de novo* pyrimidine synthesis pathway plays a critical role in apoptosis mediated by IR.

2. I have some questions regarding the specificity of RMY-186 and conclusions derived from using this compound. The authors show that RMY-186 can promote the degradation of CAD, including in cells over-expressing CAD-D1371A (and other mutations in the cleavage site). Somewhat surprisingly treatment of GC and CRC cells with RMY-186 did not induce apoptosis in cells with significantly reduced levels of CAD. Moreover, RMY-186 enhanced sensitivity of both WT CAD and CAD-D1371A knock-in mutant cells to 5-FU. Why is this? According to the authors model, it would be expected that CAD-D1371A knock-in cells are (considerably) more resistant to combination treatment than CAD WT cells.

Thank you for your insightful comment. As you noted, RMY-186 does not induce apoptosis on its own, unlike 5-FU and other chemotherapeutic drugs. This is because the increased demand for nucleotides in GC cells only arises in the presence of

genotoxic stress, not under normal conditions (current Fig. 7a and Supplementary Fig. 7a). In fact, our previous studies have shown that knockdown of CAD in GC cells does not cause cell death, although it does impair growth. In contrast, when GC cells are treated with chemotherapy drugs, their fate—whether they undergo apoptosis or not—depends on the presence of CAD: when CAD cannot be degraded, such as with mutations at its D1371 residue (to either A, E, or Y; as shown below, current Fig. 4d and Fig. 5f-g), these cells become resistant to apoptosis induced by chemotherapeutic drugs.

Supporting this idea, during this revision, we present new data demonstrating that other chemotherapy drugs, such as oxaliplatin (Oxa), the combination of Oxa and RMY-186 can also promote the degradation of CAD, leading to apoptosis. Similar to the effects observed with 5-FU treatment, we found that the pro-apoptotic effects of Oxa can be blocked by the CAD-D1371A mutation, which stabilizes CAD during Oxa treatment. Conversely, RMY-186, by degrading CAD, enhances the apoptosis of these cells (shown below, current Supplementary Fig. 7c). Therefore, we hope you agree that RMY-186 does not directly induce apoptotic death, increases chemosensitivity by targeting and degrading CAD, in tumors cells.

As for the equal sensitivity of wildtype CAD and CAD-D1371A cells to chemotherapy drugs in the presence of RMY-186, this can also be attributed to the protein levels of CAD—both are almost completely degraded, as observed in these cell types (also shown below, current Fig. 6b and Supplementary Fig. 6c). This is because RMY-186 leads to the degradation of CAD, regardless of its D1371 mutation (current Supplementary Fig. 6d). As a result, RMY-186 equalizes the apoptotic rates of GC cells and tumors under 5-FU treatment. That is why we suggest combining RMY-186

**with chemotherapy drugs as a strategy to specifically target CAD-D1371 drug-resistant mutations.**

**We are also pleased to add that we have identified the potential binding sites of CAD for RMY-186 during this revision, which explains how RMY-186 promotes the degradation of CAD. We identified this site by conducting in-silico docking assays using the AutoDock Vina software, during which the structure of CAD was predicted using the CavityPlus platform (http://www.pkumdl.cn/cavityplus; PMID 29750256). After analyzing the surface feature matching, binding energy, and potential hydrogen bonds and non-covalent interactions, we identified four potential binding pockets on CAD:**

- **Pocket 1: Glu(E)513 and Arg(R)515**
- **Pocket 2: Asp(D)622 and Glu(E)630**
- **Pocket 3: Pro(P)741 and Trp(W)743**
- **Pocket 4: Glu(E)1214 and Arg(R)1221**

**We then individually mutated these pockets to determine the effects of RMY-186 on the promotion of CAD ubiquitination. As shown in the upper panel (current Fig. 6g and Supplementary Fig. 6k), we found that a double mutation of CAD-P741 and CAD-W743 (CAD-P741/W743A), which disrupts Pocket 3 (illustrated as a ribbon diagram in the lower panel), almost completely blocked the effects of RMY-186 on CAD ubiquitination.**

3. The authors state that CAD cleavage is required for 5-FU-induced apoptosis (line 320). This statement is too strong and not supported by the data presented (see point #2). They show a reduced effect of 5-FU on the size of xenografts, but they do not demonstrate that this is mediated by resistance towards apoptosis, and they most definitely fail to show a strict requirement of CAD for 5-FU-induced apoptosis.

Again, we thank you for the insightful comments (also raised by Reviewer #2 in minor point #2). Throughout our study, we utilized cleaved PARP as a marker to assess levels of apoptosis. For instance, we demonstrated that the ectopic expression of CAD significantly inhibited the cleavage of PARP, as shown in the upper panel (also listed as current Fig. 1h and Supplementary Fig. 1l). However, we acknowledge that we did not measure cleaved PARP to evaluate the impact of CAD mutation on 5-FU-induced apoptosis, and we apologize for it. We have now determined cleaved PARP levels to evaluate apoptosis in wild-type CAD and CAD-D1371A-expressing cells. As shown in the lower panel (current Fig. 4d and Supplementary Fig. 4g), the overexpression of CAD-D1371A exhibited resistance to apoptosis. We hope you will agree that these results, along with the data presented in our initial submission, support our conclusion that CAD cleavage is necessary for 5-FU-induced apoptosis.

4. Along those lines, the effect of 5-FU is complex and has been investigated for decades. For example, incorporation into RNA is responsible for the gastrointestinal toxicity of 5-FU in mice (Houghton JA, Houghton PJ, Wooten RS. Mechanism of induction of gastrointestinal toxicity in the mouse by 5-fluorouracil, 5-fluorouridine, and 5-fluoro-2'-deoxyuridine. *Cancer Res.* 1979 Jul;39(7 Pt 1):2406-13. PMID: 156065), and it has been reported that 5-FU incorporation into RNA but not DNA was associated with cell death (Geoffroy FJ, Allegra CJ, Sinha B, Grem JL. Enhanced cytotoxicity with interleukin-1 alpha and 5-fluorouracil in HCT116 colon cancer cells. *Oncol Res.* 1994;6(12):581-91. PMID: 7787251). The current study focuses on DNA damage (H2A.X), and it would be important to also investigate effects on RNA biology.

We thank you for your comments regarding the canonical function of 5-FU, and the references you provided. In our study, we focused on the effects of 5-FU on DNA damage, particularly assessing DNA-damage markers such as p-H2A.X. We agree that both DNA and RNA damage are important mechanisms of apoptosis (PMID

34265272), and have now cited the references provided by you. However, we would like to point out that research on the relationship between chemotherapy-induced RNA damage and apoptosis is currently limited, particularly due to the lack of available readouts.

Therefore, we examined the potential role of 5-FU in regulating the mRNA levels of CAD, as 5-FU can be incorporated into RNA through its metabolite FUTP, thereby interfering with RNA stability (PMID 12724731). Our data showed that the addition of actinomycin D (ActD), a transcription inhibitor, did not affect CAD mRNA levels in the presence of 5-FU (as indicated in the upper panel, current Supplementary Fig. 2b). Moreover, data from our initial submission demonstrated that the CAD mRNA level was not significantly impacted by various chemotherapy drugs, including 5-FU (as shown in the lower panel, current Supplementary Fig. 2a). Together, these results suggest that 5-FU does not modulate the mRNA levels of CAD.

5. The results from over-expression of CAD (both WT CAD-1371 mutants) need to be taken with a grain of salt since over-expression of caspase substrates can inhibit caspase activity.

We thank you for your comment regarding the potential inhibitory effects of CAD on caspase-3 when overexpressed. In fact, prior to determining the physiological function of CAD, we carefully titrated the amount of expression plasmid used for transfection in each GC cell line. For example, in HGC27 cells, we found that 0.3 μ g of plasmid per well in a 6-well dish effectively restores CAD expression (as shown in current Fig. 1c and Supplementary Fig. 1c), while for MKN45, HCT116 and SW480

cells, 0.4 μ g, 0.35 μ g and 0.45 μ g of plasmid are needed, respectively. As shown below (current Fig. 1h and Supplementary Fig. 1l), we found that CAD, when ectopically expressed in this manner, could be effectively cleaved by caspase-3 following treatment with 5-FU, which indicates intact caspase-3 activity.

To further address your comment, we introduced Raptinal, a direct activator of caspase-3, to evaluate the potential effects of CAD on the regulation of caspase-3. Similar to the findings under 5-FU treatment, Raptinal effectively induced the cleavage of CAD. As a result, we observed significant cleavage of PARP, which could also be inhibited by the restoration of CAD through ectopic expression. Importantly, the expression of CAD did not affect the cleavage of RhoGDI, another substrate of caspase-3 (PMID 10381642, PMID 11989976). These data suggest that CAD expression, at least in the context of our study, does not impair the enzymatic activity of caspase-3.

6. The authors' model predicts that mutating the caspase-3 cleavage site in CAD should lead to specific protection against 5-FU-induced apoptosis. This should be tested, for example, by examining the response of these cells to ionizing radiation (evaluating

H2A.X, PARP cleavage, etc.).

As discussed in point #1 above, we discovered that maintaining de novo pyrimidine synthesis can inhibit IR-induced apoptosis. Following your suggestion, we also investigated the effects of the CAD mutation on IR-induced apoptosis. As shown below (current Supplementary Fig. 4h), the expression of CAD-D1371A significantly reduced IR-induced genomic instability and apoptosis compared to wild-type CAD.

7. “CAD cleavage by caspase-3 is necessary and sufficient for chemosensitivity” (Fig. 4). This is a gross overstatement that should be toned down – at best the data show an effect, but it is not as strict as implied.

We thank you for the insightful comment regarding the role of CAD cleavage in mediating apoptosis across different chemotherapy drugs. In response to your comment, we have revised the statement from “CAD cleavage by caspase-3 is necessary and sufficient for chemosensitivity” to “CAD cleavage by caspase-3 is associated with chemosensitivity”.

Intrigued by your comment, we conducted additional experiments to investigate whether CAD cleavage occurs with other apoptosis-inducing agents besides 5-FU and nucleotide analogs. We found that CAD cleavage also takes place following treatment with other apoptosis inducers, such as Oxaliplatin (which forms cross-links with DNA, thereby inhibiting DNA replication and transcription), Venetoclax (a Bcl-2 inhibitor, as recommended by Reviewer #2), Staurosporine (a non-selective protein kinase inhibitor), and Raptinal (described above) (see current Supplementary Fig. 2f). These results highlight a broader association between CAD cleavage and the chemosensitivity of GC cells.

8. It is stated (for example, in line 464) that RMY-186 when used in combination with 5-FU induces apoptosis in cells carrying the CAD-D1371A knock-in mutation, yet later on observations are cited that chemosensitized cells appear to undergo pyroptosis. This is confusing and needs better documentation and explanation/discussion.

Again, we appreciate your insightful comments. Indeed, we found that when RMY-186 is combined with 5-FU, the treated GC cells, particularly those carrying the CAD-D1371A mutant, undergo a mixed type of cell death that includes both apoptosis and pyroptosis. Although the precise role of CAD in mediating pyroptosis remains unclear, previous studies have reported similar instances of plasticity between different types of cell death.

For example, during chemotherapy in lung and pancreatic tumors, it was observed that tumor cells with high $\beta 5$ integrin expression contribute to chemotherapy resistance. These cells achieve this by remodeling sphingolipid metabolism, effectively suppressing the NLRP3-caspase-1 inflammasome to inhibit pyroptosis, which is the predominant type of cell death in this scenario. By inhibiting ceramidase in the sphingolipid metabolism pathway, these cells can restore the pyroptotic response during chemotherapy treatment (PMID 36630483). Additionally, in the ischemic and hypoxic conditions found in the centers of breast tumors, Stat3 can become phosphorylated, which interacts with PD-L1 and promotes its nuclear translocation. The nuclear-localized PD-L1-Stat3 complex then enhances the transcription of GSDMC, activating pyroptosis on the basis of apoptosis induced by TNF α (PMID 32929201). Furthermore, it has been shown that combining NLRP3 activators with mitochondrial (OXPHOS) inhibitors can switch apoptosis induced by mitochondrial inhibitors to pyroptosis by sequestering cytochrome c, which is essential for apoptosis (PMID 39571574). These findings suggest that the occurrence and transition between different types of cell death can be modulated in response to various stressors or therapeutic agents through multiple pathways.

We would also like to point out that we discuss the pyroptosis of GC cells because it enhances the response to immunotherapy, which is highly significant for treating GC.

Pyroptosis has been suggested to release damage-associated molecular patterns (DAMPs) that help activate and increase the infiltration of anti-tumor immune cells. Consistently, triggering pyroptosis could be an effective strategy to reverse the immunosuppressive microenvironment of GC, potentially leading to more successful immunotherapeutic treatments (PMID 32188940, PMID 32188939). To draw a decisive conclusion, we investigated the effects of RMY-186 and 5-FU on the efficacy of the anti-PD-1 antibody, a classic immune checkpoint inhibitor (ICI), in suppressing GC in *Cldn18-ATK Ccd^{D1371A/D1371A}* mice. We found that the combination of RMY-186 and 5-FU significantly enhances the effects of the anti-PD-1 antibody on GC suppression. This triple treatment also notably increased CD3⁺ and CD8⁺ T cell infiltration compared to other treatments (assessed by staining for CD3 and CD8a, respectively; see current Supplementary Fig. 7p-r), indicating a strong enhancement of immune responses in these tumors. Therefore, we hope you will agree with the importance of RMY-186-induced pyroptosis in cancer therapy.

9. Along these lines, the model (Fig. 8) is not clear at all and overstates what can be concluded from the available data. First, it implies a far greater role of CAD-cleavage in the execution of apoptosis than what the data support. Second, the proposed

mechanism of action of RMY-186 is only depicted for “high heterogenous tumors” – why? I suggest to present a model that focuses on the difference between cells with WT and cleavage-resistant CAD. Finally, why does caspase-3 cleavage and degradation of CAD cause apoptosis and not pyroptosis in the left panel, whereas RMY-186-induced degradation of CAD supposedly causes pyroptosis? This is particularly confusing in light of the authors’ data supporting induction of apoptosis (Fig. 7: cleaved PARP, p53, pH2A.X). There are clearly many other mechanisms at play to account for the observations and the current illustration is not very helpful.

Thank you for your insightful comments regarding the clarity of the model. As outlined in points #2, #3, and #8 above, our study focuses on the clinical challenge of chemotherapy resistance in highly metabolically heterogeneous tumors. These tumors are characterized by either high CAD expression or the presence of CAD-D1371 mutations, which confer resistance to caspase-3 cleavage. For these resistant tumors, we propose that combining the targeted CAD-degrading small molecule RMY-186 with chemotherapy successfully restores sensitivity to treatment. In contrast, tumors exhibiting low metabolic heterogeneity do not present as many challenges with existing treatment strategies.

We also acknowledge that the precise mechanism by which CAD degradation promotes pyroptosis remains unclear. In response to your comments, we have replaced the solid arrows indicating the role of CAD degradation in inducing pyroptosis with dashed lines to better reflect the current uncertainty regarding this mechanism.

Minor comment:

The English is often a bit rough and should be corrected throughout the ms. For example, in the abstract the sentence “By searching for a compound that capable for degradating the CAD-Asp1371 mutant, we discovered, ...” should be corrected to: “By searching for a compound that is capable of degrading the CAD-Asp1371 mutant, we discovered ...”

Thank you for your kind suggestion. We have carefully reviewed our manuscript and made edits to enhance its readability, as you suggested.

Responses to Reviewer #1

Reviewer #1 (Remarks to the Author):

Major critique has been satisfactorily addressed.

We sincerely appreciate your positive feedback on our revisions and additional data. Your insightful suggestions greatly strengthened this work.

Responses to Reviewer #2

Reviewer #2 (Remarks to the Author):

The authors have responded thoroughly to all of my critiques. I have no further comments and congratulate them on an interesting set of observations.

Thank you for your encouraging evaluation of our responses; your expertise has been invaluable in improving this research.

Responses to Reviewer #3

Reviewer #3 (Remarks to the Author):

After reading through the comments and updates provided by the authors, I am satisfied with their explanations and changes to their manuscript. The manuscript is formatted in a much tidier way with the additional figures in the supplemental information section.

Your professional recognition of our revisions is deeply encouraging, and we value the crucial role your feedback played in refining this paper.

Responses to Reviewer #4

Reviewer #4 (Remarks to the Author):

There heterogeneity within gastric cancer (GC) and colorectal cancer (CRC) is well established and is attributed to among other factors, the dysregulation of metabolic

reprogramming as key to resistance to chemotherapy-mediated DNA damage . In their manuscript, Ma. J. et al, showed that CAD, a fusion of glutaminase (GLN), glutaminase-dependent carbamoyl phosphate synthase (CPS-2), aspartate transcarbamoylase (ATC), and dihydroorotase (DHO), is the first rate-limiting enzyme in the de novo pyrimidine synthesis pathway, and importantly a caspase-3 substrate. CAD needs to be cleaved by caspase-3 at Asp1371 prior to ubiquitin-dependent degradation. In patient samples, mutations at Asp1371 confer resistance to several chemotherapeutic agents. Through a screening campaign and a modest SAR they identified RMY-186 as an inhibitor of CAD which, when combined with 5-FU overcomes resistance to chemotherapy. These findings and others offer CAD as a promising therapeutic target and possibly a marker for therapeutic resistance.

Looking at the reviews including mine, and I am only the Organic Chemist, I feel like the authors have done a great job addressing all issues raised by our colleagues and myself. This is very apparent in the extensive number of experiments and figures that were subsequently added.

As pointed out by my colleagues, the manuscript size is enormous, and the number of figures goes beyond the space allowed. Consolidating figures, discussions, and deciding what goes to supporting section will be a daunting task.

There are still some typos as well as rough expressions which should be addressed.

We have thoroughly reviewed the manuscript, carefully proofread the text, and polished the language to ensure clarity and precision throughout the revised version.

Reference 82 is incomplete and Page number should be added J Vis Exp. (78): 50541 (2013)

We have corrected the formatting error in the reference section.

As far as I am concerned, the chemistry is sound and well controlled.

We are all aware of metabolic reprogramming and the major role that it plays in cancer resistance and evasiveness. This paper is another addition to the field in which colleagues working on cancer therapeutics should know about. The addition of a small molecule inhibitor of CAD will offer a new tool with which to further study the pathway.

The overall merit for publication in this prestigious journal is met.

We're grateful for your constructive comments and acknowledgment of our supplemental materials. Your guidance significantly enhanced our study.

Responses to Reviewer #5

Reviewer #5 (Remarks to the Author):

The authors have responded appropriately to all of the reviewer's comments. I have no further comments on this revision.

We appreciate your favorable assessment of the additional data and responses; your rigorous review process has substantially improved this manuscript.

Responses to Reviewer #6

Reviewer #6 (Remarks to the Author):

The authors have responded very well to the criticism raised during the previous review, provided important clarification and extensive new data supporting their conclusions. Therefore I recommend rapid publication of this interesting manuscript at this time.

Thank you for acknowledging our efforts in addressing your comments. Your discerning observations were instrumental in elevating the quality of this work.